# Latitudinal patterns in stabilizing density dependence of forest communities

Lisa Hülsmann[1,2,3 ✉], Ryan A. Chisholm[4], Liza Comita[5,6], Marco D. Visser[7], Melina de Souza Leite[8], Salomon Aguilar[9], Kristina J. Anderson-Teixeira[9,10], Norman A. Bourg[10], Warren Y. Brockelman[11,12], Sarayudh Bunyavejchewin[13], Nicolas Castaño[14], Chia-Hao Chang-Yang[15], George B. Chuyong[16], Keith Clay[17], Stuart J. Davies[18], Alvaro Duque[19], Sisira Ediriweera[20], Corneille Ewango[21], Gregory S. Gilbert[22], Jan Holík[23], Robert W. Howe[24], Stephen P. Hubbell[25], Akira Itoh[26], Daniel J. Johnson[27], David Kenfack[28], Kamil Král[23], Andrew J. Larson[29,30], James A. Lutz[31], Jean-Remy Makana[21], Yadvinder Malhi[32], Sean M. McMahon[18,33], William J. McShea[10], Mohizah Mohamad[34], Musalmah Nasardin[35], Anuttara Nathalang[11], Natalia Norden[36], Alexandre A. Oliveira[8], Renan Parmigiani[8], Rolando Perez[9], Richard P. Phillips[37], Nantachai Pongpattananurak[38], I-Fang Sun[39], Mark E. Swanson[40], Sylvester Tan[34], Duncan Thomas[41], Jill Thompson[42], Maria Uriarte[43], Amy T. Wolf[44], Tze Leong Yao[35], Jess K. Zimmerman[45], Daniel Zuleta[18] & Florian Hartig[2]

Numerous studies have shown reduced performance in plants that are surrounded by neighbours of the same species[1,2], a phenomenon known as conspecific negative density dependence (CNDD)[3]. A long-held ecological hypothesis posits that CNDD is more pronounced in tropical than in temperate forests[4,5], which increases community stabilization, species coexistence and the diversity of local tree species[6,7]. Previous analyses supporting such a latitudinal gradient in CNDD[8,9] have suffered from methodological limitations related to the use of static data[10–12]. Here we present a comprehensive assessment of latitudinal CNDD patterns using dynamic mortality data to estimate species-site-specific CNDD across 23 sites. Averaged across species, we found that stabilizing CNDD was present at all except one site, but that average stabilizing CNDD was not stronger toward the tropics. However, in tropical tree communities, rare and intermediate abundant species experienced stronger stabilizing CNDD than did common species. This pattern was absent in temperate forests, which suggests that CNDD influences species abundances more strongly in tropical forests than it does in temperate ones[13]. We also found that interspecific variation in CNDD, which might attenuate its stabilizing effect on species diversity[14,15], was high but not significantly different across latitudes. Although the consequences of these patterns for latitudinal diversity gradients are difficult to evaluate, we speculate that a more effective regulation of population abundances could translate into greater stabilization of tropical tree communities and thus contribute to the high local diversity of tropical forests.

Explaining patterns of diversity across space and time is a fundamental goal of ecology[16]. Among those patterns, the latitudinal gradient in tree species diversity is particularly notable[17]. A central explanation for the exceptionally high local diversity in tropical moist forests is that their temporally stable and productive conditions allow natural enemies, such as herbivores and pathogens, to be more specialized and damaging[5,18], with the result that conspecific neighbours—by virtue of their shared natural enemies—exert more negative effects on a target tree individual than do heterospecific neighbours[19]. Similarly to intraspecific resource competition, specialized enemies can thus create a stabilizing mechanism[20], often referred to as conspecific negative density dependence (CNDD)[3], that should prevent the

dominance of any particular tree species and therefore allow species coexistence[6,7,21,22]. First proposed by Janzen and Connell five decades ago[4,5], CNDD mediated by specialized enemies is one key hypothesis for explaining the maintenance of greater local tree species diversity in tropical forests[23,24].

After several decades of research, it is well established that CNDD is widespread in both tropical and temperate forests[1,2]. Nevertheless, its effect on community composition and large-scale biodiversity patterns is still debated[25,26]. Meta-analyses on CNDD, based mostly on seed and seedling survival in field experiments, have found no variation in CNDD with latitude[1,2,23,27], possibly because of limited comparability among studies[2]. The few studies that have directly examined large-scale

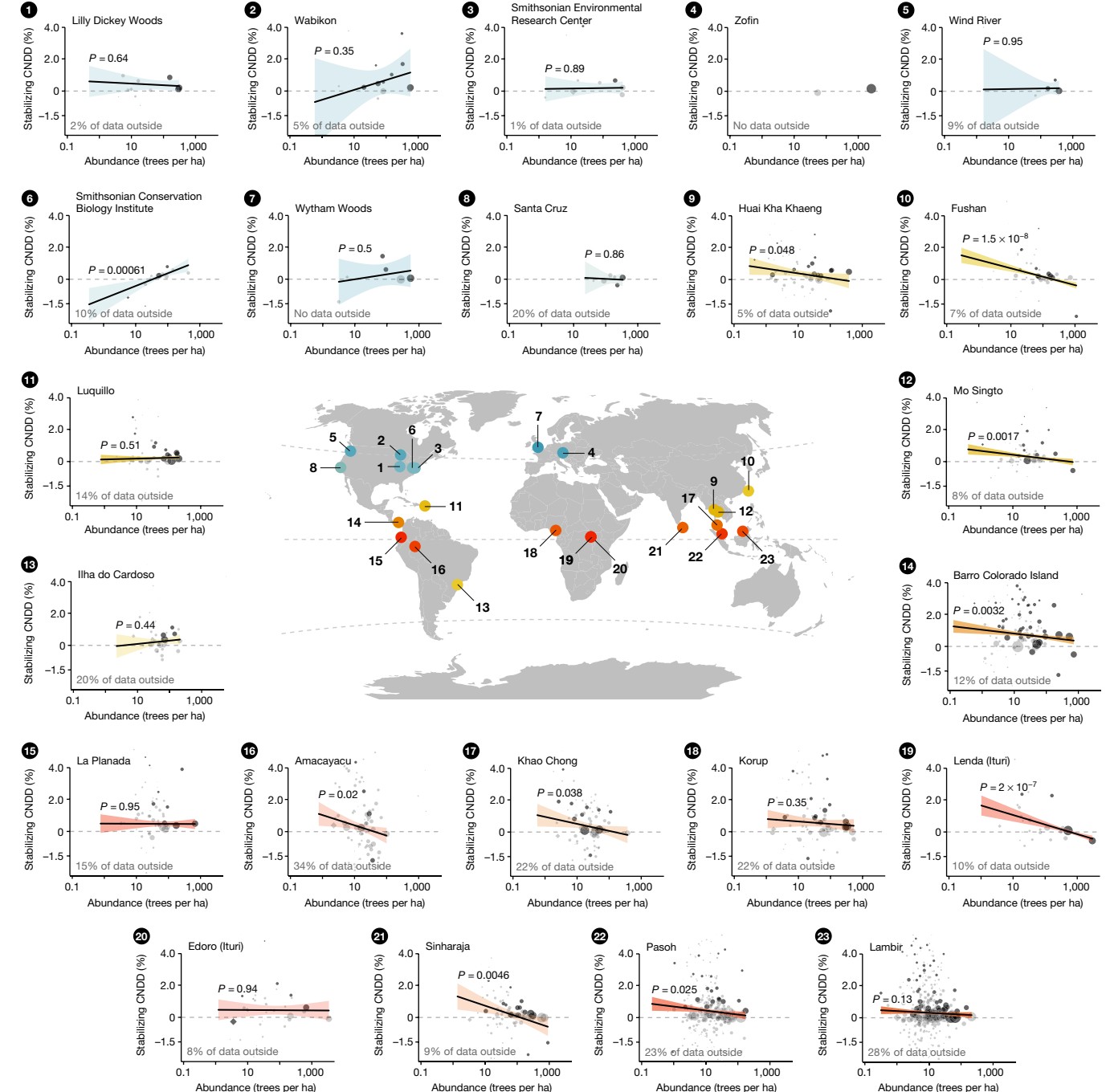

**Fig. 1 | Estimated stabilizing CNDD in tree mortality plotted against species abundance at the 23 forest plots, along with plot locations.** Points in small panels indicate CNDD estimates and abundances (number of trees with DBH ≥ 1 cm per hectare) of individual species or species groups. Larger point sizes indicate lower uncertainty (variance) in CNDD estimates. Points in dark grey indicate effects that are statistically significantly different from zero (with $\alpha = 0.05$). Circles are individual species; diamonds are rare species analysed jointly as groups of rare trees or rare shrubs. Because of the high variation in CNDD estimates, not all species-specific estimates can be shown, but the proportion of data that is represented by the estimates outside the plotting area is indicated for each site. The regression lines, 95% confidence intervals (CI) and $P$ values are based on meta-regression models fitted independently per site (except for the Zofin site, for which too few estimates were available). Dashed horizontal lines indicate zero stabilizing CNDD. Locations of forest sites and CNDD-abundance relationships are coloured by latitude (gradient from tropical forests in red–orange to subtropical forests in yellow–green and temperate forests in blue). Stabilizing CNDD is defined as the relative change (in %) in annual mortality probability (relative average marginal effect; rAME) induced by a small perturbation in conspecific density (one additional conspecific neighbour with DBH = 2 cm at a one-metre distance) while keeping total densities constant. Positive numbers indicate a relative increase in mortality with an increase in conspecific density; that is, CNDD.

geographical variation in CNDD have assessed larger tree sizes and reported a pronounced increase in CNDD with decreasing latitude[8,9]. However, these latitudinal CNDD patterns have been attributed to statistical artefacts related to the use of static data[10–12,28,29]. As a result,

there is still no conclusive evidence about if and how CNDD differs between tropical and temperate forests.

Here, we analyse latitudinal CNDD patterns using dynamic forest inventory data (longitudinal tree survival data from repeated censuses,

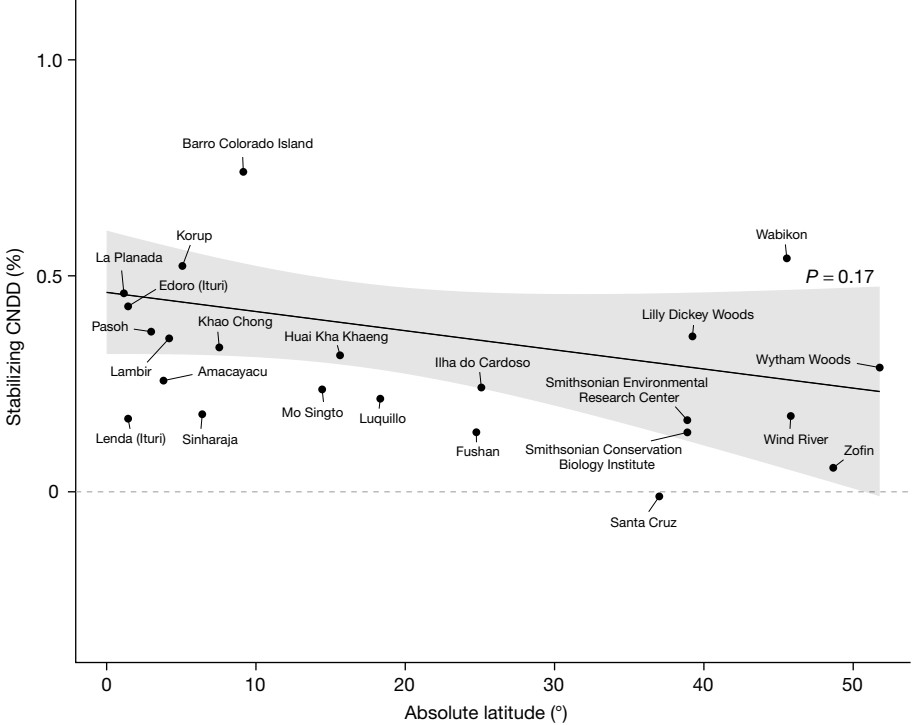

**Fig. 2 | Evaluation of the first hypothesized pattern, whereby the average strength of stabilizing CNDD across species becomes greater towards the tropics.** The estimated relationship of stabilizing CNDD to absolute latitude indicates that average species CNDD does not become significantly stronger toward the tropics ($P = 0.17$). The regression line and 95% CI are predictions from the meta-regression model fitted with species-site-specific CNDD estimates ($n = 2,534$ species or species groups from 23 forest sites) including absolute latitude as a predictor ('mean species CNDD model'; see Table 1a). Black points are mean CNDD estimates per forest site from meta-regressions fitted separately for each forest site without predictors (as in Fig. 4); note that these points are not the direct data basis for the regression line. The dashed horizontal line indicates zero stabilizing CNDD. Stabilizing CNDD is defined as in Fig. 1; for the same plot with alternative definitions of CNDD see Extended Data Figs. 4 and 5.

Extended Data Table 1) from 23 large (6–52 ha) forest sites from the ForestGEO network[30], covering a gradient from the tropics to the temperate zone (Fig. 1). We used recently developed best-practice statistical methods for measuring and comparing CNDD and making inferences about stabilization and species coexistence[10,25,31] (Methods). We fitted flexible species-site-specific mortality models and quantified CNDD as the relative change in the mortality probability of saplings (small trees with a diameter at breast height (DBH) of at least 1 cm and less than 10 cm) induced by a small perturbation in conspecific density while keeping total densities (both measured as basal area) constant ('stabilizing CNDD')[20] (Methods). By adjusting for total density, our estimate of 'stabilizing CNDD' is equivalent to the difference between CNDD and heterospecific negative density dependence (HNDD) in previous studies[3,32], and serves as a proxy for the frequency dependence of population growth rates[33]. We then aggregated estimates of stabilizing CNDD and patterns therein using multilevel meta-regressions to account for the different uncertainties in CNDD estimates resulting from different sample sizes among species[34]. Using this framework, we assessed latitudinal patterns in (i) the average strength of stabilizing CNDD (Fig. 2), (ii) its effects on species abundances (Fig. 3) and (iii) its interspecific variability (Fig. 4), thereby testing three predictions (each described in a section below) arising from the hypothesis that CNDD is more influential for maintaining local tree species diversity in the tropics.

## No latitudinal trend in average CNDD

According to the Janzen–Connell hypothesis, the average strength of stabilizing CNDD across species should become greater at lower latitudes[4,5,24], but we found no support for this hypothesis, although stabilizing CNDD was widespread. Averaged across species, the mortality of small trees increased with conspecific density at all but one site (Figs. 1 and 2, CNDD < 0 for Santa Cruz), with an average relative annual mortality increase of 6.64% when increasing conspecific density from the first to the third quantile for each species (95% confidence interval (CI): 2.80 to 10.62%; Extended Data Fig. 1). However, when comparing the strength of CNDD across latitudes, we found no significant trend: in the tropics, a perturbation in conspecific density (expressed by one additional conspecific neighbour with a DBH of 2 cm at 1 m distance; see 'Quantification of conspecific density dependence' in Methods) led to a relative increase in annual mortality probability of 0.41% (0.31 to 0.51% CI; calculated at 11.75° absolute latitude; Fig. 2). In temperate forests, the corresponding value was 0.26% (0.06 to 0.47% CI; calculated at 45° absolute latitude). Although the increase in mortality is slightly less in temperate than in tropical forests, the association of CNDD with latitude was not statistically significant ($P = 0.17$, assessed through meta-regression, Table 1a) and the absolute change in stabilizing CNDD with latitude was small relative to the variation in CNDD across species and abundances (see next subsections and Figs. 1, 3 and 4).

The lack of a latitudinal gradient in average CNDD was statistically robust (see 'Robustness tests' in Methods). When tree status (alive or dead) or conspecific densities were randomized, our analysis pipeline of mortality models and meta-regression revealed neither spurious CNDD nor noteworthy patterns of CNDD across latitudes (Extended Data Fig. 2a and Extended Data Table 2). Moreover, we obtained qualitatively the same result—that is, no latitudinal trend in average species CNDD—when statistically influential species were removed from the meta-regression (Extended Data Fig. 3a and Extended Data Table 2) and when two alternative definitions of CNDD were analysed

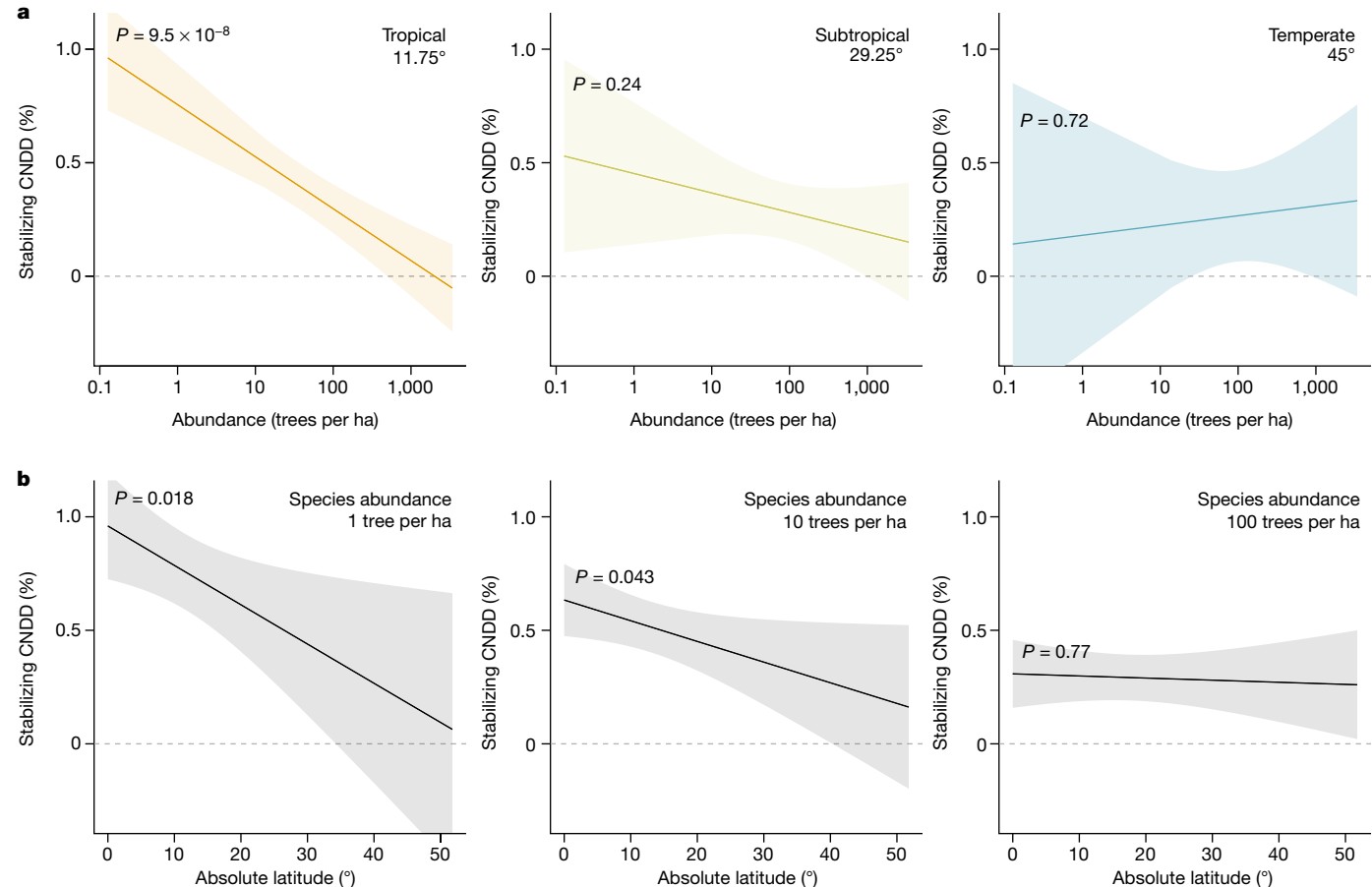

**Fig. 3 | Evaluation of the second hypothesized pattern, whereby CNDD more strongly regulates species abundances and thus community structure in the tropics. a**, The estimated relationship of stabilizing CNDD to absolute latitude and species abundance indicates that species-specific CNDD is considerably stronger for rare than for common species in tropical forests ($P = 9.5 \times 10^{-8}$), whereas species in subtropical and temperate forests show no statistically significant association between CNDD and species abundance ($P = 0.24$ and $P = 0.72$, respectively). **b**, Consequently, stabilizing CNDD of species with low abundance (here, one tree per hectare) is stronger in tropical than in temperate forests ($P = 0.018$), whereas CNDD of species with high abundance (here, 100 trees per hectare) shows no latitudinal gradient ($P = 0.77$). Note that a caveat to the comparison in **b** is that species abundance distributions and total community abundance change with latitude so that an abundance of

one tree per hectare is not necessarily biologically comparable across latitudes. The regression lines and 95% CI are predictions from the meta-regression model ($n = 2,534$ species or species groups from 23 forest sites) including absolute latitude, species abundance and their interaction as predictors ('abundance-mediated CNDD' model; see Table 1b). Predictions in **a** are shown for the centres of three latitudinal geographic zones, with the tropical zone ranging between 0° and 23.5° absolute latitude, the subtropical between 23.5° and 35° and the temperate between 35° and 66.5°. Species abundance is quantified as the log-transformed number of trees per hectare. Confidence intervals and P values are obtained by refitting the model with data centred at the respective latitude or abundance value. Dashed horizontal lines indicate zero stabilizing CNDD. Stabilizing CNDD is defined as in Fig. 1; for the same plots with alternative definitions of CNDD, see Extended Data Figs. 4 and 5.

(Extended Data Figs. 4a and 5a and Extended Data Table 3). These alternative definitions were calculated as (1) the absolute change in mortality, which we consider less relevant for fitness, but which may nevertheless be instructive if base mortality rates are independent of latitude; and (2) the (relative) change in mortality at low conspecific densities, following the invasion criterion for coexistence, which refers to the ability of a species to increase in abundance when rare[35].

Our results corroborate previous studies that found that stabilizing CNDD (that is, the negative effect of being close to conspecifics) was widespread across forest tree communities[1,2], but they do not support previous reports of a pronounced latitudinal gradient in average CNDD[8,9]. This discrepancy can be explained by various factors, including our focus on mortality rather than on recruitment. We argue that our use of robust statistical methods and dynamic rather than static data[10–12,28,36] is more reliable than previous analyses, suggesting that a latitudinal gradient in average CNDD at the sapling stage is absent or at least weaker than previously reported.

## Stronger CNDD for rare tropical species

A second pattern that has been interpreted as more effective stabilizing control of species abundances and thus as a proxy for the importance of CNDD for community structure is stronger CNDD for rare species[3,8,13,21,37]. Consistent with this, we found a marked latitudinal difference in the association between species abundance and stabilizing CNDD when expanding the meta-regression to include species abundance and allowing the relationship with abundance to be moderated by latitude ($P = 0.017$ of the interaction, Table 1b). In tropical tree communities, CNDD decreased significantly with species abundance ($P = 9.5 \times 10^{-8}$; Fig. 3a); CNDD was stronger for rare species (0.76%, 0.59 to 0.92% CI, for a species with an abundance of one tree per hectare) and weaker for common species (0.30%, 0.19 to 0.40% CI, for a species with an abundance of 100 trees per hectare). With increasing latitude, this association weakened. In temperate forests, there was no significant relationship between species abundance and CNDD ($P = 0.72$; Fig. 3a), and CNDD was actually slightly higher for common species

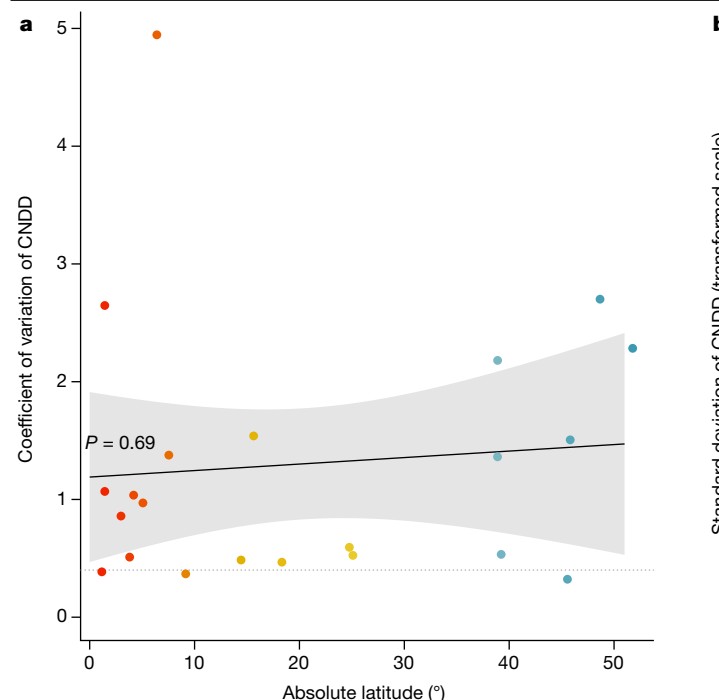

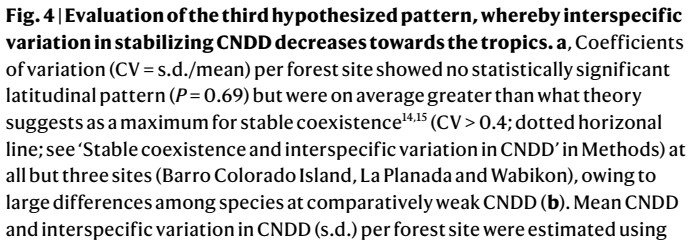

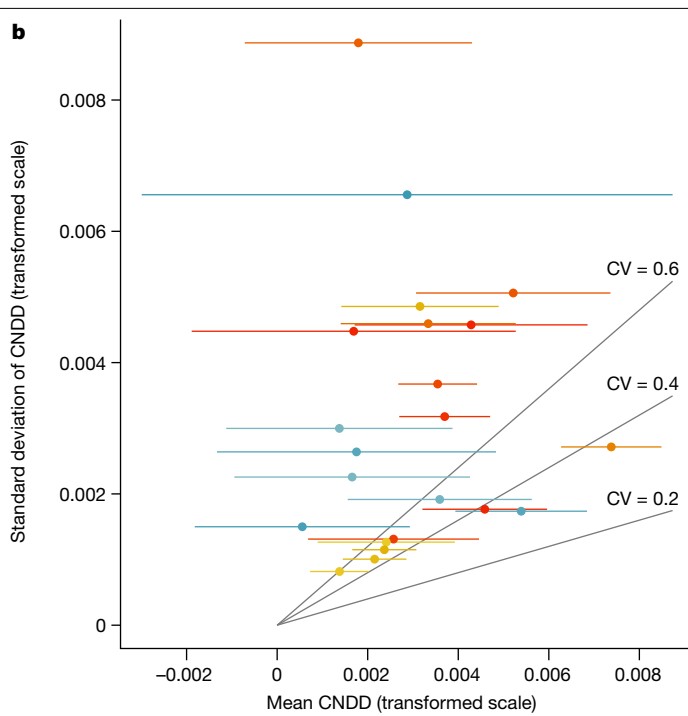

**Fig. 4 | Evaluation of the third hypothesized pattern, whereby interspecific variation in stabilizing CNDD decreases towards the tropics. a**, Coefficients of variation (CV = s.d./mean) per forest site showed no statistically significant latitudinal pattern ($P = 0.69$) but were on average greater than what theory suggests as a maximum for stable coexistence[14,15] (CV > 0.4; dotted horizontal line; see 'Stable coexistence and interspecific variation in CNDD' in Methods) at all but three sites (Barro Colorado Island, La Planada and Wabikon), owing to large differences among species at comparatively weak CNDD (**b**). Mean CNDD and interspecific variation in CNDD (s.d.) per forest site were estimated using meta-regressions without predictors fitted separately for each forest site. Points are coloured by latitude (gradient from tropical forests in red–orange to subtropical forests in yellow–green and temperate forests in blue). The regression line, 95% CI and $P$ value in **a** are based on a linear regression model. Grey lines in **b** indicate different CV values. Note that we excluded one site for which the average CNDD was less than 0 (Santa Cruz; Fig. 2), because positive conspecific density dependence is expected to be destabilizing, irrespective of species differences. Stabilizing CNDD is defined as in Fig. 1, but here means and s.d. are shown at the transformed scale; that is, log(rAME + 1).

(0.27%, 0.07 to 0.47 CI) than for rare species (0.18%, −0.33 to 0.69% CI). From these patterns it follows that CNDD of rare and intermediate abundant species is stronger in tropical than in temperate forests ($P = 0.018$ and $P = 0.043$ for species with an abundance of 1 and 10 trees per hectare, respectively; Fig. 3b), whereas CNDD of common species shows no latitudinal gradient ($P = 0.77$ for species with an abundance of 100 trees per hectare).

Although associations between CNDD and species abundance have been reported in previous studies, all but one study[8] analysed CNDD at only a single site, mostly in tropical forests. Of these, some reported stronger CNDD for rare species[3,38], others showed stronger CNDD for common species[28,39] and still others showed no association[40]. We attribute these apparently inconsistent previous results to strong between-site variability, which is evident in our data as well (Fig. 1). Our multi-site approach allows us to see past the noise and detect the signal of a large-scale pattern of stronger CNDD for rare versus common species in the tropics, but not in the temperate zone (Fig. 3a). The use of dynamic data also allows us to make more statistically robust inferences about CNDD and its association with species abundance[11,12] (Extended Data Figs. 2b,c, 3b,c, 4b,c and 5b,c and Extended Data Tables 2 and 3). Our study thus provides stronger evidence than previously available that a correlation between CNDD and species abundance exists in tropical but not in temperate forests.

We believe that the most likely explanation for the latitudinal change in the correlation between stabilizing CNDD and species abundance is that CNDD is more effective at controlling tree species abundances in the tropics[3,8,13,21,37]. To challenge this interpretation, we sought alternative explanations for the observed pattern. In particular, we considered life history strategies, which can correlate with both species rarity and CNDD[13,41–43] (see Supplementary Fig. 1a,b) and could thus act as a confounder. Accounting for life history strategies (approximated by species' demographic rates, maximum size (stature) or trade-offs therein) in the meta-regression, however, did not change the association between CNDD and species abundance in the tropics (Extended Data Table 4), ruling out those factors as important confounders. In addition to confounding, the observed pattern could also arise under reverse causality, in which species abundance controls CNDD. A possible mechanism could be that pathogen loads for common species saturate in space, thus rendering local variation in conspecific density inconsequential for infection and hence mortality probabilities.

## CNDD varies considerably between species

Theoretical studies have suggested that interspecific variation in CNDD can increase competitive differences or the risk of local extinctions from demographic stochasticity and thus reduce or even reverse the diversity-enhancing effects of CNDD[14,15]. Thus, if interspecific CNDD variation were lower in tropical than temperate forests, this would provide another avenue whereby CNDD could contribute to latitudinal differences in local tree species diversity. No previous study, however, has empirically quantified this pattern.

To test for latitudinal differences in interspecific variation in CNDD, we used meta-regressions fitted separately for each site to estimate the mean and the latent (true) standard deviation (s.d.) of species-specific CNDD. Crucially, this approach allows us to distinguish interspecific variation in CNDD from sampling uncertainty; that is, the random sampling error of CNDD estimates[34]. We then calculated the coefficient of variation (CV = s.d./mean) of CNDD per site and analysed latitudinal patterns therein. Interspecific variation of CNDD, quantified as CV, showed no significant association with latitude ($P = 0.69$, Fig. 4a). Interestingly

**Table 1 | Estimates from the meta-regressions testing the first and second hypothesized latitudinal patterns in stabilizing CNDD in tree mortality**

| Model | Characteristic | Beta | 95% CI | P value |
|---|---|---|---|---|
| **(a) Average species CNDD** $\sigma_r = 0.0018 \; \sigma_s = 0.0054$ | Intercept | 0.004087 | 0.003072, 0.005102 | **2.9×10⁻¹⁵** |
| | tLatitude | −0.000044 | −0.000107, 0.000019 | 0.17 |
| **(b) Abundance-mediated CNDD** $\sigma_r = 0.0018 \; \sigma_s = 0.0053$ | Intercept | 0.007527 | 0.005870, 0.009183 | **5.3×10⁻¹⁹** |
| | tLatitude | −0.000172 | −0.000315, −0.000030 | **0.018** |
| | tAbundance | −0.000990 | −0.001353, −0.000626 | **9.5×10⁻⁸** |
| | tLatitude:tAbundance | 0.000035 | 0.000006, 0.000064 | **0.017** |

We fitted two models for the species-site-specific CNDD estimates (n = 2,534 species or species groups from 23 forest sites): (a) absolute latitude as a predictor ('average species CNDD' model); and (b) absolute latitude, species abundance and their interaction as predictors ('abundance-mediated CNDD' model). Species abundance was measured by log-transformed number of trees with DBH ≥1 cm per hectare. Predictors were transformed (t), that is, centred at abundance = 1 tree per hectare and absolute latitude = 11.75°, so that main effects for abundance and latitude assess slopes and respective significance tests for rare, tropical species. Stabilizing CNDD is defined as in Fig. 1. For the models, CNDD estimates (rAMEs) were log-transformed after adding 1 to improve normality assumptions, so that CNDD as the relative change in annual mortality probability in per cent induced by one additional conspecific neighbour can be calculated from the model coefficients as $100 \times (e^{\beta_0 + \beta_1 x \cdots} - 1)$. Predictions of the models are shown in Figs. 2 and 3. $\sigma_r$ and $\sigma_s$ are the estimated standard deviations of random intercepts for CNDD among sites and species in sites, respectively. Bold P values are statistically significant at a significance level of 0.05.

though, the s.d. of CNDD was of a similar magnitude to community average CNDD across the forest sites (Fig. 4a,b), implying a CV on the order of 1. In simulation studies[14,15], CNDD settings with CV > 0.4 have tended to reduce rather than to stabilize species diversity (see 'Stable coexistence and interspecific variation in CNDD' in Methods). Among the 22 sites where species on average exhibited CNDD (all except the Santa Cruz site), this threshold (CV > 0.4) was exceeded at all but 3 sites (Barro Colorado Island, La Planada and Wabikon). We note, however, that there are several reasons why the CV parameters in the simulation models cannot be directly matched to our empirical estimates. One of them is that temporal variability in CNDD, possibly caused by fluctuations of herbivore and pathogen populations, might inflate the empirically measured CV above its long-term average.

## Discussion

Our results support the conclusion of numerous previous studies that the effects of conspecific neighbours on tree survival tend to be negative (CNDD)[1,2]. Contrary to long-held ecological conjectures, however, we found a latitudinal gradient consistent with the Janzen–Connell hypothesis in only one of the three CNDD patterns we tested. Most notably, the average strength of CNDD did not increase significantly toward the tropics (Fig. 2 and Table 1a). In addition, tree species in tropical communities did not experience more homogenous levels of CNDD than temperate ones did (Fig. 4a), which theoretically could have led to more effective stabilization through reduced fitness differences in the tropics[14,15]. However, we did find that CNDD correlates with species rarity in tropical but not in temperate forests (Fig. 3 and Table 1b), which suggests that CNDD could have a stronger role in structuring species abundance distributions in the tropics. The drivers and implications of stronger CNDD for rare to intermediate abundant species in tropical versus temperate forests merit closer consideration.

Assuming that species abundances are at least partly controlled by CNDD, the association of strong CNDD with species rarity in the tropics might be interpreted as an indication of more efficient control of tropical tree species abundances through self-limitation[21,37], despite average CNDD being comparable across latitudes. This interpretation is broadly consistent with the ideas of Janzen and Connell—with the nuance that the effects of specialized enemies are not necessarily stronger overall in the tropics but have greater effectiveness in controlling species abundances and thus, potentially, community assembly. A possible explanation for why species abundances are less effectively controlled by CNDD in temperate forests is that other mechanisms, such as alternative stabilizing mechanisms, dispersal, immigration or disturbances, are stronger in temperate forests and override the effects of CNDD[14,44]. When evaluating these conjectures, we caution that such

a direct causal link and its direction between CNDD and species rarity remains to be established. Although we ruled out confounding by differences in life history strategy (Extended Data Table 4), the possibility of other unobserved confounding effects or reverse causality remains and should be considered in future studies.

Our finding that rarer species experience stronger CNDD in the tropics (Fig. 3a), and therefore CNDD weakens for species at rare and intermediate abundances towards the temperate zone (Fig. 3b), motivates further research targeted at the underlying mechanisms. Identifying these mechanisms and showing that their effects differ between the tropical and the temperate zone could provide strong independent evidence for the idea that CNDD regulates tropical species abundances more strongly. This would require, first, a better understanding of how specialized natural enemies and resource competition generate CNDD[45] and how CNDD interacts with other processes (for example, facilitation[46]), and then comparisons of these mechanisms in coordinated global experiments[47]. A further consideration is that species abundances are controlled by processes that occur during the entire demographic cycle, rather than being controlled only by mortality during the sapling life stage, as considered here. It is possible that CNDD analyses of other vital rates and life stages, particularly earlier ones, would lead to stronger CNDD and different patterns and conclusions[20], because the interaction between ontogenetic and demographic processes might change with latitude. This possibility could be investigated using dynamic seedling data along latitudinal gradients, ideally with good coverage of temperate tree species, which are naturally less represented in latitudinal studies. By accumulating CNDD estimates across different vital rates and life stages, we could also move closer to the ultimate goal of estimating CNDD in a species' overall fitness and population growth rate[22,35].

Additional to the latitudinal change in the correlation with rarity, we found high interspecific variation in CNDD at all latitudes (Fig. 4a). Based on previous simulation studies, this variation would be high enough to offset the stabilizing effect of CNDD at the community level[2,14,15]. We believe that there is an urgent need to better understand the effect of CNDD on community stability and coexistence in the presence of interspecific, spatial and temporal variability. Interspecific variation in CNDD has been linked to species-specific characteristics such as mycorrhizal type[40] and life history strategy[41], as well as to population-level diversity of pathogen resistance genes[48], but our estimate of interspecific variation is also likely to reflect temporal variation due to complex host–enemy dynamics and resource competition in varying environments[49]. Future empirical and theoretical analyses should investigate in more detail the conditions under which interspecific variation in CNDD weakens or reverses the stabilizing effect of CNDD on species diversity and whether the competitive disadvantage

associated with stronger CNDD might be offset by functional traits or life history strategies[6,33,50]. For example, there are indications that trees of species with stronger CNDD grow faster[41] (but see also Extended Data Table 4), which might result in faster population growth when a species is rare[37].

In the context of the Janzen–Connell hypothesis, we interpret our results as partial support for the idea that CNDD contributes to the latitudinal gradient in tree species diversity. More specifically, our results suggest a novel, refined interpretation of this classic idea: the influence of specialized natural enemies—and, more broadly, intraspecific resource competition—might not be stronger on average in tropical than in temperate forests, but their effects might exert stronger controls on species abundances in the tropics. Therefore, we speculate that unless interspecific variability in CNDD overrides its stabilizing effect, CNDD might contribute more strongly to the maintenance of local tree species diversity in the tropics.

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

¹Ecosystem Analysis and Simulation (EASI) Lab, University of Bayreuth, Bayreuth, Germany. ²Theoretical Ecology, University of Regensburg, Regensburg, Germany. ³Bayreuth Center of Ecology and Environmental Research (BayCEER), University of Bayreuth, Bayreuth, Germany. ⁴Department of Biological Sciences, National University of Singapore, Singapore, Singapore. ⁵School of the Environment, Yale University, New Haven, CT, USA. ⁶Smithsonian Tropical Research Institute, Panama City, Panama. ⁷Institute of Environmental Sciences, Leiden University, Leiden, The Netherlands. ⁸Department of Ecology, University of São Paulo, São Paulo, Brazil. ⁹Forest Global Earth Observatory, Smithsonian Tropical Research Institute, Panama City, Panama. ¹⁰Conservation Ecology Center, Smithsonian's National Zoo & Conservation Biology Institute, Front Royal, VA, USA. ¹¹National Biobank of Thailand (NBT),

National Science and Technology Development Agency, Bangkok, Thailand. [12]Institute of Molecular Biosciences, Mahidol University, Nakhon Pathom, Thailand. [13]Thai Long Term Forest Ecological Research Project, Department of Forest Biology, Faculty of Forestry, Kasetsart University, Bangkok, Thailand. [14]Instituto Amazónico de Investigaciones Científicas Sinchi, Bogotá, Colombia. [15]Department of Biological Sciences, National Sun Yat-sen University, Kaohsiung, Taiwan. [16]Department of Plant Science, University of Buea, Buea, Cameroon. [17]Department of Ecology and Evolutionary Biology, Tulane University, New Orleans, LA, USA. [18]Forest Global Earth Observatory, Smithsonian Tropical Research Institute, Washington, DC, USA. [19]Departamento de Ciencias Forestales, Universidad Nacional de Colombia Sede Medellín, Medellín, Colombia. [20]Department of Science and Technology, Uva Wellassa University, Badulla, Sri Lanka. [21]University of Kisangani, Kisangani, Congo. [22]Environmental Studies Department, University of California, Santa Cruz, Santa Cruz, CA, USA. [23]Department of Forest Ecology, Silva Tarouca Research Institute, Brno, Czech Republic. [24]Cofrin Center for Biodiversity, Department of Biology, University of Wisconsin-Green Bay, Green Bay, WI, USA. [25]Department of Ecology and Evolutionary Biology, University of California, Los Angeles, Los Angeles, CA, USA. [26]Graduate School of Science, Osaka Metropolitan University, Osaka, Japan. [27]School of Forest, Fisheries, and Geomatics Sciences, University of Florida, Gainesville, FL, USA. [28]Global Earth Observatory (ForestGEO), Smithsonian Tropical Research Institute, Washington, DC, USA. [29]Department of Forest Management, University of Montana, Missoula, MT, USA. [30]Wilderness Institute, University of Montana, Missoula, MT, USA. [31]Department of Wildland Resources, Utah State University, Logan, UT, USA. [32]Environmental Change Institute, School of Geography and the Environment, University of Oxford, Oxford, UK. [33]Smithsonian Environmental Research Center, Edgewater, MD, USA. [34]Sarawak Forest Department, Kuching, Malaysia. [35]Forest Research Institute Malaysia, Kepong, Malaysia. [36]Instituto de Investigación de Recursos Biológicos Alexander von Humboldt, Bogotá, Colombia. [37]Department of Biology, Indiana University, Bloomington, IN, USA. [38]Department of Forest Biology, Faculty of Forestry, Kasetsart University, Bangkok, Thailand. [39]Department of Natural Resources and Environmental Studies, National Donghwa University, Hualien, Taiwan. [40]School of the Environment, Washington State University, Pullman, WA, USA. [41]Department of Botany and Plant Pathology, Oregon State University, Corvallis, OR, USA. [42]UK Centre for Ecology & Hydrology, Bush Estate, Penicuik, UK. [43]Department of Ecology, Evolution & Environmental Biology, Columbia University, New York, NY, USA. [44]Department of Biology, University of Wisconsin-Green Bay, Green Bay, WI, USA. [45]Department of Environmental Science, University of Puerto Rico, Rio Piedras, USA. ✉e-mail: lisa.huelsmann@uni-bayreuth.de

## Methods

### Overview

We used repeated census data from 23 large forest sites around the globe (Fig. 1) to analyse latitudinal patterns in stabilizing CNDD following a three-step approach. First, we fitted species-site-specific mortality models from repeated observations of individual trees. Second, we used these models to quantify CNDD for each species and site using an estimator designed to maximize robustness, comparability and relevance for fitness and stabilization. Third, we used meta-regressions to consider three distinct latitudinal patterns in CNDD derived from the hypothesis that CNDD is more influential for maintaining local tree species diversity in the tropics. Robustness of the analysis pipeline was validated by model diagnostics and randomization.

This approach is based on recently developed best-practice statistical methods for estimating CNDD. Crucially, the use of dynamic mortality data allowed us to avoid the statistical pitfalls of previous CNDD studies, in particular with regard to analyses of the static relationship of number of saplings to number of adults, in which the null hypothesis is a positive linear relationship but regression dilution flattens this relationship and thus biases analyses towards finding CNDD, especially for rare species[10–12,28,29]. By fitting mortality models in which the null hypothesis is no relationship between survival and number of conspecific neighbours, we ensure that any regression dilution has a conservative effect by reducing CNDD estimates. We also addressed other previously identified limitations of CNDD analyses; namely, nonlinear and saturating CNDD (see 'Species-site-specific mortality models'), the comparability of CNDD among species and sites (see 'Quantification of conspecific density dependence') and the extent to which CNDD estimates are meaningful for stabilization and species coexistence[10,25,31].

All analyses were conducted in R v.4.2.1 (ref. 51).

### Forest data

The data used in this study were collected at 23 sites with permanent forest dynamics plots that are part of the Forest Global Earth Observatory network (ForestGEO)[30] (Fig. 1 and Supplementary Notes), in which all free-standing woody stems with a diameter of at least 1 cm at 1.3 m from the ground (DBH) are censused. We stipulated that for plots to be suitable for analysing tree mortality in response to local conspecific density, they should be at least a few hectares in size with at least two censuses available (that is, longitudinal data on individual trees). The plots for which we obtained data vary in size between 6 ha and 52 ha (Supplementary Table 1), with between 9,718 and 495,577 mapped tree individuals at each site. Censuses have been performed with remeasurement intervals of approximately five years (Supplementary Table 1). The census data collected for each individual include species identity, DBH, spatial coordinates and status (alive or dead).

For the mortality analyses, we selected observations of all living trees of non-fern and non-palm species with DBH < 10 cm in one census and follow-up data in a consecutive census (Extended Data Table 1). We then statistically analysed how tree mortality (measured by the status 'dead' or 'alive' in the consecutive census) depends on local conspecific density and potential confounders of this relationship (see 'Species-site-specific mortality models'). We focused on saplings (small trees between 1 cm and 10 cm DBH), on the assumption that CNDD effects are most pronounced in earlier life stages[52,53].

For tree individuals with more than one stem, the individual was considered 'alive' if at least one of the stems was alive and 'dead' if all stems were dead. The DBH of multi-stem trees was calculated from the summed basal area of all stems. For trees with multiple stems at different coordinates, coordinates of the main stem were used. For the forest site Pasoh, where every stem was treated as an individual (information on which stems belong to the same tree was unavailable), we used observations of individual stems.

Observations of trees or stems were excluded when information on coordinates, species, status or date of measurement was missing. Individuals classified as morphospecies were kept and analysed as the respective morphospecies. Status assignments were checked for plausibility and corrected if necessary (for example, trees found to be alive after being recorded as dead in a previous census were set to 'alive'). If trees or stems changed their coordinates or species between censuses, the most recent information was used.

### Definition of local conspecific density

Most previous CNDD studies[3,32] have estimated separate effects for CNDD and HNDD. In the context of the Janzen–Connell hypothesis, in which CNDD is a promoter of species diversity, however, we are interested mainly in the difference between CNDD and HNDD, because only a detrimental effect of neighbouring conspecifics that exceeds the effect of any kind of neighbour (that is, irrespective of its species identity) can lead to a stabilizing effect at the population level[6,20]. We refer to this effect, that is, to the difference between CNDD and HNDD, as 'stabilizing CNDD'. This effect is more appropriate when estimating the degree of self-limitation for a tree species.

Because CNDD and HNDD are both estimated with uncertainty (characterized by the standard error), previous analyses that separately estimated CNDD and HNDD often faced challenges when formally testing whether conspecific effects are significantly more negative than are heterospecific effects[25]. Here, we circumvent this problem by estimating the effect of conspecific density, adjusted (in a multiple regression) for total tree density, which is the sum of conspecific and heterospecific density[54]. Defined in this way, the estimated effect (slope) for conspecific density in the regression corresponds to the effect of CNDD minus HNDD in previous studies[55,56] (for details, see Supplementary Methods).

Local conspecific and total densities around each focal tree were calculated as the number of neighbouring trees ($N$) or their basal area (BA) at the census preceding the census at which tree status was modelled. We considered neighbouring trees of all sizes at distances up to 30 m[54] and discarded focal trees that were within 30 m of the plot boundaries. A decrease of neighbourhood effects with increasing distance was considered using two alternative decay functions:

$$\text{exponential:} \quad f(d_k) = e^{-\frac{1}{\mu}d_k}$$

$$\text{exponential–normal:} \quad f(d_k) = e^{-\frac{1}{\mu^2}d_k{}^2}$$

with $d_k$ being the distance between a focal tree and its neighbour $k$, and the distance decay parameter $\mu$ defining how far neighbourhood effects extend on average.

The estimator for local density ($N$ or BA), the shape of the decay kernel (exponential or exponential–normal) and its parameter $\mu$ were optimized through a grid search, optimizing the fit of the mortality models (see next section). The parameter $\mu$ was optimized jointly for all species but separately for conspecific and total densities following the idea that the two effects are caused by different agents and thus may act at different spatial scales. We tested all four combinations of density definitions ($N$ or BA, with exponential or normal distance decay) varying $\mu$ between 1 and 25 m in 2-m steps. Our selection criterion was the sum of the log likelihood (LL), calculated using the set of species for which all models converged ($n_{\text{species}} = 2{,}500$). The highest overall LL was achieved when local densities were measured as BA with an exponential distance decay and $\mu = 3$ and 17 for conspecific and total density, respectively (Supplementary Fig. 2). This definition of local densities also resulted in an average area under the curve (AUC) comparable with the overall AUC optimum (0.68; difference = 0.001). To ensure that the joint optimization of $\mu$ for all species did not induce a bias that correlated with the main predictors, that is, latitude and

species abundance, we further examined species-specific optima of $\mu$ for those species for which the grid search yielded a distinct optimum of the log likelihood. We found no pattern with respect to latitude and species abundance (Supplementary Fig. 3), justifying the use of a joint optimization.

## Species-site-specific mortality models

We used binomial generalized linear mixed models (GLMMs) with a complementary log-log (cloglog) link to model the tree status ('dead' or 'alive') as a function of conspecific density conD, total density totD and tree size DBH, which were added as potential confounder or precision covariates[57]. The advantage of the cloglog link over the more traditional logit link is that the cloglog allows better accounting for differences in observation time $\Delta t$ (see Supplementary Table 1) through an offset term[58].

Because evidence suggests that CNDD could be nonlinear and in particular saturating[10,25], we used generalized additive models (GAMs) with thin plate splines[59] to allow for flexible nonlinear responses of all predictors. When the observations covered more than one census interval, 'census' was included as a random intercept. In sum, we model the status $Y_{ij}$ of observation $i$ in census interval $j$ as a binomial random variable $Y_{ij} \sim \text{Binom}(\text{Pr}(y_{ij}=1))$, where

$$\log(-\log(1-\text{Pr}(y_{ij}=1))) = \beta_0 + f_{\text{conD}}(x_{\text{conD}}) + f_{\text{totD}}(x_{\text{totD}})$$
$$+ f_{\text{DBH}}(x_{\text{DBH}}) + u_j + \log(\Delta t)$$

Here, $\text{Pr}(y_{ij}=1)$ is the mortality probability of observation $i$ in census interval $j$, $f_k$ is the smooth function of the predictor $x_k$, conD, totD and DBH are the predictor variables, $\beta_0$ is the intercept term, $u_j$ is the random intercept for census interval $j$ with $u_j \sim N(0, \sigma_u^2)$ and $\Delta t$ is the census interval length in years.

GAM smoothness selection was performed using restricted maximum likelihood estimation (REML). Basis dimensions of smoothing splines were kept at modest levels ($k = 10$) but were reduced when the number of unique values (nvals) in a predictor was less than 10 ($k = $ nvals − 2). Models were fitted with the function gam() from the package mgcv[60] (v.1.8-40).

In this set-up, we fitted species-site-specific mortality models for all species that had at least 20 alive and dead status observations each and at least 4 unique conspecific density values with a range that included the value used to calculate average marginal effects (see 'Quantification of conspecific density dependence'). The species that did not fulfil these criteria and those for which no convergence was achieved (overall 63.2% of the species) were fitted jointly in one of two groups—rare shrub species and rare tree species (Extended Data Table 1)—following the assumption that different growth forms may differ in their base mortality rate. This allows us to at least consider very rare species for our analyses, even if these species do not contribute to the results to the same extent as species with more observations do. The growth form of each tree species ('shrub' or 'tree') was derived from a species' maximum tree size. If the maximum of the average DBH of the six largest trees or stems of each species per census was more than 10 cm, a species was considered a tree, and otherwise it was considered a shrub[61,62].

## Quantification of conspecific density dependence

On the basis of the species-site-specific mortality models, we then quantified how a change in conspecific density affects mortality probability. The challenge here is that the nonlinear link in the GLMMs implies that effects at the scale of the linear predictor can translate nonlinearly to the response scale (mortality rates) when the estimated intercept differs between individual species and sites[31]. To obtain an estimate of the strength of stabilizing CNDD that is nonetheless comparable among species and sites, we calculated the average marginal effect (AME) of a small perturbation of conspecific density on mortality probability[63] at the response scale. We derived both absolute and relative

AME (aAME and rAME, respectively), which can be interpreted as the average absolute (% per year) and relative (%) change, respectively, in mortality probability caused by the increase in conspecific density. In meta-analysis and econometrics, aAME is also known as the average risk difference, and rAME + 1 as the average risk ratio[64,65].

To obtain aAME and rAME, we first calculated the absolute and relative effect of one additional conspecific neighbour on the mortality probability (response scale) for each observation $i$:

$$\text{aME}_i = p_{i,\text{conD}_i+1} - p_{i,\text{conD}_i}$$

$$\text{rME}_i = \frac{p_{i,\text{conD}_i+1}}{p_{i,\text{conD}_i}} - 1 = \frac{p_{i,\text{conD}_i+1} - p_{i,\text{conD}_i}}{p_{i,\text{conD}_i}}$$

Here, $p_i$ is the mortality probability at the response scale and $\text{conD}_i$ is the observed local conspecific density. The subscript $\text{conD}_i+1$ denotes the new conspecific density, which is obtained by adding one conspecific neighbour with DBH = 2 cm at a one-metre distance, a relatively small perturbation that was within the range of observed conspecific densities even for rare species. A larger perturbation in conspecific densities could create extrapolation problems. For each observation, $\text{aME}_i$ and $\text{rME}_i$ were calculated using observed conspecific densities. Likewise, confounders—that is, total density, DBH and census interval—were kept at observed values, and the interval length was fixed at one year. As an alternative quantification of density dependence that links to theoretical considerations from coexistence theory[7] (invasion criterion[35]), we quantified CNDD at low conspecific densities by setting $\text{conD}_i = 0$ and again increasing it by one additional conspecific neighbour with DBH = 2 cm at a one-metre distance. As a further alternative, we calculated CNDD as the change in mortality resulting from a change in conspecific density from the first to the third quantile of observed conspecific densities per species to estimate how important CNDD is effectively for small tree mortality. It must be noted that values from this latter metric should not be compared between species (or sites), because the change in conspecific density is different for each species and tends to increase with species abundance.

Individual marginal effects ($\text{aME}_i$ and $\text{rME}_i$) were averaged over all observations per species to obtain average marginal effects[31]. Because there is no analytical function to forward the uncertainty of the GAM predictions to the response scale, we estimated uncertainties; that is, sampling variances $v_{lm}$, and significance levels for species-site-specific aAME and rAME by simulation. To this end, we simulated 500 sets of new model coefficients from a multivariate normal distribution with the unconditional covariance matrix of the fitted model, calculated aAME and rAME for each set[66] and used quantiles of the simulated distributions to approximate sampling variances and significance levels of CNDD estimates.

In our results, we concentrate our discussion on rAME because we consider relative changes in mortality to be ecologically more meaningful than absolute changes. The reason is that the relevance of an increase in mortality for a species' fitness strongly depends on its base mortality rate. Vice versa, if CNDD effects exist, it is to be expected that they are higher in absolute terms for species that already have higher absolute mortality rates. Moreover, given that species-specific mortality rates may also correlate with species abundance and latitude, the use of absolute mortality rates is likely to be more prone to confounding. To be comparable with previous studies, which commonly use absolute effects, results for the two main meta-regressions are also presented for the absolute effects; that is, aAME estimates (Extended Data Fig. 4 and Extended Data Table 3).

## Meta-regressions for CNDD patterns

To test for latitudinal patterns in stabilizing CNDD, we fitted meta-regressions[34,67] using the species-site-specific CNDD estimates.

The advantage of these models is that they simultaneously account for the uncertainties in aAME and rAME estimates (sampling variances)—much like measurement error models—as well as heterogeneity among sites and species through a multilevel model:

$$AME_{lm} = b_0 + r_l + s_{lm} + e_{lm} + f(\text{predictors})$$

$$r_l \sim N(0, \sigma_r^2)$$

$$s_{lm} \sim N(0, \sigma_s^2)$$

$$e_{lm} \sim N(0, v_{lm})$$

Here, $AME_{lm}$ is the average marginal effect for site $l$ and species $m$, $b_0$ is the intercept, $r_l$ is the random effect for site $l$ (normally distributed with $\sigma_r^2$), $s_{lm}$ is the random effect of species $m$ (normally distributed with $\sigma_s^2$) and $e_{lm}$ is the uncertainty of the individual estimates (normally distributed with the species-site-specific sampling variance $v_{lm}$). Omitting the random effects would lead to inappropriate estimates because it does not consider the true interspecific variation in species' CNDD. To improve the normality assumption of the residuals of the meta-regressions, rAMEs were log-transformed after adding 1 before calculating the sampling variances (see above); aAME remained untransformed.

Depending on the respective prediction to be evaluated, we used different meta-regression models. To evaluate latitudinal patterns in average CNDD and in the association of CNDD and abundance, we fitted multilevel models to all species-site-specific estimates (see model formula above): the first including absolute latitude as a predictor (Fig. 2 and Table 1a) and the second also including log-transformed species abundance and its interaction with latitude (Fig. 3 and Table 1b).

Absolute latitude was calculated as the distance (in degrees) to the equator. This metric does not distinguish between the northern and southern hemispheres and is commonly used as a proxy for the current and past bio-climatic variables that are assumed to underlie most latitudinal biological patterns[68,69]. We calculated the abundance of each tree species per site as the number of all living trees (or stems, for the Pasoh site) with DBH ≥ 1 cm per hectare on the entire plot. Abundance for the two groups of rare species (rare trees and rare shrubs) was calculated as the average of species abundances within the respective group. The predictors were centred at abundance = 1 tree per hectare and absolute latitude = 11.75°, so that main effects reflect slopes and respective significance tests for rare tropical species (Table 1).

We also separately fitted meta-regressions for each site with species as a random intercept: first, without any predictor to obtain mean CNDD and its s.d. among species per site (Figs. 2 and 4); and then with species abundance as a predictor to illustrate site-specific relationships of CNDD and abundance (Fig. 1).

AMEs calculated for species-specific interquantile ranges were aggregated in a global meta-regression with random intercepts for sites and species within sites to obtain a global average of CNDD and assess its importance for small tree mortality (Extended Data Fig. 1).

Models were fitted with REML using the functions rma.mv() and rma() from the package metafor[70] (v.3.4-0) for the global and site-specific cases, respectively.

## Robustness tests

Statistical assumptions of the mortality models were verified on the basis of simulated residuals generated with the package DHARMa (v.0.4.6)[71]. Distributional assumptions and residual patterns against predictors were assessed visually, revealing no critical violations of assumptions and a consistently good model fit. To verify that no additional unobserved local confounders, particularly habitat effects, were affecting the relationship between conspecific density

and mortality, we tested each mortality model for spatial autocorrelation using the package DHARMa (ref. 71). After adjusting $P$ values for multiple testing using the Holm method, significant spatial autocorrelation was detected in only seven models, or 0.28% of all species–site combinations, which means that there is no indication that local species-specific CNDD estimates were affected by spatial pseudo-replication.

Model diagnostics for the meta-regressions were based on standardized residuals and visual assessments. Because of the unbalanced design (more tropical than temperate species; see Supplementary Fig. 1c), we performed additional robustness tests by identifying influential species-site-specific CNDD estimates and refitting the two main meta-regression models (see Table 1) with a reduced dataset without these observations. We removed 99 CNDD estimates that had Cook's distances larger than 0.005 in the abundance-mediated CNDD model[72]. Meta-regressions fitted with these reduced datasets revealed similar patterns and significance levels (Extended Data Fig. 3 and Extended Data Table 2).

To evaluate the robustness of the entire analysis pipeline with respect to potential abundance- and latitude-related biases[11,12], we repeated all steps of the analysis (mortality models, average marginal effects and meta-regressions) with two randomizations of the original dataset (similar tests highlighted biases in a previously described pipeline[8], see also refs. 11,12). We randomized (1) observations of tree status within each species, thus removing any relationship between mortality and predictors but maintaining species-level mortality rates; and (2) observations of local conspecific density within each species, thus removing the relationship between mortality and conspecific density but maintaining the relationships between mortality and confounders. Meta-regressions applied to these randomized datasets revealed close to zero CNDD and no considerable patterns with latitude or species abundance (Extended Data Fig. 2 and Extended Data Table 2). When randomizing tree status, rare species exhibited minimally, but significantly, stronger CNDD, but the effect sizes varied by orders of magnitude from those observed in the original dataset. We therefore consider our results robust to statistical artefacts related to species abundance and latitude.

In addition, not only statistical biases but also alternative explanations could create a spurious correlation between CNDD and species abundance. To test this, we included potential confounders for this relationship in the 'abundance-mediated CNDD' model. Following the idea that fast-growing tree species with short life spans (that is, lower survival rates) tend to be rarer[43]—a pattern also observed across the 23 forest sites analysed here (Supplementary Fig. 1a,b)—and at the same time may experience stronger CNDD[41], we considered two sets of predictors that are proxies for different life history strategies, namely: (1) species-specific growth and survival rates; and (2) species-specific values along two demographic trade-off axes[73,74]. Species-specific growth was calculated as the median of the annual DBH increment, log-transformed after adding 1. For survival, we calculated mean annual survival rates (based on the intercept of a GLM similar to the mortality models for CNDD but without predictors) and applied a logit-transformation. Both rates were standardized within sites (that is, subtracting the mean and dividing by the s.d.) to account for differences in the realized demographic spectrum between sites. The demographic trade-offs reflect the two axes 'growth–survival' and 'stature–recruitment' and were adapted from a procedure described previously[73] using species-specific growth and survival rates (as described before) and the species' maximum size (stature), calculated as the log-transformed 90th percentile of the DBH, again standardized within sites. In both cases, we included main effects of the two predictors and their interaction. Accounting for life history strategies did not change the patterns obtained, and species abundance and CNDD were still strongly and statistically significantly correlated in tropical forests (Extended Data Table 4).

## Stable coexistence and interspecific variation in CNDD

If CNDD varies strongly among species and the resulting interspecific fitness differences are not compensated by equalizing mechanisms[6,33], the stabilizing advantage of CNDD may not promote diversity. One study[14] suggested, on the basis of simulations, that the number of species maintained strongly drops when the coefficient of variation (CV = s.d./mean) for CNDD is above 0.4 (see the second figure in that study); that is, the stronger CNDD becomes, the more interspecific variation it enables. Similarly, another study[15] found considerably fewer species with increasing standard deviations of CNDD supporting a comparable threshold of CV = 0.4 (s.d. = 0.2 at mean CNDD = 0.5; see the second figure in that study). Another study[75], which also investigated the effect of interspecific variation in CNDD, identified no threshold for stable coexistence, which is most likely to be caused by the relatively small variation in CNDD that this study tested (see the second figure in that study). Although it is not entirely clear whether the threshold of CV = 0.4 is truly due to the magnitude of fitness differences or to the fact that some species tend to have almost no CNDD when interspecific variation becomes large, the consistency of this threshold, despite different implementations of CNDD[14,15], provides a starting point for evaluating the relevance of CNDD for community assembly. We estimated true interspecific variation of CNDD within forest communities fitting site-specific meta-regressions without predictors (see 'Meta-regressions for CNDD patterns'), which are particularly helpful in this case because the raw variability of species-specific CNDD estimates is also driven by statistical uncertainty.

## Reporting summary

Further information on research design is available in the Nature Portfolio Reporting Summary linked to this article.

## Data availability

The forest data that support the findings of this study are available from the ForestGEO network. For some of the sites, the data are publicly available at https://forestgeo.si.edu/explore-data. Restrictions apply, however, to the availability of the data from other sites, which were used under licence for the current study, and so are not publicly available. Raw data are available upon reasonable request and with permission of the principal investigators of the ForestGEO sites at https://www.forestgeo.si.edu/latitudinal-patterns-stabilizing-density-dependence-forest-communities. Species-site-specific CNDD estimates to reproduce the meta-analyses are available at https://github.com/LisaHuelsmann/latitudinalCNDD.

## Code availability

All custom R code used for the analyses is available at https://zenodo.org/doi/10.5281/zenodo.10646018 and in a GitHub repository at https://github.com/LisaHuelsmann/latitudinalCNDD.

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

**Acknowledgements** We thank the many people involved in establishing and maintaining the forest sites used in the analyses. A detailed list of funding sources, fieldwork permissions, acknowledgements and references for each forest site is available in the Supplementary Notes. L.H. and F.H. received funding from the Bavarian Ministry of Science and the Arts in the context of the Bavarian Climate Research Network (bayklif). L.C. received funding from the US National Science Foundation (DEB-1845403). Contributions by M.d.S.L. were supported by the ForestGEO network (2020), the Smithsonian Institute (2020-2021) and PROEX-CAPES (Coordenação de Aperfeiçoamento de Pessoal de Nível Superior—Brazil, 2022). The study benefited from the ForestGEO workshop in 2019 (NSF-2020424 to S.J.D.).

**Author contributions** L.H., F.H. and R.A.C. conceived the overall study. L.H. and M.d.S.L. homogenized the forest census and meta data. L.H. and F.H., with advice from R.A.C., L.C. and M.D.V. devised the CNDD estimator and the analysis pipeline. L.H. performed the statistical analyses and generated figures and tables. L.H., F.H., R.A.C., L.C. and M.D.V. interpreted the results and drafted the manuscript. The other authors contributed forest census data and feedback on the manuscript. All authors read and approved the manuscript.

**Funding** Open access funding provided by Universität Bayreuth.

**Competing interests** The authors declare no competing interests.

**Additional information**
**Correspondence and requests for materials** should be addressed to Lisa Hülsmann.

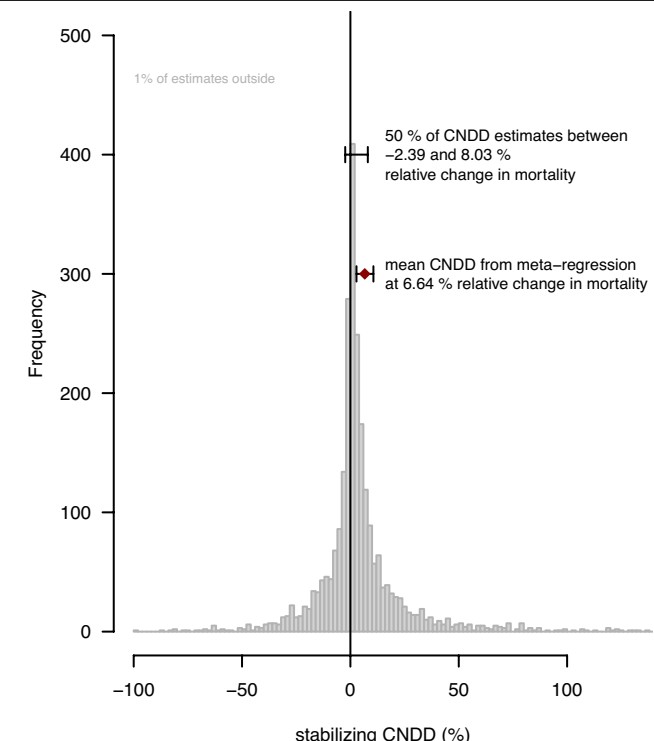

**Extended Data Fig. 1 | Distribution of stabilizing CNDD calculated over species-site-specific interquantile ranges in conspecific density.** Besides the frequency distribution of species-site-specific estimates, the figure indicates the global average assessed through meta-regression with random intercepts for sites and species in sites (red diamond with 95% CI) and the interquantile range of the estimates. Note that 1% of the CNDD estimates are outside the limits of the *x* axis. Stabilizing CNDD is defined as the relative change (in %) in annual mortality probability (relative average marginal effect; rAME) induced by changing conspecific density from the first to the third quantile of observed conspecific densities per species while keeping total densities constant.

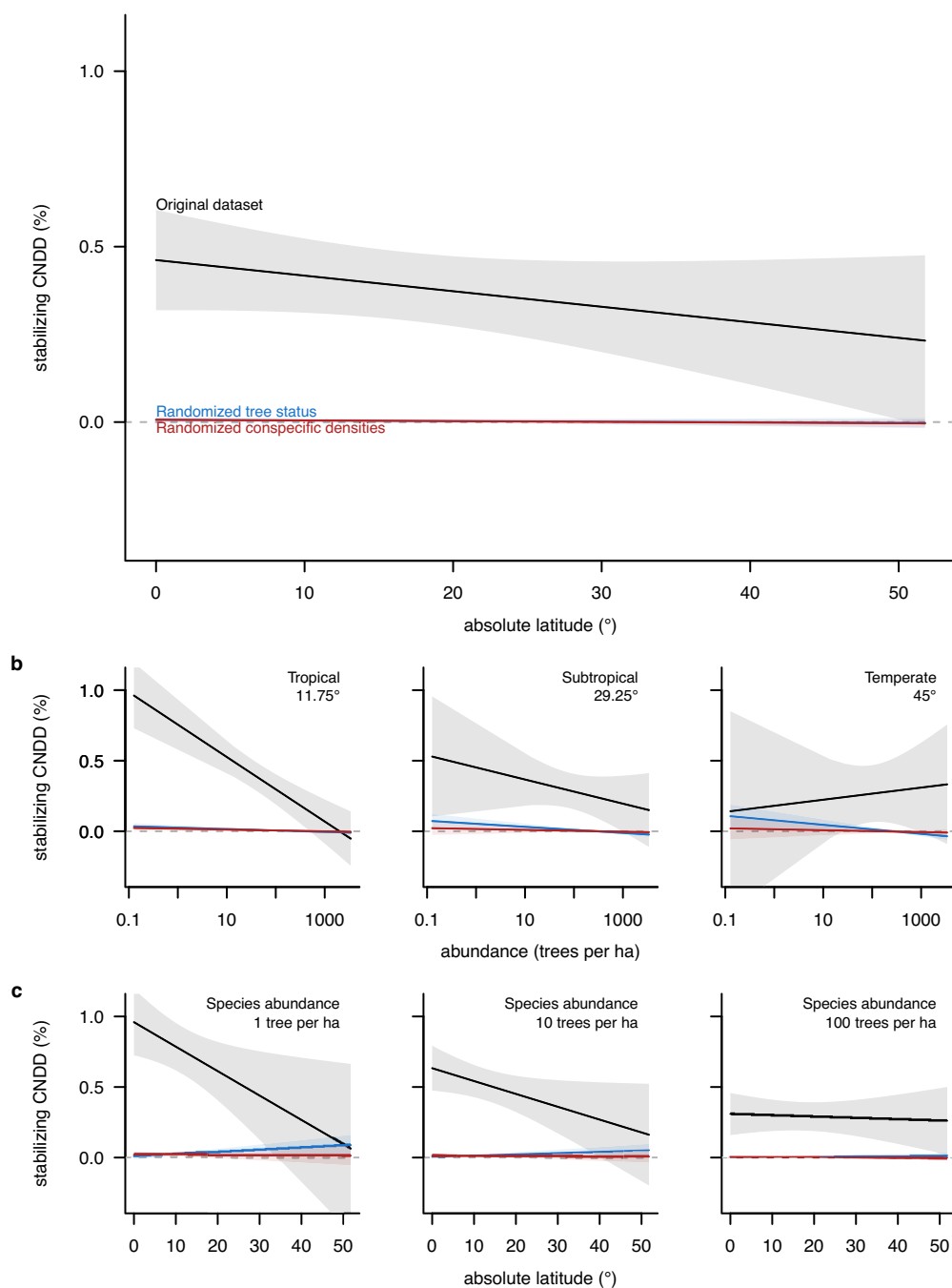

**Extended Data Fig. 2 | Robustness tests of the analysis pipeline based on randomized datasets. a–c**, When observations of tree status (blue) or conspecific density (red) were randomized, stabilizing CNDD was practically zero at all latitudes (**a**) and for all species abundances (**b**,**c**). Rare species exhibited minimally, but significantly, stronger CNDD for the dataset with randomized tree status (blue), but the effect sizes varied by orders of magnitude from those observed in the original dataset (black). See 'Robustness tests' in Methods for details. For details on the visualization and definition of CNDD in **a** and **b**,**c**, see Figs. 2 and 3, respectively. Estimates of the meta-regressions are shown in Extended Data Table 2 (randomized datasets) and Table 1 (original dataset).

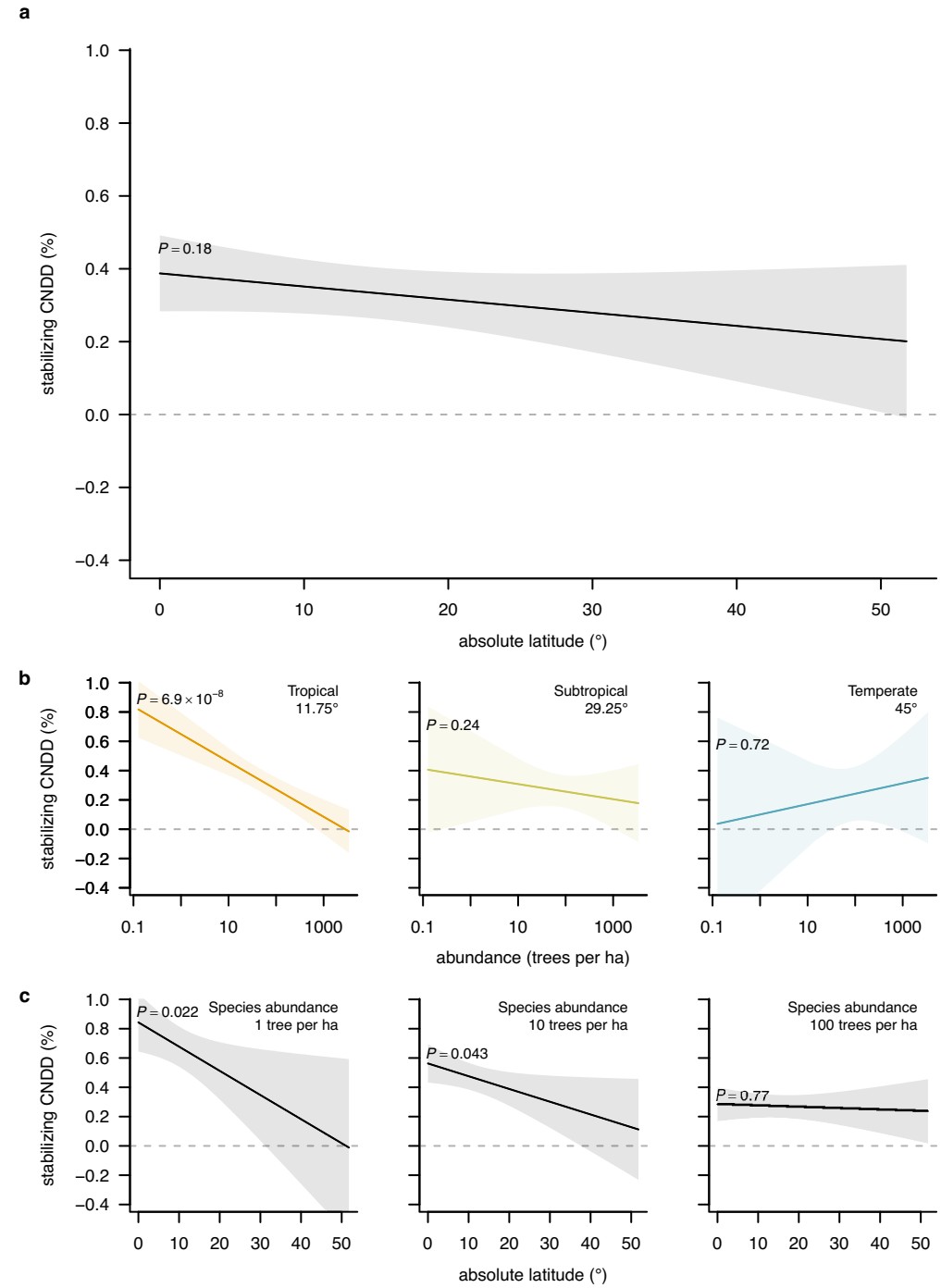

**Extended Data Fig. 3 | Robustness tests without the most influential observations. a–c**, When influential observations were removed ($n_{removed}$ = 99, see 'Robustness tests' in Methods for details), the qualitative patterns remained the same; that is, stronger CNDD for rare than for common species in the tropics (**b**,**c**) but not generally stronger tropical CNDD (**a**). For details on the visualization and definition of CNDD in **a** and **b**,**c**, see Figs. 2 and 3, respectively. Estimates of the meta-regressions are shown in Extended Data Table 2.

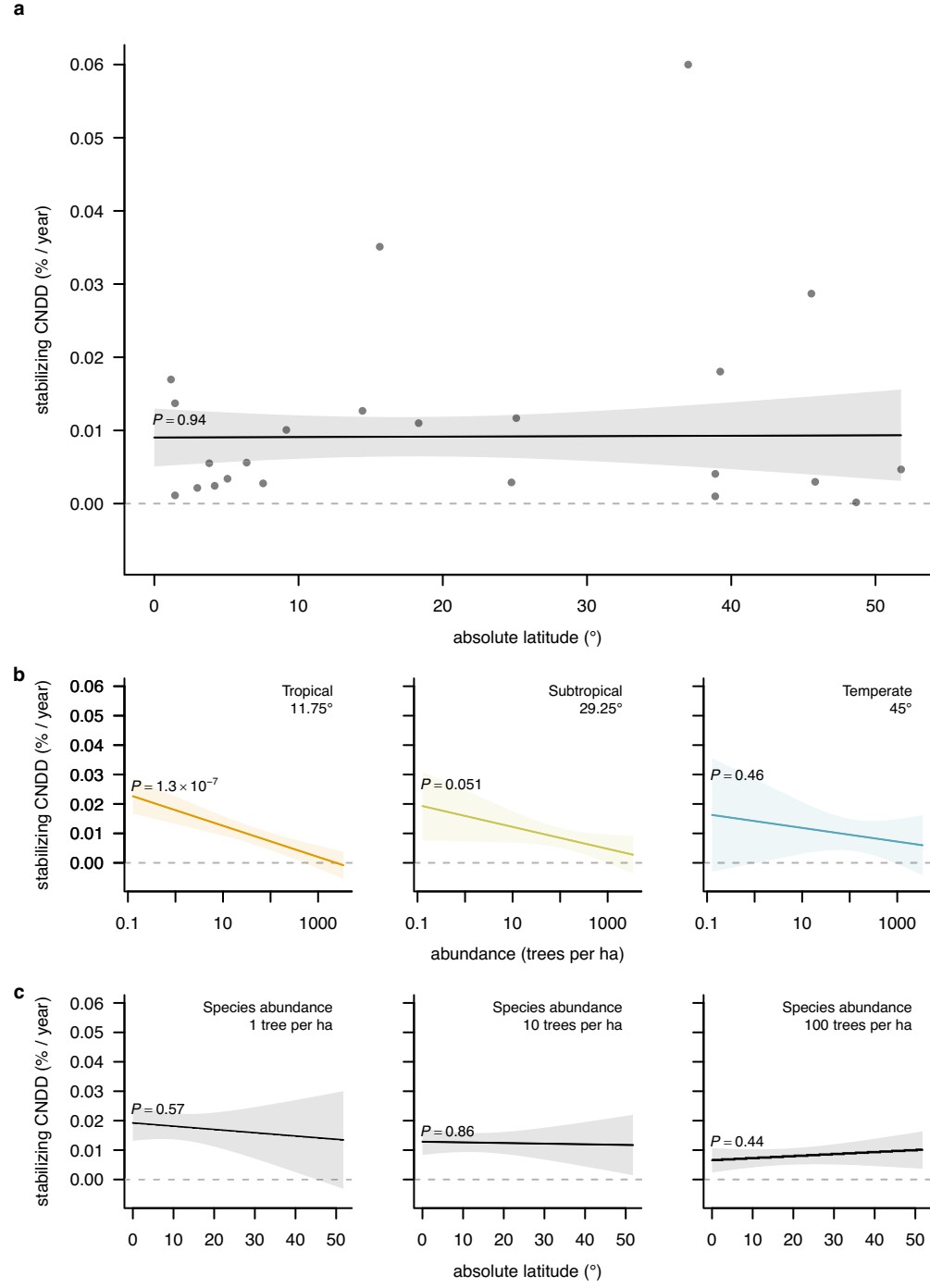

**Extended Data Fig. 4 | Alternative definition of stabilizing CNDD as the absolute change in mortality probability. a–c**, Similar patterns to the main analysis are visible; that is, stronger CNDD for rare than for common species in the tropics (**b**,**c**) but not generally stronger tropical CNDD (**a**), but, in contrast to the main analysis, the interaction of species abundance and latitude was insignificant. See 'Quantification of conspecific density dependence' in Methods for details on the definition of CNDD. For details on the visualization in **a** and **b**,**c**, see Figs. 2 and 3, respectively. Estimates of the meta-regressions are shown in Extended Data Table 3.

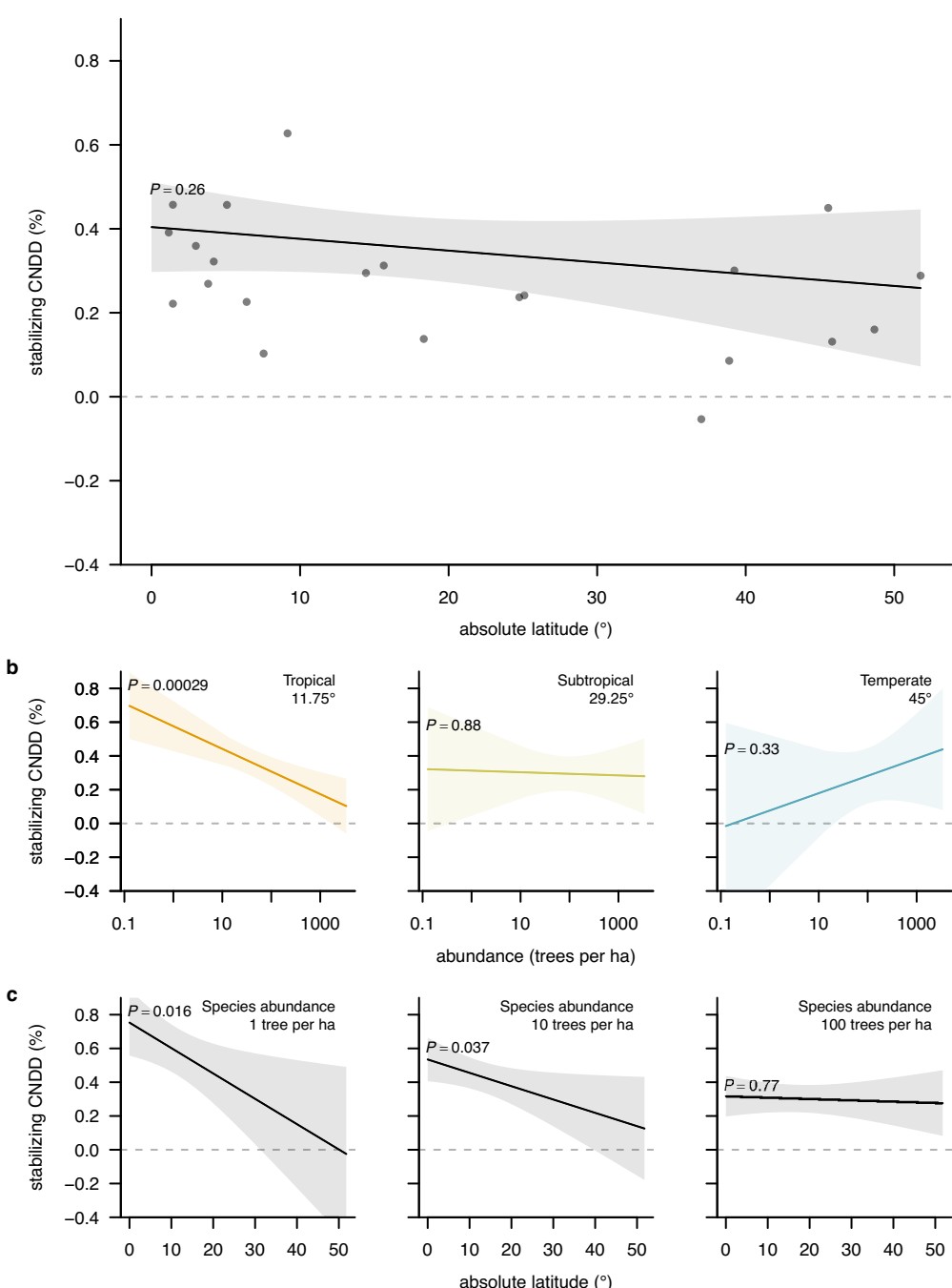

**Extended Data Fig. 5 | Alternative definition of stabilizing CNDD calculated at low conspecific densities (invasion densities).** The patterns remained qualitatively the same as in the main analysis; that is, stronger CNDD for rare than for common species in the tropics (**b**,**c**) but not generally stronger tropical CNDD (**a**). See 'Quantification of conspecific density dependence' for details on the definition of CNDD. For details on the visualization in **a** and **b**,**c**, see Figs. 2 and 3, respectively. Note that for one of the sites (Smithsonian Conservation Biology Institute), no point could be drawn for mean CNDD in **a** because the site-specific meta-regression did not converge. Estimates of the meta-regressions are shown in Extended Data Table 3.

**Extended Data Table 1 | Summary information of the data used in mortality models per forest plot**

| Site | N status observations | N species for mortality analyses | % species fitted individually | N species fitted as rare trees | N species fitted as rare shrubs | % dead status observations |
|---|---|---|---|---|---|---|
| Amacayacu | 70,167 | 1,101 | 14.1 | 479 | 467 | 18.0 |
| Barro Colorado Island | 1,179,556 | 301 | 61.8 | 69 | 46 | 11.2 |
| Edoro - Ituri | 139,804 | 347 | 17.3 | 131 | 156 | 9.8 |
| Fushan | 218,901 | 105 | 57.1 | 21 | 24 | 15.5 |
| Huai Kha Khaeng | 146,993 | 267 | 34.8 | 130 | 44 | 26.4 |
| Ilha do Cardoso | 17,503 | 116 | 50.0 | 31 | 27 | 39.7 |
| Khao Chong | 137,684 | 518 | 29.7 | 223 | 141 | 11.5 |
| Korup | 251,299 | 448 | 35.0 | 163 | 128 | 11.2 |
| La Planada | 64,854 | 196 | 44.9 | 60 | 48 | 26.7 |
| Lambir | 804,582 | 1,375 | 47.1 | 422 | 306 | 9.9 |
| Lenda - Ituri | 108,784 | 300 | 16.7 | 116 | 134 | 10.3 |
| Lilly Dickey Woods | 14,072 | 33 | 36.4 | 11 | 10 | 23.9 |
| Luquillo | 62,153 | 128 | 40.6 | 37 | 39 | 41.7 |
| Mo Singto | 180,400 | 245 | 38.4 | 112 | 39 | 16.3 |
| Pasoh | 1,147,066 | 858 | 54.0 | 213 | 182 | 7.3 |
| Santa Cruz | 5,163 | 26 | 38.5 | 0 | 0 | 24.0 |
| Sinharaja | 273,824 | 221 | 45.7 | 66 | 54 | 9.4 |
| Smithsonian Conservation Biology Institute | 14,336 | 57 | 26.3 | 27 | 15 | 16.5 |
| Smithsonian Environmental Research Center | 13,406 | 58 | 13.8 | 21 | 29 | 14.4 |
| Wabikon | 47,548 | 32 | 53.1 | 6 | 9 | 18.5 |
| Wind River | 14,992 | 22 | 31.8 | 0 | 10 | 9.5 |
| Wytham Woods | 22,063 | 17 | 29.4 | 8 | 0 | 9.1 |
| Zofin | 39,401 | 11 | 18.2 | 0 | 0 | 0.8 |

Observations for the mortality analyses (*N* status observations) were selected as follows: (1) no fern or palm species, (2) no missing information on coordinates, species, status, or date of measurement, (3) alive in the first census and alive or dead in the consecutive census, (4) DBH between 1 and 10 cm in the first census, (5) more than 30 m away from the plot boundaries. From the total number of species in the mortality dataset (*N* species for mortality analyses), only some proportion could be successfully fit (% species fitted individually). The remaining species were jointly fitted in species groups (*N* species fitted as rare trees or shrubs): these were species with fewer than 20 alive and dead observations each, species with fewer than four unique values of conspecific density, species with a range of conspecific density values not including the value used to calculate average marginal effects, or species for which no convergence of the mortality model was achieved. In some cases, also the mortality model for a species group did not converge (indicated by *N* = 0 in the respective column). Note that the percentage of dead trees (% dead status observations) does not correspond to mortality rates because of varying interval lengths. Numbers of species can include morphospecies. Note that for the Pasoh site, each stem was counted as an individual tree (see 'Forest data' in Methods).

**Extended Data Table 2 | Estimates for the two main meta-regressions using randomized and reduced datasets**

| Dataset | Model | Characteristic | Beta | 95% CI | *P* value |
|---|---|---|---|---|---|
| **Randomized tree status**<br><br>n = 2534 species or species groups from 23 forest sites | **a) Average species CNDD**<br>$\sigma_r = 0.0000712$<br>$\sigma_s = 0.0000005$ | intercept | 0.0000434 | -0.0000210, 0.0001078 | 0.19 |
| | | tLatitude | -0.0000017 | -0.0000054, 0.0000020 | 0.38 |
| | **b) Abundance-mediated CNDD**<br>$\sigma_r = 0$<br>$\sigma_s = 0$ | intercept | 0.0002565 | 0.0000986, 0.0004144 | **0.0015** |
| | | tLatitude | 0.0000157 | -0.0000012, 0.0000325 | 0.069 |
| | | tAbundance | -0.0000435 | -0.0000744, -0.0000127 | **0.0056** |
| | | tLatitude:tAbundance | -0.0000029 | -0.0000058, 0.0000000 | 0.052 |
| **Randomized conspecific density**<br><br>n = 2533 species or species groups from 23 forest sites | **a) Average species CNDD**<br>$\sigma_r = 0.00000025$<br>$\sigma_s = 0.00001387$ | intercept | 0.000044 | -0.000001, 0.000088 | 0.056 |
| | | tLatitude | -0.000002 | -0.000005, 0.000001 | 0.16 |
| | **b) Abundance-mediated CNDD**<br>$\sigma_r = 0.00000000$<br>$\sigma_s = 0.00000021$ | intercept | 0.000181182 | 0.000026150, 0.000336214 | **0.022** |
| | | tLatitude | -0.000001172 | -0.000017144, 0.000014799 | 0.89 |
| | | tAbundance | -0.000026545 | -0.000054760, 0.000001671 | 0.065 |
| | | tLatitude:tAbundance | -0.000000065 | -0.000002732, 0.000002602 | 0.96 |
| **Reduced dataset**<br><br>n = 2435 species or species groups from 22 forest sites | **a) Average species CNDD**<br>$\sigma_r = 0.0013$<br>$\sigma_s = 0.0027$ | intercept | 0.003445 | 0.002729, 0.004161 | **$4.1 \times 10^{-21}$** |
| | | tLatitude | -0.000036 | -0.000089, 0.000017 | 0.18 |
| | **b) Abundance-mediated CNDD**<br>$\sigma_r = 0.0014$<br>$\sigma_s = 0.0027$ | intercept | 0.006465 | 0.005085, 0.007846 | **$4.3 \times 10^{-20}$** |
| | | tLatitude | -0.000164 | -0.000305, -0.000023 | **0.022** |
| | | tAbundance | -0.000813 | -0.001108, -0.000518 | **$6.9 \times 10^{-8}$** |
| | | tLatitude:tAbundance | 0.000034 | 0.000004, 0.000064 | **0.027** |

We randomized observations of tree status within each species, thus removing any relationship between mortality and predictors but retaining species-level mortality rates, and observations of local conspecific density within each species, thus removing the relationship between mortality and conspecific density but retaining the relationships between mortality and confounders (see 'Robustness tests' in Methods). For the reduced dataset, we removed *n* = 99 influential species-site-specific CNDD estimates with Cook's distances larger than 0.005 to evaluate the possibility that a few observations were responsible for the observed patterns. Species-site-specific CNDD estimates and predictors are defined as in Table 1. Predictions of the meta-regressions are shown in Extended Data Figs. 2 and 3.

**Extended Data Table 3 | Estimates for the two main meta-regressions using two alternative definitions of stabilizing CNDD**

| CNDD definition | Model | Characteristic | Beta | 95% CI | *P* value |
|---|---|---|---|---|---|
| **Absolute marginal effect (aAME)** | **a) Average species CNDD**<br>$\sigma_r = 0.000055$<br>$\sigma_s = 0.000106$ | intercept | 0.00009104 | 0.00006248, 0.00011960 | **4.2 × 10⁻¹⁰** |
| | | tLatitude | 0.00000006 | -0.00000159, 0.00000171 | 0.94 |
| | **b) Abundance-mediated CNDD**<br>$\sigma_r = 0.000057$<br>$\sigma_s = 0.000105$ | intercept | 0.00017884 | 0.00013425, 0.00022343 | **3.8 × 10⁻¹⁵** |
| | | tLatitude | -0.00000111 | -0.00000499, 0.00000276 | 0.57 |
| | | tAbundance | -0.00002300 | -0.00003153, -0.00001447 | **1.3 × 10⁻⁰⁷** |
| | | tLatitude:tAbundance | 0.00000039 | -0.00000034, 0.00000111 | 0.29 |
| **Relative marginal effect (rAME) at low conspecific densities** | **a) Average species CNDD**<br>$\sigma_r = 0.0013$<br>$\sigma_s = 0.0035$ | intercept | 0.003704 | 0.002946, 0.004463 | **1.0 × 10⁻²¹** |
| | | tLatitude | -0.000028 | -0.000076, 0.000020 | 0.26 |
| | **b) Abundance-mediated CNDD**<br>$\sigma_r = 0.0014$<br>$\sigma_s = 0.0036$ | intercept | 0.005743 | 0.004355, 0.007130 | **5.0 × 10⁻¹⁶** |
| | | tLatitude | -0.000150 | -0.000272, -0.000028 | **0.016** |
| | | tAbundance | -0.000580 | -0.000893, -0.000266 | **0.00029** |
| | | tLatitude:tAbundance | 0.000031 | 0.000006, 0.000056 | **0.015** |

Species-site-specific CNDD estimates (*n* = 2,534 species or species groups from 23 forest sites) were calculated as the absolute change in mortality probability (aAME) and as the relative change in mortality probability (rAME) but at low conspecific densities (invasion densities; see 'Quantification of conspecific density dependence' in Methods). For the meta-regressions, aAMEs were not transformed and can be simply multiplied by 100 to obtain the absolute change in annual mortality probability induced by additional conspecific neighbour in per cent. For rAMEs, back-transformation is necessary as in Table 1. Predictions of the meta-regressions are shown in Extended Data Figs. 4 and 5.

**Extended Data Table 4 | Estimates for the meta-regression testing the second hypothesized latitudinal pattern in stabilizing CNDD additionally accounting for potential confounding by life history strategies**

| Model | Characteristic | Beta | 95% CI | *P* value |
|---|---|---|---|---|
| **Demographic rates model** $\sigma_r = 0.0018$ $\sigma_s = 0.0053$ | intercept | 0.007549 | 0.005799, 0.009299 | **$2.8 \times 10^{-17}$** |
| | tLatitude | -0.000146 | -0.000289, -0.000002 | **0.046** |
| | tAbundance | -0.000966 | -0.001357, -0.000576 | **$1.2 \times 10^{-6}$** |
| | growth | -0.000405 | -0.000952, 0.000142 | 0.15 |
| | survival | -0.000337 | -0.000910, 0.000237 | 0.25 |
| | tLatitude:tAbundance | 0.000032 | 0.000003, 0.000061 | **0.033** |
| | growth:survival | -0.000321 | -0.000635, -0.000008 | **0.045** |
| **Demographic tradeoffs model** $\sigma_r = 0.0018$ $\sigma_s = 0.0053$ | intercept | 0.008219 | 0.006369, 0.010069 | **$3.1 \times 10^{-18}$** |
| | tLatitude | -0.000171 | -0.000315, -0.000028 | **0.019** |
| | tAbundance | -0.001041 | -0.001442, -0.000640 | **$3.5 \times 10^{-7}$** |
| | tradeoff1 | 0.000497 | -0.000058, 0.001052 | 0.079 |
| | tradeoff2 | -0.000778 | -0.001295, -0.000262 | **0.0031** |
| | tLatitude:tAbundance | 0.000036 | 0.000007, 0.000065 | **0.016** |
| | tradeoff1:tradeoff2 | -0.000151 | -0.000493, 0.000190 | 0.38 |

The original 'abundance-mediated CNDD' model (see Table 1b) was extended to include either the demographic rates growth and mortality or demographic trade-offs (see 'Robustness tests' in Methods). Demographic rates and trade-off axes were centred and scaled. Species-site-specific CNDD estimates (*n*=2,534 species or species groups from 23 forest sites) and predictors (latitude and abundance) are defined as in Table 1.

# Reporting Summary

## Statistics

For all statistical analyses, confirm that the following items are present in the figure legend, table legend, main text, or Methods section.

| n/a | Confirmed | |
|---|---|---|
| ☐ | ☒ | The exact sample size (*n*) for each experimental group/condition, given as a discrete number and unit of measurement |
| ☐ | ☒ | A statement on whether measurements were taken from distinct samples or whether the same sample was measured repeatedly |
| ☐ | ☒ | The statistical test(s) used AND whether they are one- or two-sided *Only common tests should be described solely by name; describe more complex techniques in the Methods section.* |
| ☐ | ☒ | A description of all covariates tested |
| ☐ | ☒ | A description of any assumptions or corrections, such as tests of normality and adjustment for multiple comparisons |
| ☐ | ☒ | A full description of the statistical parameters including central tendency (e.g. means) or other basic estimates (e.g. regression coefficient) AND variation (e.g. standard deviation) or associated estimates of uncertainty (e.g. confidence intervals) |
| ☐ | ☒ | For null hypothesis testing, the test statistic (e.g. $F$, $t$, $r$) with confidence intervals, effect sizes, degrees of freedom and $P$ value noted *Give P values as exact values whenever suitable.* |
| ☒ | ☐ | For Bayesian analysis, information on the choice of priors and Markov chain Monte Carlo settings |
| ☐ | ☒ | For hierarchical and complex designs, identification of the appropriate level for tests and full reporting of outcomes |
| ☐ | ☒ | Estimates of effect sizes (e.g. Cohen's *d*, Pearson's *r*), indicating how they were calculated |

*Our web collection on statistics for biologists contains articles on many of the points above.*

## Software and code

Policy information about availability of computer code

| Data collection | No software was used for data collection. |
|---|---|
| Data analysis | The analyses were conducted in R version 4.2.1 using the packages mgcv, metafor, and DHARMa. Additional packages were used for data management and visualization (see code at https://github.com/LisaHuelsmann/latitudinalCNDD). |

For manuscripts utilizing custom algorithms or software that are central to the research but not yet described in published literature, software must be made available to editors and reviewers. We strongly encourage code deposition in a community repository (e.g. GitHub). See the Nature Portfolio guidelines for submitting code & software for further information.

## Data

Policy information about availability of data

All manuscripts must include a data availability statement. This statement should provide the following information, where applicable:
- Accession codes, unique identifiers, or web links for publicly available datasets
- A description of any restrictions on data availability
- For clinical datasets or third party data, please ensure that the statement adheres to our policy

The forest data that support the findings of this study are available from the ForestGEO network. For some of the sites, the data is publicly available at https://forestgeo.si.edu/explore-data. Restrictions apply, however, to the availability of the data from other sites, which were used under license for the current study, and

so are not publicly available. Raw data are available from the authors upon reasonable request and with permission of the principal investigators of the ForestGEO sites. Species-site-specific CNDD estimates to reproduce the meta-analyses are available at https://github.com/LisaHuelsmann/latitudinalCNDD.

## Human research participants

Policy information about studies involving human research participants and Sex and Gender in Research.

| | |
|---|---|
| Reporting on sex and gender | does not apply - I think this part should have been hidden |
| Population characteristics | *Describe the covariate-relevant population characteristics of the human research participants (e.g. age, genotypic information, past and current diagnosis and treatment categories). If you filled out the behavioural & social sciences study design questions and have nothing to add here, write "See above."* |
| Recruitment | *Describe how participants were recruited. Outline any potential self-selection bias or other biases that may be present and how these are likely to impact results.* |
| Ethics oversight | *Identify the organization(s) that approved the study protocol.* |

Note that full information on the approval of the study protocol must also be provided in the manuscript.

# Field-specific reporting

Please select the one below that is the best fit for your research. If you are not sure, read the appropriate sections before making your selection.

☐ Life sciences   ☐ Behavioural & social sciences   ☒ Ecological, evolutionary & environmental sciences

For a reference copy of the document with all sections, see nature.com/documents/nr-reporting-summary-flat.pdf

# Ecological, evolutionary & environmental sciences study design

All studies must disclose on these points even when the disclosure is negative.

| | |
|---|---|
| Study description | We used repeated census data from twenty-three large forest sites around the globe to analyze latitudinal patterns in conspecific negative density dependence (CNDD) following a three-step approach: We fitted species-site-specific mortality models from repeated observations of individual trees using neighborhood densities as a predictor. Then, we used these models to quantify CNDD for each species and site using an estimator designed to maximize robustness, comparability, and relevance for fitness and stabilization. Finally, we used meta-regressions to explore latitudinal patterns in CNDD. Robustness of the analysis pipeline was validated by model diagnostics and randomization. |
| Research sample | The data used in this study were collected at twenty-three forest sites with permanent forest dynamics plots that are part of the Forest Global Earth Observatory network (ForestGEO). |
| Sampling strategy | All free-standing woody stems with diameter ≥1 cm at 1.3 m from the ground (DBH) are censused at each site, with between 9,718 and 495,577 mapped tree individuals at each site. For the mortality analyses, we selected observations of all alive trees of non-fern and non-palm species with a DBH < 10 cm in one census and follow-up data in a consecutive census. |
| Data collection | The census data collected for each individual stem or tree include species, DBH, spatial coordinates and status (alive or dead). |
| Timing and spatial scale | The plots vary in size between 6 and 52 ha. At all sites included in this study, two or more censuses have been carried out with remeasurement intervals of approximately five years. Censuses were carried out between 1981 and 2021. |
| Data exclusions | Mortality analyses were restricted to small trees (between 1 and 10 cm DBH) because it is assumed that CNDD effects are most pronounced in early life stages. Observations of trees or stems were excluded when information on coordinates, species, status, or date of measurement was missing. |
| Reproducibility | This study is based on observational data, and no experiments were conducted. |
| Randomization | This study is based on observational data. Therefore, randomization into groups does not apply. |
| Blinding | Blinding is not relevant for this study because the data were not specifically collected to assess density dependence. |

Did the study involve field work?   ☐ Yes   ☒ No

# Reporting for specific materials, systems and methods

We require information from authors about some types of materials, experimental systems and methods used in many studies. Here, indicate whether each material, system or method listed is relevant to your study. If you are not sure if a list item applies to your research, read the appropriate section before selecting a response.

| Materials & experimental systems | | Methods | |
|---|---|---|---|
| n/a | Involved in the study | n/a | Involved in the study |
| ☒ | Antibodies | ☒ | ChIP-seq |
| ☒ | Eukaryotic cell lines | ☒ | Flow cytometry |
| ☒ | Palaeontology and archaeology | ☒ | MRI-based neuroimaging |
| ☒ | Animals and other organisms | | |
| ☒ | Clinical data | | |
| ☒ | Dual use research of concern | | |

