## [Peer Review File · Nature]

Manuscript Title: Latitudinal patterns in stabilizing density dependence of forest communities

Editorial Notes:

Reviewer Comments & Author Rebuttals

Reviewer Reports on the Initial Version:

Referees' comments:

Referee #1 (Remarks to the Author):

The manuscript addresses questions that have long been puzzling ecologists: What mechanisms allow so many tree species to coexist in species-rich forests, and does the role of these vary across latitudes? The authors report the results of a study in which the likelihood of mortality of individual trees was investigated in relation to the density of conspecific and heterospecific tree individuals in the neighbourhood. Using a large data set compiled from 23 forest dynamics plots spanning a latitudinal gradient of more than 50 degrees, they explore the potential role of conspecific negative density-dependence (CNDD) in promoting species coexistence in forest tree communities across latitudes. This is the first attempt to assess latitudinal patterns in CNDD using tree mortality data rather than static tree distribution data. By doing this, the study avoids the statistical biases that some previous studies have been criticised for.

Three main results are presented and discussed: 1) Contrary to predictions, the average CNDD does not vary across latitudes (and is relatively weak overall); 2) Abundant species experience stronger CNDD at tropical sites but not at temperate sites, and 3) The degree of interspecific variation in CNDD tends to be high within tree communities. Taken together, these results advance our understanding of the ways in which CNDD may contribute to observed latitudinal patterns of forest tree diversity: The first result is interesting and important since it is in contrast with that from a study by La Manna et. al. (2017; Science) which was subsequently criticised due to potential statistical biases. The second result provides more conclusive evidence for CNDD being stronger in rare than in common species in tropical forests (a pattern which has been reported in previous studies which were subsequently criticised for statistical biases). The third result implies that the stabilising role of CNDD may be less pronounced in forest systems than what has often been assumed.

The study appears methodologically sound, and the analyses expertly done. The analytical pipeline involves several decisions which could potentially influence the results, but methodological choices are backed up by references to relevant literature and randomisation studies have been conducted to evaluate the robustness of the results. The authors have also explored the role of some potential confounding factors that could drive some of the reported relationships (e.g. between rarity and CNDD). Altogether, the conclusions seem robust and well supported by the data.

The presentation is clear and logically structured. I have identified a few points that might benefit from some more information and/or rewording:

-The way the Janzen-Connell hypothesis is introduced and cited (e.g. L8-9; L65; L77) might be slightly misleading for readers who are not familiar with the original papers by Janzen and Connell. Although both Janzen and Connell mention the possibility that the role of natural enemies in controlling the abundances of individual species and thereby preventing competitive exclusion is likely to vary across latitudes this is not dealt with in huge detail in the original publications (which focus primarily on describing a potential mechanism that could explain the enigma of high species richness in tropical forests). Some of the current wording might be interpreted as if the Janzen-Connell hypothesis *is* the specific idea that CNDD will vary in strength across latitudes. This could perhaps be clarified for the broader readership who might not have read the original papers.

- One of the main take home messages is that the results from the analyses presented in the manuscript do not support hypothesised predictions inspired by ideas in Janzen's and Connell's classical papers. As the authors point out (L199-L201), these results may or may not hold if similar analyses were to be conducted for other life stages. This point is important and could perhaps be reiterated in a more prominent place in the manuscript. The reported weak levels of CNDD might not be too surprising given that the analyses focused on the mortality of individuals in size classes that may already be way past the stage at which mortality caused by host-specific enemies might be most important (e.g. early seedling stage).

- The study uses an impressive data set collected through a network of study sites. The map on the ForestGEO website (<https://forestgeo.si.edu/>) suggests that many sites have not been included. There are probably good reasons for selecting the current 23 sites, but it would be good to know why the full network of sites was not utilised and these particular 23 sites chosen for the analyses.

I also came across several more specific points in the text that would benefit from clarification or rewriting, and some details that could be clarified in the figures and tables (listed below with associated line numbers):

- L12 'latitudinal patterns in species-specific CNDD': This wording could possibly be misinterpreted as if you examined how CNDD of individual species occurring at multiple sites varied across latitude?

- L36: Is the word 'fiercely' needed here? As far as I remember, the two papers cited are not particularly fierce in their contents.

- L43: differs in -> differs between?

- L46: could perhaps indicate what you mean by 'large' in this context?

- L46: is the word 'globally' needed here? (Yes, sites are distributed across several continents, but still many parts of the globe that are not covered.)

- L54-55: Based on information provided up to this point in the text, it is not clear what the relative change in mortality probability of trees induced by an additional conspecific neighbour means. I would suggest adding some detail (conspecific individual of dbh 2cm located 1 m from the tree?) or refer to

methods section. Same applies to L70 and L72.

- L119-120: This sentence comes across as a bit repetitive (similar information given in the previous paragraph)
- L194: abundance -> abundances?
- L199: '(...) not only mortality during early life stages' – this phrasing might imply that the current study deals with the earliest life stages of trees which isn't necessarily true. Many individuals might have died before reaching the 1cm DBH threshold of the ForestGEO plot data, and some species might reach maturity at smaller sizes than 10cm DBH.
- Table 1: Is the 'Dataset' column needed given that there is only one entry (raises the question of what other data sets than the 'original' data set there are)? If included, could specify that n (presumably) refers to number of species rather than anything else?
- Table 1: Not immediately clear what the 't's in front of the predictors mean
- L301 (and elsewhere): Is 'absolute latitude' is the correct term here? I believe that you used the distance (degrees) from the equator to test for latitudinal effects, but did not distinguish between sites on the Northern versus Southern hemisphere? Googling 'absolute latitude' did not yield any useful hits. Please confirm that the term is more widely established and used.
- L366: log transformed trees -> log-transformed number of trees
- Fig. 1: This figure is informative, but there is quite a lot going on in terms of colour coding, symbol sizes etc which makes me doubt whether all this information be visible in the printed version. I was struggling to pick everything up (e.g. circles vs diamonds) even when zooming in heavily on my screen.
- L388-390 (& L419-420): "Locations of forest sites and CNDD-abundance relationships are colored by latitude (tropical forests in reddish colors, temperate in bluish)." What about orange/yellow colours? Do the subtle differences in e.g. red colours have a meaning? If the idea is to allow cross-reference to data points in other figures (e.g. 3) colour contrasts appear too small.
- Fig. 2: For maximum clarity, you could perhaps explain what the horizontal lines at $y = 0$ depict? Likewise, in Fig. 3a it might be helpful to more explicitly mention why there is a dotted line at $y = 0.4$ (not all readers might dig into the specifics of the methods sections). (Same comments are also relevant in the context of matching Extended Data figures.)
- L418-419: 'fitted individually per forest site' – would 'fitted separately for each forest site' be clearer?
- L437: Add citation to R
- L442: Are stems shorter than 1.3m not recorded, regardless of their thickness?
- L452: the meaning on 'potential confounders of this relationship' is not immediately clear.
- L453-454: As hinted above, for some species, trees of this size might not necessarily represent 'early life stages'.
- L464: Not crystal clear what 'resurrected trees' mean in this case.
- L496-497: I can see that species-specific optimisation of mu-values might not be feasible, but similar mu values across species may also be unrealistic if species are (following predictions from the Janzen-Connell hypothesis) attacked by different sets of host specific enemies that potentially operate at different spatial scales. Could this averaging across species have had any effects on your overall results?
- L510: tradition -> traditional
- L530-L531: 'at least 20 alive and dead status observations each' – Just to check, does this mean at least 20 alive status observations AND at least 20 dead status observations? You could perhaps also mention

what proportion of species did not fulfil this criterion (and were therefore lumped into one of the rare species categories)?

- L550: remove commas around 'relative and absolute change' for easier reading.

- L561: If I understand correctly, the DBH = 2cm and 1m distance scenario is an arbitrary choice made by the authors. A few words justifying the use of these specific values (rather than some other size and/or distance) might be helpful.

- L580: could perhaps change 'pathogens' to the more general word 'pest' since the specific enemies contributing to conspecific density-dependent mortality are typically not known.

- Extended Data figure 5: A bit more explanation on what is shown would be helpful. I assume that dots are individual species; what do the blue 'clouds' depict? Explain curves and confidence intervals.

- Extended Data figure 6: Explain colour scheme.

- Extended Data Table 2: A bit more detail would be helpful to interpret the numbers in columns 'N trees' and 'N mortality observations'. Is 'N trees' the overall number of tree individuals with a unique tag that were included in the study (some of them across several census periods)? Does 'N mortality observations' show the total number of status recordings (dead or alive) conducted across all species and census periods? If so, I wonder whether the column heading could be changed to 'N status observations' for increased clarity? On that note, it might be helpful to somewhere in the text or in a table give an indication of how common tree mortality is in the studied size classes of trees. This will obviously vary a lot among species and sites, but might still be helpful background information to readers who are not very familiar with tropical forest ecology.

Referee #2 (Remarks to the Author):

Review of Hülsmann et al. Nature, MS 2022-10-17122:

Hülsmann et al. present a study of latitudinal patterns in conspecific negative density dependence (CNDD), which they also call stabilizing CNDD. The authors use data from multiple censuses at 23 ForestGEO plots (large globally-distributed forest inventory plots with all stems >1 cm DBH mapped and measured every ~5 years) to calculate CNDD for each species at each site. They calculate CNDD as the change in annual mortality probability associated with a standardized increase in conspecific density while holding total density constant. Latitudinal patterns across species and sites are then analyzed with meta-regressions that account for hierarchical variability among sites and species as well as within species. The authors report several key findings. First, the authors report no change in mean species CNDD across latitudes. This finding appears to contradict earlier findings of a latitudinal gradient in mean species CNDD¹. Second, the authors report that CNDD varies with species abundance among tropical species (stronger stabilizing CNDD for rare tropical than common tropical species) but not among temperate species. They note that this finding is similar to that reported in a previous study¹. Third, the authors note similar variability in CNDD among species within each site (greater variability in the temperate zone may have indicated greater fitness differences and therefore weaker stabilization). The authors conclude that CNDD may contribute to the latitudinal species diversity gradient, but in more nuanced ways than previously thought – CNDD has greater effectiveness at controlling species abundances in tropical latitudes than at temperate latitudes “despite CNDD being present at comparable levels at all latitudes” (lines 178-179).

The large research team seems well qualified to conduct these analyses, and these ForestGEO data are quite impressive. However, I have several concerns with the analyses and, more critically, major concerns with the presentation and interpretation of findings from these analyses. My primary concerns are that some key findings are not currently presented properly or discussed in the manuscript, so I detail those primary concerns first. I follow up with several other concerns for the authors to consider about the presentation of their findings. Finally, I offer a few concerns about the analyses themselves. While I suspect that these analytical concerns may not substantially affect the key findings, I encourage the authors to consider and verify this.

Primary concern about presentation and interpretation of findings

As mentioned above, the authors report no latitudinal trend in average species CNDD. They base this assertion on meta-regressions of species-by-site CNDD estimates across the latitudinal gradient. I have no issues with the meta-regressions as carried out – they seem quite sound. However, I do have a major concern with the presentation and interpretation of these meta-regressions. Two primary explanatory variables were used in the meta-regressions: latitude and species abundance. Both have previously been found to have potential influences on CNDD¹⁻³. Moreover, one previous study (as mentioned by the authors) found a significant interaction between the two¹. Indeed, the present manuscript reports a similar significant interaction between latitude and species abundance. However, when reporting no latitudinal trend in CNDD, the authors use a different meta-regression model than the model that shows a significant interaction between latitude and species abundance that only includes latitude (i.e., not species abundance).

This presentation is problematic for several reasons and, importantly, obscures some of the authors' most important findings – significant relationships between CNDD and latitude for rare species as well as for species at mean and median abundances (see Fig. R1 below). Figure R1 was produced by me from the same data, model, R code, and even plotting functions that were provided by the authors. In fact, using the code and data provided by the authors, I was able to determine that a significant latitudinal gradient in CNDD is evident for 57% of species in this study (i.e., all species with abundances lower than 12 trees / ha; mean log(abundance) across species = 10.2 trees / ha).

Figure R1. Same as Fig. 2 in manuscript, except this figure (unlike Fig. 2) uses the same meta-regression model to produce relationships in panels A and B (a meta-regression with latitude, species abundance, and their significant interaction as fixed effects). (A) A significant relationship between CNDD and latitude was found for species at rare and intermediate (mean) abundances but not for common species. (B) Same as Fig. 2b in manuscript. This figure was produced with the same meta-regression models, data, code, and plotting functions as provided by the authors. Fig. 2 in the manuscript uses two different meta-regression models for panels A and B. Fig. 2a uses a meta-regression model with latitude only, and Fig. 2b uses a meta-regression model with latitude, species abundance, and their significant interaction.

Such latitudinal gradients in CNDD for species at lower abundances have been predicted in previous literature to contribute to the latitudinal gradient in species diversity. Specifically, if common species experience some degree of stabilization via CNDD across the latitudinal gradient (right panel in Fig. R1a), then stronger CNDD for rare and intermediately-abundant tropical than temperate species (left and center panels in Fig. R1a) should stabilize rare species more strongly in tropical forests relative to temperate forests. This would be expected to lead to the persistence of more rare species and even more intermediately-abundant species in tropical than in temperate forests, contributing to lower extinction rates in tropical than in temperate forests and potentially explaining why tropical forests are sometimes thought to be “museums of diversity”^{1,4,5}.

The need to interpret two main effects in light of a significant interaction: Main effects that are involved in a significant interaction with another main effect should be interpreted at the different levels of each main effect. This is what is done for species abundance in Fig. 2b in the manuscript. The relationship between CNDD and species abundance changes across different latitudes, going from a positive relationship in tropical latitudes to no relationship in temperate latitudes (hence the three panels for Fig. 2b in the manuscript). In the same way, the significant interaction between latitude and species abundance also means that the relationship between CNDD and latitude changes with species abundance (Fig. R1a). There is a significant latitudinal gradient in CNDD for rare species, still a latitudinal gradient in CNDD (but weaker) for species at mean abundances, and the latitudinal gradient in CNDD is not present for common species. This significant interaction between latitude and species abundance was bolded in Table 1 in the manuscript, and the significant relationship between CNDD and latitude for rare species (defined by the authors as a species with 1 tree / ha) was also bolded in Table 1 ($p = 0.02$; this is the same relationship as shown in the upper left panel of Fig. R1).

The authors were kind enough to provide the species-by-site CNDD estimates used in their meta-regressions for my evaluation. This is how I was able to produce Fig. R1 – code was already written in their scripts to produce Fig. R1 as seen here with all 6 panels – all I had to do was slightly modify their code to use the same meta-regression model to produce both panels A and B in Fig. 2. Instead of presenting both main effects in light of the level of the other (as shown here in Fig. R1 and written in the authors' code as such), the authors appeared to adjust their scripts so that panel A shows the results from a different meta-regression model that includes only latitude, while panel B shows results from the model with both latitude and species abundances (and their significant interaction).

What explains this seemingly odd presentation and interpretation of a significant interaction? It is likely that the authors were interested in comparing mean CNDD across latitudes without incorporating information about species abundances for some reason. However, this is not appropriate statistically or conceptually. First, it is clear that CNDD varies with species abundance, and not incorporating this information when calculating mean CNDD in each forest plot adds extra error or “noise” to the calculation of each plot's mean CNDD. Second, mean species abundance changes across plots, going from lower values at tropical plots to higher values at temperate plots. This means that the meta-regression model that does not incorporate known variation in CNDD with species abundances (presented in Fig. 2a) compares CNDD of rarer tropical species to more common temperate species. Note that CNDD is actually greater for common than for rare species at temperate latitudes (Fig. 2b in the manuscript and Fig. R1a & R1b above). **Thus, by comparing mean CNDD across latitudes in a model without species abundance, CNDD is compared at different species abundances.** It is statistically and conceptually best to compare CNDD across latitudes for species at the same abundances. One main reason for this is that species with lower abundances generally experience fewer numbers of nearby conspecifics (they are rare) whereas species with greater abundances tend to experience greater numbers of nearby conspecifics (they are nearly everywhere across the plot). In other words, rare species can often (although not necessarily always) encounter a different range of conspecific densities than common species or intermediately-abundant species. In addition, these analyses do not include all individuals or species in these forests (some marked individuals were excluded within the plot due to boundary issues, and certainly there are other individuals and species beyond the plot boundaries in the wider matrices of forests surrounding the plots). Thus, to avoid artifacts from comparing CNDD across species of different abundances, CNDD should ideally be compared among species at similar abundances.

Conceptual implications: Though there is apparent evidence for a latitudinal gradient in CNDD for species at mean and lower abundances in Fig. R1, the finding of a significant latitudinal gradient in CNDD for rarer species has important implications that should be discussed in this manuscript

beyond what is already discussed therein. Specifically, previous studies have predicted that differences in CNDD for species at lower abundances may be pivotal to maintaining the latitudinal diversity gradient^{1,4,5}. The 2017 *Science* paper (ref 1 here) was often cited in this manuscript, so in due diligence, I read that paper again carefully. There, the authors (including many of the authors of the current manuscript) make a clear prediction that, assuming common species experience some degree of CNDD across the latitudinal gradient, stronger CNDD for rare tropical than for rare temperate species should stabilize rare species more strongly in tropical forests relative to temperate forests. This would, in effect, contribute to greater persistence of rare species in tropical than in temperate forests^{1,4,5}, potentially explaining why tropical forests might represent a “museum of diversity” preserving many more rare species from extinction relative to temperate forests. Of course, other factors that differ across tropical and temperate forests, including differences in speciation rates, likely also contribute to the latitudinal diversity gradient⁶. However, the findings in this manuscript offer strong confirmation of a previously published (i.e., *a priori*) prediction and suggests an important role for latitudinal differences in CNDD for rare and mean-abundance species in reducing extinction rates in tropical relative to temperate forests and contributing to the latitudinal diversity gradient.

Recommendation for authors: A revised version of Fig. 2 that presents appropriate results from the meta-regression model with species abundance and latitude and their interaction (i.e., Fig. R1 above) should be included in a revised version of this manuscript. In addition, the previously predicted and important implications of the authors’ findings relative to significant latitudinal gradients in CNDD for species at rare and mean abundances should be explicitly discussed in a revised version of this manuscript. It should be highlighted that the finding of a latitudinal gradient in CNDD for rare species offers confirmation of a previous prediction about how CNDD might contribute to the latitudinal diversity gradient^{1,4,5}.

Other major concerns about presentation and interpretation of findings

Comparison to CNDD based on spatial patterns of sapling recruitment. The authors also explicitly compare their findings (based on CNDD in sapling survival) to a previous study⁽¹⁾ of CNDD based on spatial patterns of sapling recruitment (e.g., lines 93-97). They mention that differences between these two studies are striking given that the two studies used similar datasets. However, this ignores the fact that studies of CNDD using spatial patterns of sapling recruitment relative to conspecific adults incorporate many more life stages than just sapling survival – indeed, this is likely why so many other processes may potentially influence those spatial measures of CNDD^{7,8}. The current manuscript examines CNDD in sapling survival, but spatial patterns of sapling recruitment from adults incorporate potential CNDD in other life stages, including at least: 1) seed production, 2) dispersal, 3) germination, 4) seedling survival (up to 1 cm DBH) and 5) sapling survival. Thus, the striking thing to me was the fact that there were so many congruities between ref 1 and the current manuscript, given that the current study examines only one life stage of the many life stages examined in ref 1. Both studies found evidence for previously predicted significant relationships between latitude and CNDD for species at rare and intermediate abundances (compare Fig. 2c in ref 1 to the left panel in Fig. R1a above; note the y-axis is reversed in the two figures, so a positive relationship in the former would be the same as a negative relationship in the latter). Thus, the present manuscript seems to offer strong confirmation of a previously-made prediction and finding (i.e., latitudinal gradient in CNDD for species at lower abundances) despite examining CNDD in only one life stage. Meta-analyses suggest that CNDD may be even stronger in the seedling life stage, and previous studies have found CNDD in dispersal and other relevant life stages^{2,9-11}. Thus, one might expect the patterns in CNDD for sapling survival alone to be weaker or non-existent relative to those found in ref 1. All of this should be explicitly acknowledged and discussed as mentioned above.

Reporting of relatively weak CNDD across gradient: The authors state on line 75 and elsewhere that they observed relatively weak CNDD. However, they fail to mention in the main manuscript that the strength of CNDD shown is largely dependent on an arbitrarily chosen standard increase in conspecific density. This is mentioned in the methods (lines 567-568), but the usual reader may not read the detailed methods. This arbitrarily chosen standard increase in conspecific density is equivalent to adding one 2 cm stem at 1 m distance from a focal individual. Thus, the relatively weak values of CNDD reported (i.e., decrease in annual mortality probability associated with a standardized increase in conspecific density) are partially due to the addition of a small conspecific stem whose distance from the focal individual is 50 times greater than the stem's diameter ($1\text{ m} = 2\text{ cm} * 50$). Perhaps the 2 cm stem size was chosen to incorporate a stem size that is common to most if not all species, but this was not clearly described. Also, the rationale for the standard distance of 1 m was not justified. Why place the stem at 1 m? Why not place it at 2 cm or 10 cm distance? Clearly, if the 2 cm stem was placed closer to the focal individual (or a larger stem size chosen), then the values of CNDD would be much higher than those reported here. Thus, it seems odd to speak of relatively weak CNDD when the observation of weak CNDD is largely due to arbitrary choices made during the analysis. I recommend that the authors carefully justify their arbitrary choices here (i.e., why 2 cm and why 1 m distance?) and discuss the implications of those choices (i.e., what if a closer distance had been chosen, how would that affect the interpretation of relatively weak CNDD?).

Other analytical concerns

cloglog vs. piecewise exponential survival models: Analysis of mortality hazard with a cloglog link function and $\log(\text{interval})$ as an offset is an appropriate way to analyze mortality of one interval per individual (because individuals are not included more than once in the dataset). However, it appears that multiple census intervals were used for some of these ForestGEO plots. BCI had 8 censuses (7 census intervals), some plots had 3-4 intervals, and others only had 1 interval. Census number was used as a random effect in the analyses in this manuscript, but this does not completely account for repeated assessments of survival of the same individuals (i.e., in plots with more than 1 interval, individuals were included more than once in the likelihood). Complicating this is the fact that an individual random effect cannot be included in these cloglog models. One way to handle such analyses is with exponential survival models, such as piecewise exponential survival models (e.g., <https://grodrigo.github.io/glms/notes/c7s4>). This could be implemented by using a "poisson" link instead of "binomial" with cloglog link, as described in the website above, and by including a year-in-census effect (where 1 is the first year an individual is in the census, 2 is the second year, etc., regardless of the census number; i.e., an individual first surveyed in the 5th census would get a 1 for that year). I tried this quickly with the BCI data and analysis code that the authors provided – this quick trial revealed very high correlations between estimates of CNDD between the exponential survival approach and the cloglog approach and nearly similar estimates. Given the degree of similarity between estimates from these two approaches, I am not necessarily recommending the exponential survival approach over the cloglog approach. However, it might be worth checking to see if results are indeed reasonably similar using the other plots and discussing trade-offs between different approaches to modeling mortality in the methods section.

Twice as many tropical sites as temperate sites: I noticed that there are twice as many tropical sites as temperate sites in this analysis. I understand that permission for each plot's data must be sought for inclusion in this analysis. However, there are several other temperate plots that have completed at least 2 censuses. Was no attempt made to include more temperate plots so that the number of plots in each zone was approximately equivalent? Either way, I encourage the authors to discuss what impact this might have on their meta-regressions and findings.

Masked data for assessment and reproducibility by readers: I very much appreciate the authors' willingness to share the meta-regression data for evaluation (with species names masked). I also appreciate that the analyses codes are available on GitHub. This was quite helpful for my assessment of the analyses and their interpretation. I suspect that other readers would equally appreciate the opportunity to reproduce the findings of this study, even if using data that are masked. I recognize that this can be difficult with many different PIs involved. However, I strongly recommend that the authors have a plan to make all data necessary to completely reproduce their analyses available to readers in some form (e.g., species names anonymized) if/when the paper is ultimately published. Otherwise, I would have a major concern about reproducibility, for example, if readers of the manuscript cannot access the data directly in some form and instead have to submit 23 separate data-use requests to each set of ForestGEO plot PIs included in these analyses. In my opinion, this presents an undue burden for a reader – what if some of the plot PIs do not approve of a reader accessing their data for reproducibility? If so, then the findings would not really be reproducible. The authors of this manuscript include all relevant ForestGEO plot PIs, and I recommend that there should ideally be some agreement among the authors and plot PIs prior to acceptance that would allow some minimal but complete form of the data (e.g., species names anonymized) to be available to readers who may want to fully reproduce these findings. I know that such arrangements have been made in the past with ForestGEO data, so there is a precedent for this. I do hope that the authors see the potential benefits of such an arrangement for the widespread reproducibility and acceptance of their work.

References

1. LaManna, J. A. *et al.* Plant diversity increases with the strength of negative density dependence at the global scale. *Science* **356**, 1389–1392 (2017).
2. Comita, L. S., Muller-Landau, H. C., Aguilar, S. & Hubbell, S. P. Asymmetric density dependence shapes species abundances in a tropical tree community. *Science* **329**, 330–332 (2010).
3. Mangan, S. A. *et al.* Negative plant-soil feedback predicts tree-species relative abundance in a tropical forest. *Nature* **466**, 752–755 (2010).
4. Yenni, G., Adler, P. B. & Ernest, S. M. Strong self-limitation promotes the persistence of rare species. *Ecology* **93**, 456–461 (2012).
5. Yenni, G., Adler, P. B. & Ernest, S. K. Do persistent rare species experience stronger negative frequency dependence than common species? *Global Ecol. Biogeogr.* **26**, 513–523 (2017).
6. Mittelbach, G. G. *et al.* Evolution and the latitudinal diversity gradient: speciation, extinction and biogeography. *Ecol. Lett.* **10**, 315–331 (2007).
7. Hülsmann, L. & Hartig, F. Comment on “Plant diversity increases with the strength of negative density dependence at the global scale”. *Science* **360**, eaar2435 (2018).
8. LaManna, J. A., Mangan, S. A. & Myers, J. A. Conspecific negative density dependence and why its study should not be abandoned. *Ecosphere* **12**, e03322 (2021).
9. Jansen, P. A., Visser, M. D., Joseph Wright, S., Rutten, G. & Muller-Landau, H. C. Negative density dependence of seed dispersal and seedling recruitment in a Neotropical palm. *Ecol. Lett.* **17**, 1111–1120 (2014).
10. Comita, L. S. *et al.* Testing predictions of the Janzen–Connell hypothesis: a meta-analysis of experimental evidence for distance- and density-dependent seed and seedling survival. *J. Ecol.* **102**, 845–856 (2014).
11. Song, X., Lim, J. Y., Yang, J. & Luskin, M. S. When do Janzen–Connell effects matter? A phylogenetic meta-analysis of conspecific negative distance and density dependence experiments. *Ecology Letters* **24**, 608–620 (2021).

Referee #3 (Remarks to the Author):

The authors test the Janzen-Connell hypothesis that host-specific herbivores, pathogens, or other natural enemies make the areas near a parent tree inhospitable for the survival of seedlings using a dataset of tree mortality along a latitudinal gradient.

Novelty:

At several points in the article, the authors state that no relationship between CNDD and latitude was found by previous research (lines 36-38) (e.g.: 1-3) (the conclusion of the submitted manuscript), but that the statistics were not adequate. If a pattern was found with latitude, methodological limitations were again mentioned as a reason (lines 41-44). So while at this point it is certainly still a methodologically and statistically contentious discussion, the overall results presented are not necessarily new. The most novel part of the study is certainly the relationship between CNDD, rare species, and latitude. Again, the results have been shown before by another study but have been criticized.

Also, I would suggest the authors change the tone of the introduction a bit. At the moment, one gets the impression that all the previous studies are flawed and that only the study presented uses appropriate statistics, even though the previous authors came to the same conclusions. Although the authors certainly address some of the limitations of the previous studies (e.g. between-species and between-site effects), they only test a limited set of confounding variables. I suggest that the authors instead show what is really new about the study presented. Although the article is generally well written, statistically I have some comments below, which the authors might be able to answer. However, in light of my previous comments, I wonder if it would not fit better in a more specialized journal, but that's something that should be decided by the editorial board.

Methods:

The authors use only two different predictors here, namely latitude and species diversity. As the authors mention, the CNDD would be higher in tropical forests than in temperate forests. Although latitude is an indicator for tropical and temperate forests, it is not necessarily the one that best fits the theory. The Janzen-Connell hypothesis states that these are tropical rainforests, which in this case is a categorical variable and not a continuous one. Why should the latitude 'within' the rainforests play a role? It is also unclear why only these two predictors were tested. The sample size of 23 is also relatively small, which might explain to some extent why a) the significance is low and b) more predictors were not used. It is clear that there is much more data behind it at each site, since time series of tree mortality are used, but in the end, the conclusions are based on 23 data points. As the authors mention, the data set is unbalanced, most likely also with regard to covariates. Thus, 22 data points are really at the bottom of the list of observations needed to prove that there is no effect at all. The authors have tested the robustness of the model, but they cannot verify precisely whether the fact that they do not observe a significant effect is simply due to the small sample size. On this point, the authors need to state more clearly in a revised manuscript that they do not observe low significance solely because of the small sample size. Furthermore, the pattern they observe between rare species, CNDD, and latitude could be confusing here, as with an additional factor in the meta-regression, the required sample size would be even higher.

To publish this manuscript in Nature, the authors should:

1. Clearly show how their results are new from what we already know. At the moment this is not clear. 2 . Show that the low sampling size does influence their conclusions here. The sampling really is at the lower side to answer the question if the latitudinal gradient does have an influence or not.

Minor comments:

Line 13: What is stabilized here? Please clarify.

Line 26: Stable with respect to what exactly? Please clarify.

Line 45: Since you use latitude in the manuscript geographically, longitudinal might be misunderstood here.

Line 75: What is relatively weak in numbers? What would you expect to be strong? Could you give an example. E.g. a strong effect would be....

Line 135-137: This seems counterintuitive here. If CNDD is more effective in the tropics in controlling abundances, would that on another scale also spill over to diversity?

Line 168: Strictly speaking you are testing for latitude only (continuous), not tropical vs. temperate (categorical). Please clarify.

Line 217: You do not look at 'specialized natural enemies' in you analysis specifically, unless you use it as a synonym here for something else. It's part of the Janzen-Connell hypothesis, but not specifically addressed in this study specifically.

Figure 2: Why is the y-axis going from -0.5 to 1.5? That clearly distorts the slope of the correlation. If you plot it within the CNDD range of the data you would see an increase in CNDD with decreasing absolute latitude, although not significant. Why are there no datapoints shown in b?

References.

1. Comita, L. S. et al. Testing predictions of the Janzen–Connell hypothesis: a meta-analysis of experimental evidence for distance- and density-dependent seed and seedling survival. *Journal of Ecology* 102, 845–856 (2014).
2. Hyatt, L. A. et al. The distance dependence prediction of the Janzen-Connell hypothesis: a meta-analysis. *Oikos* 103, 590–602 (2003).
3. Hille Ris Lambers, J., Clark, J. S. & Beckage, B. Density-dependent mortality and the latitudinal gradient in species diversity. *Nature* 417, 732–735 (2002).

**Author Rebuttals to Initial Comments:
Revision for Nature 2022-10-17122**

I. Referees' comments:

A. Referee #1 (Remarks to the Author):

The manuscript addresses questions that have long been puzzling ecologists: What mechanisms allow so many tree species to coexist in species-rich forests, and does the role of these vary across latitudes? The authors report the results of a study in which the likelihood of mortality of individual trees was investigated in relation to the density of conspecific and heterospecific tree individuals in the neighbourhood. Using a large data set compiled from 23 forest dynamics plots spanning a latitudinal gradient of more than 50 degrees, they explore the potential role of conspecific negative density-dependence (CNDD) in promoting species coexistence in forest tree communities across latitudes. This is the first attempt to assess latitudinal patterns in CNDD using tree mortality data rather than static tree distribution data. By doing this, the study avoids the statistical biases that some previous studies have been criticised for.

Three main results are presented and discussed: 1) Contrary to predictions, the average CNDD does not vary across latitudes (and is relatively weak overall); 2) Abundant species experience stronger CNDD at tropical sites but not at temperate sites, and 3) The degree of interspecific variation in CNDD tends to be high within tree communities. Taken together, these results advance our understanding of the ways in which CNDD may contribute to observed latitudinal patterns of forest tree diversity: The first result is interesting and important since it is in contrast with that from a study by La Manna et. al. (2017; Science) which was subsequently criticised due to potential statistical biases. The second result provides more conclusive evidence for CNDD being stronger in rare than in common species in tropical forests (a pattern which has been reported in previous studies which were subsequently criticised for statistical biases). The third result implies that the stabilising role of CNDD may be less pronounced in forest systems than what has often been assumed.

The study appears methodologically sound, and the analyses expertly done. The analytical pipeline involves several decisions which could potentially influence the results, but methodological choices are backed up by references to relevant literature and randomisation studies have been conducted to evaluate the robustness of the results. The authors have also explored the role of some potential confounding factors that could drive some of the reported relationships (e.g. between rarity and CNDD). Altogether, the conclusions seem robust and well supported by the data.

We thank the referee for their comprehensive and supportive feedback, which helped us to further improve the manuscript. In the following, we respond to all comments in detail.

The presentation is clear and logically structured. I have identified a few points that might benefit from some more information and/or rewording:

The way the Janzen-Connell hypothesis is introduced and cited (e.g. L8-9; L65; L77) might be slightly misleading for readers who are not familiar with the original papers by Janzen and Connell. Although both Janzen and Connell mention the possibility that the role of natural enemies in controlling the abundances of individual species and thereby preventing competitive exclusion is likely to vary across latitudes this is not dealt with in huge detail in the original publications (which focus primarily on describing a potential mechanism that could explain the enigma of high species richness in tropical forests). Some of the current wording might be interpreted as if the Janzen-Connell hypothesis *is* the specific idea that CNDD will vary in strength across latitudes. This could perhaps be clarified for the broader readership who might not have read the original papers.

It is difficult to conclusively prove what the authors really had in mind when writing their papers over 50 years ago, also because new concepts have influenced the reception of these papers over time. Nevertheless, we do understand their intention differently. The second sentence in Janzen (1970) states that:

“Despite reports that adults of some species of lowland tropical trees show clumped distributions (Poore 1968; Ashton 1969), I believe that a third generalization is possible about tropical tree species as contrasted with temperate ones: for most species of lowland tropical trees, adults do not produce new adults in their immediate vicinity (where most seeds fall).”

Our interpretation of this passage is that at least Janzen (1970) believed that CNDD is stronger in the tropics compared to the temperate zone. It is true that this claim is less pronounced in Connell (1971), who concentrates more on local variation in diversity; and that many of the early studies that focused on the question of whether CNDD exists at all were conducted only in the tropics (see, e.g., Hülsmann et al. 2021 for a review), but we would again argue that a difference in the strength of CNDD in the temperate vs. the tropical zone is formulated by Janzen (1970) and also in some sense required for the overall idea, because if there was no difference between latitudes, how could the JC hypothesis “explain” the differences in local diversity between tropical and temperate zones?

We have nevertheless checked all the places the referee mentions and have added further references where we thought this would be helpful (L9-10, L72).

One of the main take home messages is that the results from the analyses presented in the manuscript do not support hypothesised predictions inspired by ideas in Janzen’s and Connell’s classical papers. As the authors point out (L199-L201), these results may or may not hold if similar analyses were to be conducted for other life stages. This point is important and could perhaps be reiterated in a more prominent place in the manuscript. The reported weak levels of CNDD might not be too surprising given that the analyses focused on the mortality of individuals in size classes that may already be way past the stage at which mortality caused by host-specific enemies might be most important (e.g. early seedling stage).

We agree that this is a limitation and have mentioned this possibility now more prominently in L207-212: “A further consideration is that species abundances are controlled by processes occurring throughout the entire demographic cycle, not only by mortality during the life stages considered here. It is possible that CNDD analyses of other vital rates and life stages,

particularly earlier ones, would lead to stronger CNDD and different patterns and conclusions²⁰, because the interaction between ontogenetic and demographic processes may change with latitude.”

The study uses an impressive data set collected through a network of study sites. The map on the ForestGEO website (<https://forestgeo.si.edu/>) suggests that many sites have not been included. There are probably good reasons for selecting the current 23 sites, but it would be good to know why the full network of sites was not utilised and these particular 23 sites chosen for the analyses.

We agree that the ForestGEO network is an outstanding data source. Regarding the selection of the forest plots: we requested data from all plots that were at least ten hectares in size (but later got data from a smaller subset for one plot) and that had at least two censuses which is required for mortality analyses (this resulted in a total of 36 sites). In the ForestGEO network, it is then up to the plot PIs to make the final decision whether they want to contribute their data to a particular study. We included all 23 sites for which we received permission. We have added information on the selection of the forest plots in the methods section (L459ff).

I also came across several more specific points in the text that would benefit from clarification or rewriting, and some details that could be clarified in the figures and tables (listed below with associated line numbers):

Thanks for the helpful suggestions. We have implemented most of them and provide further clarification below.

L12 ‘latitudinal patterns in species-specific CNDD’: This wording could possibly be misinterpreted as if you examined how CNDD of individual species occurring at multiple sites varied across latitude?

Modified to “of latitudinal CNDD patterns using dynamic mortality data to estimate species-site-specific CNDD across 23 sites” (L12-13).

L36: Is the word ‘fiercely’ needed here? As far as I remember, the two papers cited are not particularly fierce in their contents.

“Fiercely” has been deleted (L40).

L43: differs in -> differs between?

Done (L48-49).

L46: could perhaps indicate what you mean by ‘large’ in this context?

Done (L51).

L46: is the word ‘globally’ needed here? (Yes, sites are distributed across several continents, but still many parts of the globe that are not covered.)

“Globally distributed” has been omitted (L51).

L54-55: Based on information provided up to this point in the text, it is not clear what the relative change in mortality probability of trees induced by an additional conspecific neighbour means. I would suggest adding some detail (conspecific individual of dbh 2cm located 1 m from the tree?) or refer to methods section. Same applies to L70 and L72.

We have the impression that the introduction would become too technical if we added more details about the change in conspecific density. We therefore refer to the methods section as proposed by the referee. Additionally, at the point where we present the first results on stabilizing CNDD (L78-80), we have added the relevant details about the additional conspecific neighbor. We think that this information is best placed there to provide context for the estimates of stabilizing CNDD. Moreover, as we explain in more detail below, the exact change in conspecific density is not crucial because we are ultimately interested in the marginal effect of increasing conspecific density. We have therefore introduced the term ‘small density perturbation’ (see below) (L60, L78).

L119-120: This sentence comes across as a bit repetitive (similar information given in the previous paragraph).

We agree that the sentence repeats the pattern we described in more detail in the previous paragraph, but we believe that this condensed statement is a helpful reminder for the reader contributing to the overall message of the manuscript.

L194: abundance -> abundances?

Done (L204).

L199: ‘(...) not only mortality during early life stages’ – this phrasing might imply that the current study deals with the earliest life stages of trees which isn’t necessarily true. Many individuals might have died before reaching the 1cm DBH threshold of the ForestGEO plot data, and some species might reach maturity at smaller sizes than 10cm DBH.

We agree and have modified the phrasing to “not only by mortality during the life stages considered here” (L209).

Table 1: Is the ‘Dataset’ column needed given that there is only one entry (raises the question of what other data sets than the ‘original’ data set there are)? If included, could specify that n (presumably) refers to number of species rather than anything else?

True. The column has been removed and the number of species-site-specific CNDD estimates has been added to the caption (L376).

Table 1: Not immediately clear what the ‘t’s in front of the predictors mean

The 't's stand for transformed. We have added this information in the table caption (L379-382).

L301 (and elsewhere): Is 'absolute latitude' is the correct term here? I believe that you used the distance (degrees) from the equator to test for latitudinal effects, but did not distinguish between sites on the Northern versus Southern hemisphere? Googling 'absolute latitude' did not yield any useful hits. Please confirm that the term is more widely established and used.

"Absolute latitude" is exactly what the referee assumes: the distance (in degrees) to the equator not distinguishing between the northern and southern hemisphere. The approach and the term are common in latitudinal analyses (e.g. Lamanna et al. 2014, Nishizawa et al. 2022). We now define the term and what we consider it to be a proxy for in the Methods (L643-646).

L366: log transformed trees -> log-transformed number of trees.

Done (L379).

Fig. 1: This figure is informative, but there is quite a lot going on in terms of colour coding, symbol sizes etc which makes me doubt whether all this information be visible in the printed version. I was struggling to pick everything up (e.g. circles vs diamonds) even when zooming in heavily on my screen.

We agree that the figure is complicated, but we also find it crucial to provide this information because it forms the basis for the entire rest of the study. To aid comprehension, we have rearranged the panels around the map to provide more space for each panel and optimized the y-axis limits so that the relationship between abundance and CNDD is more visible (see Fig.A). This relationship is the crucial information in this figure. We hope that the modifications have improved overall readability. Also, note that the original figure (see pdf version), which is already optimized for the dimensions and fonts of figures in Nature, is larger than it appears here and in the manuscript for review. We hope that remaining issues can be solved with the Nature production team.

Fig.A | Rearranged main Fig.1.

L388-390 (& L419-420): “Locations of forest sites and CNDD-abundance relationships are colored by latitude (tropical forests in reddish colors, temperate in bluish).” What about orange/yellow colours? Do the subtle differences in e.g. red colours have a meaning? If the idea is to allow cross-reference to data points in other figures (e.g. 3) colour contrasts appear too small.

The colors were taken from a gradient from red (tropics) over orange and yellow to blue (temperate). Therefore, there are subtle differences, e.g., among the reddish colors. These illustrate latitude but should not allow cross-reference to other figures. We have clarified in the figure caption that the color gradient goes “from tropical forests in red-orange to subtropical forests in yellow-green and temperate forests in blue” (L404-406).

Fig. 2: For maximum clarity, you could perhaps explain what the horizontal lines at $y = 0$ depict? Likewise, in Fig. 3a it might be helpful to more explicitly mention why there is a dotted line at $y = 0.4$ (not all readers might dig into the specifics of the methods sections). (Same comments are also relevant in the context of matching Extended Data figures.)

Done (L403-404). For details on the visualization in the matching Extended Data Figures, we refer to main Fig.2.

L418-419: 'fitted individually per forest site' – would 'fitted separately for each forest site' be clearer?

Done (L425).

L437: Add citation to R

Done (L454).

L442: Are stems shorter than 1.3m not recorded, regardless of their thickness?

Yes, exactly.

L452: the meaning on 'potential confounders of this relationship' is not immediately clear.

Thank you, we now refer to the respective subsection where the confounders are explained in more detail (L471).

L453-454: As hinted above, for some species, trees of this size might not necessarily represent 'early life stages'.

We have added that the analyses were carried out using trees between 1 and 10 cm DBH and rephrased "early" to "earlier" (L472-473).

L464: Not crystal clear what 'resurrected trees' mean in this case.

Rephrased to "trees found alive after being recorded as dead in a previous census were set to 'alive'" (L483-484).

L496-497: I can see that species-specific optimisation of mu-values might not be feasible, but similar mu values across species may also be unrealistic if species are (following predictions from the Janzen-Connell hypothesis) attacked by different sets of host specific enemies that potentially operate at different spatial scales. Could this averaging across species have had any effects on your overall results?

Indeed, we assume that the distance decay parameter is identical for all species. The explicit optimization of mu in this study is an improvement over many earlier studies that used a fixed neighborhood or compared just a few alternative definitions (e.g., Wu et al. 2016, Zhu et al. 2018, Brown et al. 2021).

Estimating species-specific optimization would have been possible, but we decided against it for two reasons: The first is that optimizing the parameter separately for each species would induce considerable uncertainty and instability in the estimate for each single species, in particular for rare species that typically have fewer observations. This statistical error has to

be considered against the possible bias created by neglecting species-specific differences. Although both are a problem, we believe that the trade-off is in favor of estimating a global value for rare species. A second reason for preferring a fixed value is that with species-specific density definitions, CNDD slopes would not be directly comparable between species, which would create considerable problems for our downstream analysis. For both reasons, we believe that our current approach is the most parsimonious and practical solution.

Nevertheless, it is possible that the true species-specific μ values may vary, and the referee is right to ask what effect this might have on the analysis. An incorrect μ value could lead to an error in the assumed effective density experienced by individuals of that species. This should then lead to a regression dilution effect, which in our analysis pipeline would lead to an underestimation of CNDD.

We do not believe that this is a big concern, however. As we stress in the introduction of the paper, the main advantage of our longitudinal (mortality) CNDD analysis over earlier CNDD studies based on size-structure is that this regression dilution effect is conservative, i.e., it creates a bias towards lower CNDD. As the “effective density kernel felt by each individual” cannot be directly measured and likely varies not only between species, but also within species with topography, environmental variables etc., it was a main goal of this analysis to achieve robustness against the regression dilution problem (see also Detto et al. 2019).

The only way in which this possible regression dilution could lead to wrong conclusions in our framework is if the error due to species-specific μ values correlated with our main predictors (i.e., abundance and latitude). In this case, the regression dilution bias could have a confounding effect on our analyses. Therefore, we tested whether the species-specific optimal μ values depend on latitude and abundance for those species for which the grid search yielded a distinct optimum of the log likelihood (Fig.B). We find no pattern of species-specific optima in μ for conspecific densities with latitude or species abundance.

We included this result in Supplementary Fig.3 and discuss it in L524ff.

Fig.B | Relationship between species-specific optima in μ for conspecific densities and (a) absolute latitude and (b) species abundance. Species-specific optima in μ are shown for those species for which the grid search yielded a distinct optimum of the log likelihood ($n = 1207$) and for the setting

that was optimal overall, i.e., exponential distance decay and densities measured as basal area. Blue dots and smoothed densities were obtained through 2D kernel density estimation. Red regression lines and 95% confidence intervals were obtained from generalized additive quantile regressions with smoothing splines for the 0.25, 0.5, and 0.75 quantile in each panel.

L510: tradition -> traditional

Done (L534).

L530-L531: 'at least 20 alive and dead status observations each' – Just to check, does this mean at least 20 alive status observations AND at least 20 dead status observations? You could perhaps also mention what proportion of species did not fulfil this criterion (and were therefore lumped into one of the rare species categories)?

Yes, the referee's interpretation is correct. We have added the overall percentage of species for which species-specific CNDD estimates could not be calculated (L558) and refer to Extended Data Table 2 for more detailed numbers per site (L559-560).

L550: remove commas around 'relative and absolute change' for easier reading.

Done (L574).

L561: If I understand correctly, the DBH = 2cm and 1m distance scenario is an arbitrary choice made by the authors. A few words justifying the use of these specific values (rather than some other size and/or distance) might be helpful.

This increase in conspecific density was chosen because it is relatively small and therefore within the range of observed conspecific densities even for rare species. Larger values could cause extrapolation problems. We have added more explanation on this choice (L584ff).

Moreover, we would like to clarify that for comparing between sites and species (which is our main purpose here), the exact choice of the change in conspecific density for which the mortality response is calculated ('density perturbation') is less relevant – as long as the range of observed values is not critically exceeded, which would cause extrapolation (cf. L587). We now explicitly refer to this change in conspecific density as a 'small perturbation in conspecific density' (e.g., L60 and L586) to emphasize this point.

The choice of the density perturbation would only be important if we wanted to make statements about the absolute (not relative) effect of CNDD on mortality. In our previous submission, we did not such a statement about the absolute CNDD effect sizes. Inspired by this comment, however, we realized that an assessment of the absolute effect (and thus the ecological relevance) of CNDD would be a valuable addition to our analyses.

We therefore calculated the relative increase in mortality over the species-specific interquantile ranges in conspecific density (Fig.C). These estimates provide a more interpretable quantification of the typical CNDD effects for each species in the community and thus allow to assess how important CNDD is for each species. Note, however, that in turn, the values are now not comparable between species or latitudes, as they are calculated with a change in conspecific density that is different for each species and community.

We find that CNDD effects between the first and third quantile ranged between -2.39 and 8.03 % relative change in mortality (Fig.C), with a mean from meta-regression of 6.64 %, which we consider large enough to be potentially relevant for population growth rates and community assembly. Based on these new results, we expanded on the average strength of CNDD in the text (L74-76) and included Fig.C as Supplementary Fig.4 (Methods explained in L594ff).

Fig.C | Distribution of stabilizing CNDD (rAMEs) calculated over species-site-specific interquartile ranges in conspecific density. Besides the frequency distribution of species-site-specific estimates, the figure indicates the global average assessed through meta-regression with random intercepts for sites and species in sites (red diamond with 95% confidence interval) and the interquartile range of the estimates. Note that 1% of the estimates are outside the limits of the x-axis.

L580: could perhaps change ‘pathogens’ to the more general word ‘pest’ since the specific enemies contributing to conspecific density-dependent mortality are typically not known.

We realized that the actual mechanisms are not really relevant for this sentence and therefore have removed this part. Moreover, we have made sure that we use both herbivores and pathogens or the more general term ‘(specialized) natural enemies’ in all parts of the manuscript (e.g., L28-30, 205, L230ff).

Extended Data figure 5: A bit more explanation on what is shown would be helpful. I assume that dots are individual species; what do the blue ‘clouds’ depict? Explain curves and confidence intervals.

Done (L929, now Supplementary Fig.1). Blue dots were removed to improve readability.

Extended Data figure 6: Explain colour scheme.

Done (L940, now Supplementary Fig.2).

Extended Data Table 2: A bit more detail would be helpful to interpret the numbers in columns 'N trees' and 'N mortality observations'. Is 'N trees' the overall number of tree individuals with a unique tag that were included in the study (some of them across several census periods)? Does 'N mortality observations' show the total number of status recordings (dead or alive) conducted across all species and census periods? If so, I wonder whether the column heading could be changed to 'N status observations' for increased clarity? On that note, it might be helpful to somewhere in the text or in a table give an indication of how common tree mortality is in the studied size classes of trees. This will obviously vary a lot among species and sites, but might still be helpful background information to readers who are not very familiar with tropical forest ecology.

Thank you! We have now better explained the numbers in all columns and added the percentage of dead trees. Note that these numbers do not correspond to mortality rates because of varying interval lengths. We have removed the column 'N trees' as, in the strict sense, it is not a property of the data used for the mortality analyses (L879ff).

B. Referee #2 (Remarks to the Author):

Hülsmann et al. present a study of latitudinal patterns in conspecific negative density dependence (CNDD), which they also call stabilizing CNDD. The authors use data from multiple censuses at 23 ForestGEO plots (large globally-distributed forest inventory plots with all stems >1 cm DBH mapped and measured every ~5 years) to calculate CNDD for each species at each site. They calculate CNDD as the change in annual mortality probability associated with a standardized increase in conspecific density while holding total density constant. Latitudinal patterns across species and sites are then analyzed with meta-regressions that account for hierarchical variability among sites and species as well as within species. The authors report several key findings. First, the authors report no change in mean species CNDD across latitudes. This finding appears to contradict earlier findings of a latitudinal gradient in mean species CNDD 1. Second, the authors report that CNDD varies with species abundance among tropical species (stronger stabilizing CNDD for rare tropical than common tropical species) but not among temperate species. They note that this finding is similar to that reported in a previous study 1. Third, the authors note similar variability in CNDD among species within each site (greater variability in the temperate zone may have indicated greater fitness differences and therefore weaker stabilization). The authors conclude that CNDD may contribute to the latitudinal species diversity gradient, but in more nuanced ways than previously thought – CNDD has greater effectiveness at controlling species abundances in tropical latitudes than at temperate latitudes “despite CNDD being present at comparable levels at all latitudes” (lines 178-179).

The large research team seems well qualified to conduct these analyses, and these ForestGEO data are quite impressive. However, I have several concerns with the analyses and, more critically, major concerns with the presentation and interpretation of findings from these analyses. My primary concerns are that some key findings are not currently presented properly or discussed in the manuscript, so I detail those primary concerns first. I follow up with several other concerns for the authors to consider about the presentation of

their findings. Finally, I offer a few concerns about the analyses themselves. While I suspect that these analytical concerns may not substantially affect the key findings, I encourage the authors to consider and verify this.

We would like to thank the referee for the constructive criticism and careful review of our analyses and their presentation and interpretation, which even involved rerunning our code and analysis pipeline. We have thoroughly examined the points of criticism and made adjustments where we consider it beneficial. In particular, this allowed us to sharpen the conceptual underpinnings of our work and the interpretation of the results. More detailed answers and explanations below.

Primary concern about presentation and interpretation of findings

As mentioned above, the authors report no latitudinal trend in average species CNDD. They base this assertion on meta-regressions of species-by-site CNDD estimates across the latitudinal gradient. I have no issues with the meta-regressions as carried out – they seem quite sound. However, I do have a major concern with the presentation and interpretation of these meta-regressions. Two primary explanatory variables were used in the meta-regressions: latitude and species abundance. Both have previously been found to have potential influences on CNDD 1–3. Moreover, one previous study (as mentioned by the authors) found a significant interaction between the two 1. Indeed, the present manuscript reports a similar significant interaction between latitude and species abundance. However, when reporting no latitudinal trend in CNDD, the authors use a different meta-regression model than the model that shows a significant interaction between latitude and species abundance that only includes latitude (i.e., not species abundance). ***This presentation is problematic for several reasons and, importantly, obscures some of the authors' most important findings – significant relationships between CNDD and latitude for rare species as well as for species at mean and median abundances (see Fig. R1 below).*** Figure R1 was produced by me from the same data, model, R code, and even plotting functions that were provided by the authors. In fact, using the code and data provided by the authors, I was able to determine that a significant latitudinal gradient in CNDD is evident for 57% of species in this study (i.e., all species with abundances lower than 12 trees / ha; mean log(abundance) across species = 10.2 trees / ha).

Figure R1. Same as Fig. 2 in manuscript, except this figure (unlike Fig. 2) uses the same meta-regression model to produce relationships in panels A and B (a meta-regression with latitude, species abundance, and their significant interaction as fixed effects). (A) A significant relationship between CNDD and latitude was found for species at rare and intermediate (mean) abundances but not for common species. (B) Same as Fig. 2b in manuscript. This figure was produced with the same meta-regression models, data, code, and plotting functions as provided by the authors. Fig. 2 in the manuscript uses two different meta-regression models for panels A and B. Fig. 2a uses a meta-regression model with latitude only, and Fig. 2b uses a meta-regression model with latitude, species abundance, and their significant interaction.

Such latitudinal gradients in CNDD for species at lower abundances have been predicted in previous literature to contribute to the latitudinal gradient in species diversity. Specifically, if common species experience some degree of stabilization via CNDD across the latitudinal gradient (right panel in Fig. R1a), then stronger CNDD for rare and intermediately-abundant tropical than temperate species (left and center panels in Fig. R1a) should stabilize rare species more strongly in tropical forests relative to temperate forests. This would be expected to lead to the persistence of more rare species and even more intermediately-abundant species in tropical than in temperate forests, contributing to lower extinction rates in tropical than in temperate forests and potentially explaining why tropical forests are sometimes thought to be “museums of diversity” 1,4,5.

We thank the referee for these detailed comments. In these statements, the referee proposes three modifications for the representation and interpretation of our results, which are partly repeated with a slightly different focus below:

- A. The interaction plot with latitude as predictor (Fig. R1a) for different (fixed) levels of abundance shows that there is a latitudinal gradient in CNDD (which seems to confirm expectations of the Janzen-Connell hypothesis and to contradict our analysis which shows no trend when we test CNDD against latitude).
- B. The interaction of abundance and latitude should be independently interpreted for both main effects.
- C. Stronger CNDD for rare species suggests that there is stronger community stabilization via CNDD in the tropics.

For the purpose of clarity, we address each point in detail here first, explain the extent to which we have made changes to the manuscript, and refer to these detailed responses where the topics reappear in the rest of the text.

A) How to quantify CNDD across latitude

In our view, the Janzen–Connell hypothesis states that average CNDD in the community is higher in tropical than in temperate forest communities. We therefore directly regressed CNDD against latitude.

The referee suggests instead using our regression model with both main terms and the interaction of abundance and latitude and plotting CNDD across latitude for species with a fixed abundance of, e.g. 1 individual/ha or 10 individuals/ha. We agree that such a plot can be produced, and we are not surprised by the results, as the effects displayed in the figure above are, albeit less explicit, also visible in our Fig. 2b (note that interactions effects in OLS are symmetric). Thus, given that we report a significant interaction and a CNDD-abundance relationship that is only strong in the tropics (our Fig. 2b, or R1b produced by the referee), it follows from the interaction symmetry that rare species will have stronger CNDD in the tropics than in temperate forests (as displayed in Fig. R1a). We had originally produced Fig. R1a, but ultimately decided not to include it in the manuscript for the reasons discussed in the following paragraphs.

One reason that the plot R1a does not seem relevant to us for the purpose of testing the JC hypothesis is that, as stated above, the JC hypothesis speaks about average CNDD in a community. Thus, our goal in this analysis is to estimate how average CNDD varies across latitude (our first prediction, L70ff).

However, when using the model with an interaction, the abundance is set to a reference value (e.g., mean or median abundance, like the referee did in the middle panel of Figure R1). This implicitly assumes that plots in the temperate and tropical zones have the same mean abundance, which is not the case in the empirical data (see Fig.D, which corresponds to Extended Data Fig.5c in the manuscript).

Fig.D | Relationship between species abundance and absolute latitude. Smoothed densities (blue) were obtained through 2D kernel density estimation. Regression lines and 95% confidence intervals (red) were obtained from generalized additive quantile regressions with smoothing splines for the 0.25, 0.5, and 0.75 % quantile in each panel.

In other words, Fig. R1a compares CNDD effects across latitudes for a fixed abundance, but a species with 10 trees / ha may be rare in a temperate, but not rare in a tropical forest. The referee similarly indicates below that species abundance distribution change with latitude. Thus, we view Fig. R1a as interesting, but misleading for testing the JC hypothesis, and we believe that the univariate regression of CNDD versus latitude is the most appropriate way to test for this question. We note that this is the way it has also been tested in previous latitudinal CNDD studies (Johnson et al. 2012, LaManna et al. 2017).

We hope that this explanation clarifies our reasoning and approach. Moreover, we have made sure that in all cases where we refer to this latitudinal pattern (e.g., L71ff, L104-105, L179, or L233-234), we explicitly speak of ‘average CNDD’.

B) How to interpret the model with an interaction of latitude and abundance

Related to point A, and independent of the JC test, the referee suggests that the interaction abundance * latitude should be interpreted both from the viewpoint of latitude moderating the effect of abundance and abundance moderating the effect of latitude.

We have added this alternative representation in Extended Data Fig.5 and refer to it in a corresponding verbal interpretation in the text (L123-125), but we have decided against using it in the main text for the following reason:

The two representations of the interaction (cf. Fig.R1 a and b) are not independent lines of evidence, but two sides of the same coin, and we have to decide in which direction we assume the causal mechanism to work.

Our interpretation is that environmental differences across latitude causally affect the CNDD ~ abundance relationship and not vice versa (technically, we view latitude as a moderator of the CNDD ~ abundance relationship). We therefore prefer to show the plot as presented in the original submission.

This representation has the additional advantage that each panel in Fig.2b can be seen as an idealized community at the respective latitude. We find this focus helpful as this is the level at which stabilization occurs. In this way, stability is not assessed using the interaction coefficients of individual (rare) species, but of all species within a community (Broekman et al. 2019). Along these lines, we also consider it misleading to state that "a significant latitudinal gradient in CNDD is evident for 57% of species" (we assume that the referee means that species with intermediate and rare abundances make up 57% of all species and both have a significant latitudinal gradient).

C) Is there evidence that there is stronger stabilization in the tropics

The referee argues that "*stronger CNDD for rare and intermediately-abundant tropical than temperate species (left and center panels in Fig. R1a) should stabilize rare species more strongly in tropical forests relative to temperate forests.*" We presented this result as a stronger correlation of CNDD with abundances in the tropics (see our response B) and agree with the interpretation that this result provides potential evidence that CNDD may be more effective in the tropics, which we consider to be consistent with the Janzen-Connell hypothesis (L189ff). This is the statement that we already made in the originally submitted version of the paper.

The referee goes further and concludes that forest communities in the tropics must therefore also be more strongly stabilized. Theoretical research of the last years, however, has shown that we have to be careful with such conclusions, as it is not guaranteed that stronger mean CNDD also leads to stronger community stabilization (see discussions in Stump and Comita 2018, Broekman et al. 2019, May et al. 2020, Hülsmann et al. 2021). In particular, if CNDD varies considerably between species, CNDD can actually be destabilizing (Stump and Comita 2018, May et al. 2020). These findings are the reason why we added an analysis on the coefficient of variation (CV) of CNDD in the paper (third prediction, Fig.3, L147ff). Although we did not find a latitudinal gradient in the within-plot heterogeneity, i.e., interspecific variability in CNDD, we did find a level of heterogeneity that has been found to be de-stabilizing in theoretical studies, which casts severe doubt on whether we should equate stronger CNDD effectiveness with stronger community stabilization.

For these reasons, we prefer not to speculate too much about the implications of our results for community stabilization and species coexistence or for patterns at macroevolutionary scales. Overall, we find evidence for one out of the three predictions we tested and therefore interpret our results as "limited support" (L229-230) for the Janzen-Connell hypothesis and suggest "a novel, refined interpretation of this classical idea" (L232ff). To better reflect this conclusion in our summary, we have rephrased its end as follows: "Overall, our results support the idea that CNDD, although not stronger, may be more effective in regulating population abundances in tropical than in temperate forests. Provided that interspecific variability in CNDD is not overly destabilizing, this may translate into greater

stabilization of diverse tropical tree communities.” (L19-23). We thank the referee for the comment, which has helped to sharpen our manuscript with respect to this distinction.

The need to interpret two main effects in light of a significant interaction: Main effects that are involved in a significant interaction with another main effect should be interpreted at the different levels of each main effect. This is what is done for species abundance in Fig. 2b in the manuscript. The relationship between CNDD and species abundance changes across different latitudes, going from a positive relationship in tropical latitudes to no relationship in temperate latitudes (hence the three panels for Fig. 2b in the manuscript). In the same way, the significant interaction between latitude and species abundance also means that the relationship between CNDD and latitude changes with species abundance (Fig. R1a). There is a significant latitudinal gradient in CNDD for rare species, still a latitudinal gradient in CNDD (but weaker) for species at mean abundances, and the latitudinal gradient in CNDD is not present for common species. This significant interaction between latitude and species abundance was bolded in Table 1 in the manuscript, and the significant relationship between CNDD and latitude for rare species (defined by the authors as a species with 1 tree / ha) was also bolded in Table 1 ($p = 0.02$; this is the same relationship as shown in the upper left panel of Fig. R1).

See our response B). We have added the alternative representation of the interaction in Extended Data Fig.5 but would like to emphasize that it doesn't provide additional evidence for a latitudinal gradient in stabilizing CNDD.

The authors were kind enough to provide the species-by-site CNDD estimates used in their meta-regressions for my evaluation. This is how I was able to produce Fig. R1 – code was already written in their scripts to produce Fig. R1 as seen here with all 6 panels – all I had to do was slightly modify their code to use the same meta-regression model to produce both panels A and B in Fig. 2. Instead of presenting both main effects in light of the level of the other (as shown here in Fig. R1 and written in the authors' code as such), the authors appeared to adjust their scripts so that panel A shows the results from a different meta-regression model that includes only latitude, while panel B shows results from the model with both latitude and species abundances (and their significant interaction).

See our response A) – it was indeed a conscious decision to present the relationship between CNDD and latitude using the univariate meta-regression, which is motivated directly from the Janzen-Connell hypothesis. Panel a and b test two individual predictions and thus rely on two different meta-regressions.

What explains this seemingly odd presentation and interpretation of a significant interaction? It is likely that the authors were interested in comparing mean CNDD across latitudes without incorporating information about species abundances for some reason. However, this is not appropriate statistically or conceptually. First, it is clear that CNDD varies with species abundance, and not incorporating this information when calculating mean CNDD in each forest plot adds extra error or “noise” to the calculation of each plot's mean CNDD. Second, mean species abundance changes across plots, going from lower values at tropical plots to higher values at temperate plots. This means that the meta-regression model that does not incorporate known variation in CNDD with species abundances (presented in Fig. 2a) compares CNDD of rarer tropical species to more common

temperate species. Note that CNDD is actually greater for common than for rare species at temperate latitudes (Fig. 2b in the manuscript and Fig. R1a & R1b above). **Thus, by comparing mean CNDD across latitudes in a model without species abundance, CNDD is compared at different species abundances.** It is statistically and conceptually best to compare CNDD across latitudes for species at the same abundances.

We agree with the referee's assessment that our analysis compares species communities with different abundances. However, as we detail in response A) this is what we want.

Using a fixed abundance does not improve the situation because one would compare CNDD at abundances that are comparably rare in one geographic zone and fairly common in another. This is because rare and common has a different meaning depending on latitude.

We would also like to point out that we did include a species random effect in the meta-regression so that the model accounts for variation among species (L629ff). That is, we are not ignoring variation among species in the model – we are just not explicitly including the relationship with abundance.

One main reason for this is that species with lower abundances generally experience fewer numbers of nearby conspecifics (they are rare) whereas species with greater abundances tend to experience greater numbers of nearby conspecifics (they are nearly everywhere across the plot). In other words, rare species can often (although not necessarily always) encounter a different range of conspecific densities than common species or intermediately-abundant species.

We agree that rare species tend to have fewer conspecific neighbors on average, but as we account for that in our regressions and the calculation of the average marginal effects, we don't understand how this leads to the conclusion that we should compare CNDD at a fixed abundance across latitude if abundance distributions differ between latitudes. See our response A) above.

In addition, these analyses do not include all individuals or species in these forests (some marked individuals were excluded within the plot due to boundary issues, and certainly there are other individuals and species beyond the plot boundaries in the wider matrices of forests surrounding the plots). Thus, to avoid artifacts from comparing CNDD across species of different abundances, CNDD should ideally be compared among species at similar abundances.

We do not think that boundary effects impose any limitations on the validity of our conclusions. In our calculations, individuals for which no complete densities could be calculated due to boundary effects were excluded. The remaining data points should be fully valid for the statistical analysis. Of course, it would always be better to have more observations to increase statistical power, but unlike previous studies, which had pronounced small sample size biases, we have tested the robustness of our analysis against small sample sizes and did not find any biases. Moreover, even if boundary effects were a problem, we do not see how this would be solved by comparing CNDD at a particular reference abundance across latitude.

Conceptual implications: Though there is apparent evidence for a latitudinal gradient in CNDD for species at mean and lower abundances in Fig. R1, the finding of a significant latitudinal gradient in CNDD for rarer species has important implications that should be discussed in this manuscript beyond what is already discussed therein. Specifically, previous studies have predicted that differences in CNDD for species at lower abundances may be pivotal to maintaining the latitudinal diversity gradient 1,4,5. The 2017 *Science* paper (ref 1 here) was often cited in this manuscript, so in due diligence, I read that paper again carefully. There, the authors (including many of the authors of the current manuscript) make a clear prediction that, assuming common species experience some degree of CNDD across the latitudinal gradient, stronger CNDD for rare tropical than for rare temperate species should stabilize rare species more strongly in tropical forests relative to temperate forests.

As described in our responses B) and C), we agree that the relationship of species abundance and CNDD is a 'striking' pattern (cf. L114) that seems important for understanding the role of CNDD. This is why we have given this pattern so much space in our discussion (L186-200) and put a lot of effort into revealing the underlying causality (L138ff).

However, as we explain in response A), we believe it would be misleading to present this pattern as a latitudinal trend in mean CNDD. What we rather have is a latitudinal pattern in the relationship of CNDD and abundance, with tropical plots showing a stronger relationship than temperate plots. Notably, the 2017 *Science* paper by LaManna et al. referenced by the referee discusses this pattern very similarly: "the strength of CNDD was also associated with species abundance within forest communities, but the slope of this relationship changed systematically across latitudes" (see also their Fig. 2D).

This would, in effect, contribute to greater persistence of rare species in tropical than in temperate forests 1,4,5, potentially explaining why tropical forests might represent a "museum of diversity" preserving many more rare species from extinction relative to temperate forests. Of course, other factors that differ across tropical and temperate forests, including differences in speciation rates, likely also contribute to the latitudinal diversity gradient 6. However, the findings in this manuscript offer strong confirmation of a previously published (i.e., *a priori*) prediction and suggests an important role for latitudinal differences in CNDD for rare and mean-abundance species in reducing extinction rates in tropical relative to temperate forests and contributing to the latitudinal diversity gradient.

Again, we agree that the relationship of species abundance and CNDD is a striking pattern and have dedicated an entire paragraph of our discussion to this pattern (L186-200). Nevertheless, we are hesitant to suggest that the stronger CNDD for rare species is causally responsible for higher tropical diversity for the reasons discussed in response C), i.e., stronger, but heterogenous CNDD is not necessarily stabilizing at the community level (see our reasons and references in C). Instead, it seems reasonable to us to presume that if rare species have stronger CNDD, they might be rare because their CNDD limits them from becoming common (Chisholm and Muller-Landau 2011, Yenni et al. 2012, Stump and Comita 2018) (L186ff and L234-235).

Recommendation for authors: A revised version of Fig. 2 that presents appropriate results from the meta-regression model with species abundance and latitude and their interaction (i.e., Fig. R1 above) should be included in a revised version of this manuscript. In addition,

the previously predicted and important implications of the authors' findings relative to significant latitudinal gradients in CNDD for species at rare and mean abundances should be explicitly discussed in a revised version of this manuscript. It should be highlighted that the finding of a latitudinal gradient in CNDD for rare species offers confirmation of a previous prediction about how CNDD might contribute to the latitudinal diversity gradient 1,4,5.

Thank you again. As mentioned above, we have included Fig. R1 in Extended Data Fig.5, and we prominently present the results of this model in our main figures, i.e., the latitudinal pattern in the association of CNDD and species abundance (Fig.2b) and interpret this as evidence for exactly the prediction to which the referee refers: stronger CNDD for rare species indicates more effective stabilizing control of species abundance by CNDD in the tropics (L112ff). As argued above (B), we assume the causality to work in the direction that latitudinal gradients in environment affect the relationship between CNDD ~ abundance, and thus we prefer to emphasize the original plot in our paper.

We have also cited the papers mentioned by the referee in which this prediction has previously been suggested (L113).

Other major concerns about presentation and interpretation of findings

Comparison to CNDD based on spatial patterns of sapling recruitment: The authors also explicitly compare their findings (based on CNDD in sapling survival) to a previous study (1) of CNDD based on spatial patterns of sapling recruitment (e.g., lines 93-97). They mention that differences between these two studies are striking given that the two studies used similar datasets. However, this ignores the fact that studies of CNDD using spatial patterns of sapling recruitment relative to conspecific adults incorporate many more life stages than just sapling survival – indeed, this is likely why so many other processes may potentially influence those spatial measures of CNDD 7,8. The current manuscript examines CNDD in sapling survival, but spatial patterns of sapling recruitment from adults incorporate potential CNDD in other life stages, including at least: 1) seed production, 2) dispersal, 3) germination, 4) seedling survival (up to 1 cm DBH) and 5) sapling survival. Thus, the striking thing to me was the fact that there were so many congruities between ref 1 and the current manuscript, given that the current study examines only one life stage of the many life stages examined in ref 1. Both studies found evidence for previously predicted significant relationships between latitude and CNDD for species at rare and intermediate abundances (compare Fig. 2c in ref 1 to the left panel in Fig. R1a above; note the y-axis is reversed in the two figures, so a positive relationship in the former would be the same as a negative relationship in the latter). Thus, the present manuscript seems to offer strong confirmation of a previously-made prediction and finding (i.e., latitudinal gradient in CNDD for species at lower abundances) despite examining CNDD in only one life stage. Meta-analyses suggest that CNDD may be even stronger in the seedling life stage, and previous studies have found CNDD in dispersal and other relevant life stages 2,9–11. Thus, one might expect the patterns in CNDD for sapling survival alone to be weaker or non-existent relative to those found in ref 1. All of this should be explicitly acknowledged and discussed as mentioned above.

Indeed, the two studies agree on the latitudinal pattern in the association between species abundance and CNDD, with stronger CNDD for rare than common species in tropical forests. We acknowledge this agreement in our manuscript (L134-135), but because the statistical bias of this previous study creates a correlation between CNDD and abundance even for

random data, we view the results regarding a CNDD ~ abundance correlation as “inconclusive”. We formulate this in our technical comment on this paper (Hülsmann and Hartig 2018), where we write that:

“The biased estimator questions the evidence for the reported CNDD patterns, but does not constitute proof of their absence. Yet, when combining our analysis with general ecological knowledge, we find it rather unlikely that the patterns reported in LaManna et al. are primarily caused by CNDD.”

We still stand by this assessment, which is supported by other re-analyses and methodological papers (Chisholm and Fung 2018, Detto et al. 2019, Xu et al. 2022): although the statistical problems in the previous study do not prove that the pattern is absent, it does not provide compelling evidence for the existence of the pattern either. For the same reason, we don’t agree that the effect sizes of the previous study should be used as a reference for other studies.

Moreover, there is also a crucial difference between the results of the two studies: while LaManna et al. (2017) found a striking latitudinal gradient in average CNDD with stronger CNDD in tropical than temperate forests, our analyses did not confirm such a pattern (cf. L76ff). We note that the reason may again be the regression dilution bias in this study, which creates a trivial bias towards higher CNDD in the tropics (see also Chisholm and Fung 2018, Detto et al. 2019, Xu et al. 2022).

The tradeoff between the inclusion of more life stages and processes in static data at the expense of a bias toward stronger CNDD has been discussed at length in previous papers (Chisholm and Fung 2018, Hülsmann and Hartig 2018, Detto et al. 2019, Hülsmann et al. 2021). We have therefore decided against reopening this discussion in our manuscript. We agree that ideally we would observe the direct effect of CNDD on fitness, and the use of the recruit-to-adult ratio is a better proxy for this than mortality of small trees, but the issue with analyzing recruit-to-adult ratios is the regression dilution effect, for which there is no clear statistical solution (Detto et al. 2019). As we expressed in Hülsmann et al. (2021), we find that a longitudinal analysis estimating CNDD in mortality, although less tightly linked to fitness, is preferable, because it is statistically far better controlled.

We do agree with the referee that earlier life stages, especially the seedling stage, may reveal different or stronger CNDD effects and could possibly show a latitudinal signal. We have acknowledged this more explicitly in our discussion and endorse future studies investigating a latitudinal gradient in CNDD using repeated seedling data (L209-213).

Reporting of relatively weak CNDD across gradient: The authors state on line 75 and elsewhere that they observed relatively weak CNDD. However, they fail to mention in the main manuscript that the strength of CNDD shown is largely dependent on an arbitrarily chosen standard increase in conspecific density. This is mentioned in the methods (lines 567-568), but the usual reader may not read the detailed methods. This arbitrarily chosen standard increase in conspecific density is equivalent to adding one 2 cm stem at 1 m distance from a focal individual. Thus, the relatively weak values of CNDD reported (i.e., decrease in annual mortality probability associated with a standardized increase in conspecific density) are partially due to the addition of a small conspecific stem whose

distance from the focal individual is 50 times greater than the stem's diameter ($1\text{ m} = 2\text{ cm} * 50$). Perhaps the 2 cm stem size was chosen to incorporate a stem size that is common to most if not all species, but this was not clearly described. Also, the rationale for the standard distance of 1 m was not justified. Why place the stem at 1 m? Why not place it at 2 cm or 10 cm distance? Clearly, if the 2 cm stem was placed closer to the focal individual (or a larger stem size chosen), then the values of CNDD would be much higher than those reported here. Thus, it seems odd to speak of relatively weak CNDD when the observation of weak CNDD is largely due to arbitrary choices made during the analysis. I recommend that the authors carefully justify their arbitrary choices here (i.e., why 2 cm and why 1 m distance?) and discuss the implications of those choices (i.e., what if a closer distance had been chosen, how would that affect the interpretation of relatively weak CNDD?).

We thank the referee for raising this issue, because it showed us where we can clarify the text so that misunderstandings are avoided.

1) When we wrote “the change in CNDD across latitude was not statistically significant and, more importantly, was relatively weak”, we did not comment on the absolute strength of CNDD, but only on the strength of the *change* of CNDD across latitude. Moreover, to decide what is weak, we should have added what our reference is. What we meant is that latitudinal effects are weak compared against interspecific variability and other effects. In particular, latitude explains much less variation than abundance. We have rephrased the latter part to “the change in CNDD across latitude was not statistically significant and also relatively small compared to the variation in CNDD across species and abundances (see next subsections, Fig.1, 2b, 3)” (L84-86).

2) To make the previous statements, we choose to measure the strength of stabilizing CNDD by perturbing the community by a small difference in conspecific density. This is because what we effectively want is to calculate the derivative of mortality with respect to density around the reference value by making a small perturbation (see also comments to Rev 1). The chosen ‘density perturbation’ ensures that we are staying within the observed density range even for most of the rare species, and we are therefore not extrapolating outside the range of the data. To provide more clarity, we have added the risk of extrapolation as a reason for choosing one additional conspecific neighbor of 2cm at 1m distance (L587) and explicitly stated to which increase in conspecific density our estimate of stabilizing CNDD refers when we report an effect size for the first time in the main text (L78-80). We note that the choice of the perturbation does not alter “the strength in the change in CNDD across latitude compared to abundance” as it would the absolute strength of CNDD. We hope that the changes in the text now make this clear.

Inspired by this comment, we thought it would be a valuable addition to our results to provide a more indicative estimate of the average strength of CNDD, i.e., to assess how important CNDD is for small tree mortality. In this case, the comment of the referee is correct – we must decide on how much we want to change conspecific densities to calculate an effect strength. We decided to calculate absolute and relative changes in mortality probability resulting from a change in conspecific density from the first to the third quantile of observed conspecific densities per species (see Fig.C in response to R1). Note that these values should not be compared between species, as the change in conspecific density is

different for each species. Instead, the numbers provide an approximation for the importance of conspecific density to explain variability in the mortality process.

The CNDD effects between the first and third quantile ranged between -2.39 and 8.03% relative change in mortality (Fig.C in response to R1), with a mean from meta-regression of 6.6%, which we consider large enough to be potentially relevant for population growth rates and community assembly. Based on these new results, we expanded on the average strength of CNDD in the text (L74-76) and included Fig.C as Supplementary Fig.4.

Other analytical concerns

cloglog vs. piecewise exponential survival models: Analysis of mortality hazard with a cloglog link function and $\log(\text{interval})$ as an offset is an appropriate way to analyze mortality of one interval per individual (because individuals are not included more than once in the dataset). However, it appears that multiple census intervals were used for some of these ForestGEO plots. BCI had 8 censuses (7 census intervals), some plots had 3-4 intervals, and others only had 1 interval. Census number was used as a random effect in the analyses in this manuscript, but this does not completely account for repeated assessments of survival of the same individuals (i.e., in plots with more than 1 interval, individuals were included more than once in the likelihood). Complicating this is the fact that an individual random effect cannot be included in these cloglog models. One way to handle such analyses is with exponential survival models, such as piecewise exponential survival models (e.g., <https://grodrigo.github.io/glims/notes/c7s4>). This could be implemented by using a “poisson” link instead of “binomial” with cloglog link, as described in the website above, and by including a year-in-census effect (where 1 is the first year an individual is in the census, 2 is the second year, etc., regardless of the census number; i.e., an individual first surveyed in the 5th census would get a 1 for that year). I tried this quickly with the BCI data and analysis code that the authors provided – this quick trial revealed very high correlations between estimates of CNDD between the exponential survival approach and the cloglog approach and nearly similar estimates. Given the degree of similarity between estimates from these two approaches, I am not necessarily recommending the exponential survival approach over the cloglog approach. However, it might be worth checking to see if results are indeed reasonably similar using the other plots and discussing trade-offs between different approaches to modeling mortality in the methods section.

We thank the referee for pointing out the survival time models and their implementation via piecewise exponential regression. We agree that this model better corrects for the repeated assessments of survival of the same individual. A similar approach was used in a previous study (LaManna et al. 2022), and we acknowledge that this is a slightly more elegant approach to deal with the time problem.

However, we hold that for the data we have the effect is most likely negligible. The reason is that the time series of status observations are rather short (Extended Data Table 2). For nine of the twenty-three forest sites, there is only one mortality interval available, and thus there would be no difference for these sites. The site for which the referee tested the exponential survival approach, Barro Colorado Island, has the longest time series with seven intervals. Because the CNDD estimates were similar between the two approaches (based on the referee's reports), we believe it is practically certain that there would be no relevant

difference in the overall results. We thus did not perform this additional analysis but would be happy to do so if the referee or editor insist.

Twice as many tropical sites as temperate sites: I noticed that there are twice as many tropical sites as temperate sites in this analysis. I understand that permission for each plot's data must be sought for inclusion in this analysis. However, there are several other temperate plots that have completed at least 2 censuses. Was no attempt made to include more temperate plots so that the number of plots in each zone was approximately equivalent? Either way, I encourage the authors to discuss what impact this might have on their meta-regressions and findings.

We requested data from all plots that were at least ten hectares in size (but later got data from a smaller subset for one plot) and that had at least two censuses, i.e., the minimum for mortality analyses, approx. two years ago. In the ForestGEO network it is then up to the plot PIs to make the final decision whether they want to contribute their data to a particular study. We included all sites for which we received permission. We have added information on the selection of the forest plots in the methods section (L459ff).

We do not see why the current selection of ForestGEO sites in our study would lead to bias. Of course, a larger number of sites, especially temperate ones, would increase the statistical power of the analyses and would be preferable, but the meta-regressions do not require that the predictor variables are uniformly distributed. We have noted that future studies should pay particular attention to good coverage of temperate tree species in the discussion (L213-215).

Masked data for assessment and reproducibility by readers: I very much appreciate the authors' willingness to share the meta-regression data for evaluation (with species names masked). I also appreciate that the analyses codes are available on GitHub. This was quite helpful for my assessment of the analyses and their interpretation. I suspect that other readers would equally appreciate the opportunity to reproduce the findings of this study, even if using data that are masked. I recognize that this can be difficult with many different PIs involved. However, I strongly recommend that the authors have a plan to make all data necessary to completely reproduce their analyses available to readers in some form (e.g., species names anonymized) if/when the paper is ultimately published. Otherwise, I would have a major concern about reproducibility, for example, if readers of the manuscript cannot access the data directly in some form and instead have to submit 23 separate data-use requests to each set of ForestGEO plot PIs included in these analyses. In my opinion, this presents an undue burden for a reader – what if some of the plot PIs do not approve of a reader accessing their data for reproducibility? If so, then the findings would not really be reproducible. The authors of this manuscript include all relevant ForestGEO plot PIs, and I recommend that there should ideally be some agreement among the authors and plot PIs prior to acceptance that would allow some minimal but complete form of the data (e.g., species names anonymized) to be available to readers who may want to fully reproduce these findings. I know that such arrangements have been made in the past with ForestGEO data, so there is a precedent for this. I do hope that the authors see the potential benefits of such an arrangement for the widespread reproducibility and acceptance of their work.

From an "ideal science" perspective, we fully agree with the referee on this point, and we have done our best to make the analysis as transparent as possible. Consequently, we not only provide all the code for our analysis, but also extensive comments that we hope will be helpful to interact with our results, either critically or to build on them (see <https://github.com/LisaHuelsmann/latitudinalCNDD>).

At the same time, we find it important to consider that especially in a study that uses the work of many researchers from the Global South, there are other considerations that apply (de Lima et al. 2022). For example, we would like to point out that the decision to release all data would not only affect the plot PIs, but also their working groups, e.g., PhD students that currently work on studies based on these data.

With these ideas in mind, we have carefully discussed the issue of a data release among the co-authors. We propose that, in addition to the code, the data with the CNDD estimates for all individual species be made available with species names anonymized, as we did for the first review round, so that the meta-analyses are reproducible. Estimates are available at https://github.com/LisaHuelsmann/latitudinalCNDD/tree/main/reproducibility_exports. We have modified the data availability statement accordingly (L743-745).

References

1. LaManna, J. A. *et al.* Plant diversity increases with the strength of negative density dependence at the global scale. *Science* **356**, 1389–1392 (2017).
2. Comita, L. S., Muller-Landau, H. C., Aguilar, S. & Hubbell, S. P. Asymmetric density dependence shapes species abundances in a tropical tree community. *Science* **329**, 330–332 (2010).
3. Mangan, S. A. *et al.* Negative plant-soil feedback predicts tree-species relative abundance in a tropical forest. *Nature* **466**, 752–755 (2010).
4. Yenni, G., Adler, P. B. & Ernest, S. M. Strong self-limitation promotes the persistence of rare species. *Ecology* **93**, 456–461 (2012).
5. Yenni, G., Adler, P. B. & Ernest, S. K. Do persistent rare species experience stronger negative frequency dependence than common species? *Global Ecol. Biogeogr.* **26**, 513–523 (2017).
6. Mittelbach, G. G. *et al.* Evolution and the latitudinal diversity gradient: speciation, extinction and biogeography. *Ecol. Lett.* **10**, 315–331 (2007).
7. Hülsmann, L. & Hartig, F. Comment on "Plant diversity increases with the strength of negative density dependence at the global scale". *Science* **360**, eaar2435 (2018).
8. LaManna, J. A., Mangan, S. A. & Myers, J. A. Conspecific negative density dependence and why its study should not be abandoned. *Ecosphere* **12**, e03322 (2021).
9. Jansen, P. A., Visser, M. D., Joseph Wright, S., Rutten, G. & Muller-Landau, H. C. Negative density dependence of seed dispersal and seedling recruitment in a Neotropical palm. *Ecol. Lett.* **17**, 1111–1120 (2014).
10. Comita, L. S. *et al.* Testing predictions of the Janzen–Connell hypothesis: a meta-analysis of experimental evidence for distance- and density-dependent seed and seedling survival. *J. Ecol.* **102**, 845–856 (2014).
11. Song, X., Lim, J. Y., Yang, J. & Luskin, M. S. When do Janzen–Connell effects matter? A phylogenetic meta-analysis of conspecific negative distance and density dependence experiments. *Ecology Letters* **24**, 608–620 (2021).

C. Referee #3 (Remarks to the Author):

The authors test the Janzen-Connell hypothesis that host-specific herbivores, pathogens, or other natural enemies make the areas near a parent tree inhospitable for the survival of seedlings using a dataset of tree mortality along a latitudinal gradient.

We thank the referee for insightful comments that allowed us to better outline the novelty of our results and to more clearly justify our approach.

Novelty:

At several points in the article, the authors state that no relationship between CNDD and latitude was found by previous research (lines 36-38) (e.g.: 1–3) (the conclusion of the submitted manuscript), but that the statistics were not adequate. If a pattern was found with latitude, methodological limitations were again mentioned as a reason (lines 41-44). So while at this point it is certainly still a methodologically and statistically contentious discussion, the overall results presented are not necessarily new. The most novel part of the study is certainly the relationship between CNDD, rare species, and latitude. Again, the results have been shown before by another study but have been criticized.

The referee is correct that many studies in high IF journals have looked at specific aspects of the Janzen-Connell hypothesis. However, rather than diminishing the novelty of our results, we believe this underscores their importance. The JC hypothesis is, after all, one of the most important concepts in tropical forest ecology, and there is an active ongoing debate on its validity and importance with highly disparate views about CNDD and species diversity (Terborgh 2012, Cannon et al. 2021, Hülsmann et al. 2021).

Moreover, only a very small subset of these studies addressed large scale patterns in CNDD, and these studies were very controversially discussed. In our view, the issue of latitudinal patterns in CNDD is indeed largely unresolved, and our study makes several major advances in this field:

1) First of all, the previous studies that we refer to as problematic because of statistical problems did in fact NOT come to the same conclusions as we do. Crucially, the two most widely known latitudinal studies based on static data (Johnson et al. 2012, LaManna et al. 2017) reported a significant latitudinal trend in average CNDD, which contrasts with our result (Fig.2a). These differences can be explained from methodological limitations in previous work, which is known to show a bias towards stronger CNDD for rare species and thus – by the fact that tropical communities harbor more rare species – also towards the tropics. Hence, our study corrects and challenges previous results regarding a latitudinal pattern in mean CNDD. We have made this clearer in the text (L103-110).

2) We do find similar results to LaManna et al. (2017) regarding the correlation of CNDD with abundance, which we acknowledge (L134-135). We agree with R2 and R3 that this is a striking and important finding of our study. Stronger CNDD for rare species was first reported by Comita et al. (2010) on Barro Colorado Island and then also reported in a global analysis by LaManna et al. (2017). We have examined the latter study in detail (see Chisholm

and Fung 2018, Hülsmann and Hartig 2018) and showed that the methodological setup creates a statistical bias that enforces a correlation between estimated CNDD and species abundance (see also Detto et al. 2019, Xu et al. 2022). In our opinion, the results of this study therefore provide no evidence for or against the pattern, and thus a new test was urgently needed. We have revised the comparison of our results with the previous study to emphasize this aspect (L133-137). Together with our first result, the association of CNDD and rarity in the tropics support a novel, refined interpretation of the original hypothesis, as we argue in our discussion (L231ff).

3) Finally, to our knowledge as the first study ever, building on new theoretical insights (Stump and Comita 2018, May et al. 2020), we test for a latitudinal trend in the interspecific variation of CNDD within forest communities. Our results confirm previous findings that there is large interspecific variability in CNDD across all latitudes (L147ff), which questions how much stability is created by this mechanism, but we do not find a latitudinal trend in this variation. We have added that this has not been tested previously (L155-156).

We hope that these points convince the referee that our results are novel and different from previous studies in many ways. The comment has certainly helped us to better work out the novelty in the text.

Also, I would suggest the authors change the tone of the introduction a bit. At the moment, one gets the impression that all the previous studies are flawed and that only the study presented uses appropriate statistics, even though the previous authors came to the same conclusions.

We thank the referee for their comments. It was not our intention to provide the message that all previous CNDD studies have been flawed.

First of all, our critique is not towards CNDD studies in general, and not even specifically towards comparative CNDD studies, but specifically towards the few existing studies that examine large-scale gradients of CNDD (Johnson et al. 2012, LaManna et al. 2017).

Second, we do not want to give the impression that we do not appreciate these studies – indeed, they were an inspiration for us to embark on this project. However, there is an important limitation in the methodological approach employed in both studies (Chisholm and Fung 2018, Hülsmann and Hartig 2018, Detto et al. 2019, Hülsmann et al. 2021). Those limitations cast clear doubts on the validity of the conclusions of these two papers and have sparked many discussions in the community doing research on CNDD. We therefore think it was important to address the same questions using a new analysis approach.

We have revised L38-39 in our introductory statements and more clearly stated that the statistical biases are responsible for the result of stronger CNDD in the tropics (L45-48). We will gladly make further changes if the referee points out specific statements that they think are misrepresenting the accepted knowledge of the field.

We would like to emphasize that our study does contain several significant improvements compared to previous studies (see also next comment), and R1 seems to agree with us on this point.

Although the authors certainly address some of the limitations of the previous studies (e.g. between-species and between-site effects), they only test a limited set of confounding variables. I suggest that the authors instead show what is really new about the study presented.

We have clarified the novelty of the ecological results above. Regarding the methodological advances: in our view, the present study makes a number of significant methodological advances over previous studies.

In particular, our approach is conservative with respect to regression dilution (Detto et al. 2019), provides CNDD estimates that are meaningful for stabilization and species coexistence (Hülsmann et al. 2021) and comparable among species by assessing CNDD at the response scale, i.e. in mortality (Mood 2010, LaManna et al. 2022) and aggregates these estimates by accurately accounting for different uncertainties using meta-regressions (Hedges and Vevea 1998). These advantages are highlighted in L53ff. Also, we did account for confounding by tree size and total density in our mortality models. We think that confounding is less an issue in the meta-regression because we are interested in latitudinal *patterns* in CNDD, and don't make any claims about the underlying mechanisms. Moreover, where we do make more causal claims about the relationship between CNDD and abundance, we included life history strategies as potential confounders of this relationship in the meta-regressions (cf. L695ff).

Although the article is generally well written, statistically I have some comments below, which the authors might be able to answer. However, in light of my previous comments, I wonder if it would not fit better in a more specialized journal, but that's something that should be decided by the editorial board.

We hope that our explanations above regarding the ecological and the methodological novelty was able to change the referee's view on this. We believe that our work provides the most comprehensive picture to date of the large-scale importance of CNDD for tree species diversity, which we believe is of broad scientific interest.

Methods:

The authors use only two different predictors here, namely latitude and species diversity. As the authors mention, the CNDD would be higher in tropical forests than in temperate forests. Although latitude is an indicator for tropical and temperate forests, it is not necessarily the one that best fits the theory. The Janzen-Connell hypothesis states that these are tropical rainforests, which in this case is a categorical variable and not a continuous one. Why should the latitude 'within' the rainforests play a role?

We used latitude and species abundance (not diversity) in our main analyses because the predictions of the Janzen-Connell hypothesis suggest patterns for these variables and because these variables were also selected by earlier studies.

The referee probably missed the fact that we did consider potential confounders of the relationship between CNDD and species abundance (namely life history strategies, cf. L695ff) to establish evidence for a causal interpretation, i.e., that species abundances are controlled

by CNDD. We don't think it is necessary to account for confounding when testing for a latitudinal gradient, since we consider latitude to be a proxy for other variables (see next paragraph).

We thank the referee for stimulating us to think about how to test for latitudinal patterns in CNDD and whether the choice of latitude as a continuous predictor is appropriate. We agree that the original papers (Janzen 1970, Connell 1971) speak about lowland tropical forests against temperate forests, but we don't think that the authors necessarily envisioned that the comparison only holds between those specific groups which, after all, also do not form monolithic vegetation types but could be further subdivided. Instead, we believe that if JC effects exist, gradients in present and likely past climate (e.g., temperature, moisture, and seasonality) would be causally responsible for those via their influence on the degree of specialization of herbivores and pathogens. This is also the reason why we think that CNDD could vary across environmental gradients within the tropics. Latitude is an excellent proxy for such environmental gradients (Pontarp et al. 2018), and it has been used in all previous studies that explored CNDD patterns at large spatial scales (Johnson et al. 2012, Comita et al. 2014, LaManna et al. 2017, Song et al. 2020). We have explained the use of absolute latitude as a proxy for environmental gradients in L643-646.

Nevertheless, to address the comment, we have now also tested whether a categorical test of latitude using geographic zones would change our conclusions (see below).

It is also unclear why only these two predictors were tested. The sample size of 23 is also relatively small, which might explain to some extent why a) the significance is low and b) more predictors were not used. It is clear that there is much more data behind it at each site, since time series of tree mortality are used, but in the end, the conclusions are based on 23 data points.

This is a misunderstanding, probably caused by the grey points in Fig.2a. We fitted species-site-specific mortality models and therefore have in total 2534 data points for the analysis. These data points are structured in 23 plots (which we consider as random effects in the models). As we explain in the figure caption, the grey points in Fig.2a are mean CNDD estimates per forest site from site-specific meta-regressions, each relying on several species-specific CNDD estimates. So, the global meta-regressions are based on 2534 data points. The low significance is not explained by the low number of replicates, but rather by the high variability of CNDD estimates among species. We have specified the number of data points for the meta-regressions in the captions of both Table 1 (L376) and Fig.2 (L418).

See our explanation above for why the two variables, i.e., absolute latitude and species abundance, were considered (and when we included life history strategies as potential confounders).

As the authors mention, the data set is unbalanced, most likely also with regard to covariates. Thus, 22 data points are really at the bottom of the list of observations needed to prove that there is no effect at all.

The referee is right that there are indeed more data points in the tropics (cf. L674-675, Supplementary Fig.1c). However, we do not think that the data is unusually unbalanced for

an ecological study on latitudinal gradients. Crucially, as explained above, our meta-regressions are based on 2534 (not 23) data points (see Table 1), which are clustered across 23 sites. We account for the clustering by using random effects in the meta-regressions. Additionally, we have rerun the meta-regressions excluding the most influential species-site-specific CNDD estimates as a robustness check and obtained similar patterns and significance levels (L675-680). The data is relatively balanced with respect to species abundance (Supplementary Fig.1c).

The authors have tested the robustness of the model, but they cannot verify precisely whether the fact that they do not observe a significant effect is simply due to the small sample size. On this point, the authors need to state more clearly in a revised manuscript that they do not observe low significance solely because of the small sample size.

Again, the sample size is not as small as the referee assumes, and we do not think that the study was underpowered to detect large latitudinal CNDD differences. This is evidenced by the fact that we do find a significant latitudinal effect in the interaction with abundance. In comparison to the abundance effect, the effect of latitude is simply much smaller (cf. Fig.2).

It is an inherent feature of Null hypothesis Significance Tests (NHST) that we cannot obtain a p-value for whether an effect is really absent, but we can consider the magnitude and confidence interval of the latitude effect (Table 1a), and thus conclude that an effect, if it exists, is rather small. Based on our results, we can thus rule out large latitudinal gradients in CNDD.

To clarify this, we now state in the manuscript not only that the change of CNDD with latitude is not statistically significant, but also that it is small compared to the abundance effect (L85-86).

Furthermore, the pattern they observe between rare species, CNDD, and latitude could be confusing here, as with an additional factor in the meta-regression, the required sample size would be even higher.

We agree with the statement that adding an additional predictor increases requirements on the sample size to obtain adequate power, but the significant interaction between latitude and abundance as well as the confidence intervals on the effects indicate that our study is not underpowered.

To publish this manuscript in Nature, the authors should:

1. Clearly show how their results are new from what we already know. At the moment this is not clear.
2. Show that the low sampling size does influence their conclusions here. The sampling really is at the lower side to answer the question if the latitudinal gradient does have an influence or not.

Thank you. We have addressed both comments above.

Minor comments:

Line 13: What is stabilized here? Please clarify.

This information has been added in L9.

Line 26: Stable with respect to what exactly? Please clarify.

This statement refers mostly to temporal stability. We have added this (L28).

Line 45: Since you use latitude in the manuscript geographically, longitudinal might be misunderstood here.

We agree with the referee that the term may be confusing in the context of a latitudinal study (although it is technically the more correct term). We have replaced 'longitudinal' with 'dynamic' and provided an explanation in parenthesis where we clarify that this data corresponds to 'longitudinal tree data' (L50-51).

Line 75: What is relatively weak in numbers? What would you expect to be strong? Could you give an example. E.g. a strong effect would be....

We refer to the latitudinal difference in CNDD here, which is in comparison to the interspecific variability and the effect of species abundance relatively weak. We have clarified this (L85-86).

Line 135-137: This seems counterintuitive here. If CNDD is more effective in the tropics in controlling abundances, would that on another scale also spill over to diversity?

In testing whether there is a stronger correlation between species abundance and CNDD in the tropics, we consider that CNDD may have greater efficiency in controlling species abundances and thus potentially community assembly. We discuss this result further in the 'Discussion' (L186ff). However, such an observation does not necessarily imply that CNDD is responsible for species coexistence and greater diversity (Stump and Comita 2018, Broekman et al. 2019). We therefore prefer a more cautious interpretation of the results, especially in combination with the finding that interspecific variability in CNDD is considerably high and may even be destabilizing.

Line 168: Strictly speaking you are testing for latitude only (continuous), not tropical vs. temperate (categorical). Please clarify.

The referee is correct that we test for a continuous latitudinal trend in CNDD – a prediction that we believe derives from the Janzen-Connell hypothesis (see above). In some cases, when presenting and interpreting our results, we directly compare tropical and temperate forests, which follows directly from the slope estimates of latitude, because we feel that this comparison is very descriptive to the reader and can be well articulated. We do not see a problem with this presentation and think it is common for papers testing for latitudinal gradients to refer to low latitudes as "the tropics" and higher latitudes as "the temperate zone".

To ensure that a categorical subdivision yielded a similar result, we have repeated the global meta-regression with geographic zones (Fig.E). The result is consistent with our main finding

from Fig.2a: there are no major differences in stabilizing CNDD between the latitudinal geographic zones.

Fig.E | Estimated dependence of stabilizing CNDD on geographic zones. Fitted in analogy to the model from Table 1a but with latitude represented as three geographic zones. Limits for the three geographic zones are the same as in Fig.2b.

Line 217: You do not look at ‘specialized natural enemies’ in your analysis specifically, unless you use it as a synonym here for something else. It’s part of the Janzen-Connell hypothesis, but not specifically addressed in this study specifically.

That is correct. We have added that, more broadly, also intraspecific resource competition could be responsible for CNDD (L233).

Figure 2: Why is the y-axis going from -0.5 to 1.5? That clearly distorts the slope of the correlation. If you plot it within the CNDD range of the data you would see an increase in CNDD with decreasing absolute latitude, although not significant. Why are there no datapoints shown in b?

We deliberately chose the same y-axis limits in panel a and b, because we believe it is important to show that the variability among species with different abundances (in tropical forests) is larger than the variability across latitude (which remains insignificant no matter how we plot it). Conversely, it seems rather misleading to us to use different y-limits in the panels. For more efficient use of the plot area, we have optimized the y-limits to a range from -0.4 to 1.1 in all panels. Please also note that the actual data points vary much more (cf. Fig.1).

No additional grey points are shown in panel b because the species-site-specific CNDD estimates against species abundance are visualized in Fig.1. Note that the points in panel a are not datapoints but community averages of CNDD per forest site obtained through site-specific meta-regression (L424-426).

References.

1. Comita, L. S. et al. Testing predictions of the Janzen–Connell hypothesis: a meta-analysis of experimental evidence for distance- and density-dependent seed and seedling survival. *Journal of Ecology* 102, 845–856 (2014).
2. Hyatt, L. A. et al. The distance dependence prediction of the Janzen-Connell hypothesis: a meta-analysis. *Oikos* 103, 590–602 (2003).
3. Hille Ris Lambers, J., Clark, J. S. & Beckage, B. Density-dependent mortality and the latitudinal gradient in species diversity. *Nature* 417, 732–735 (2002).

II. References

- Broekman, M. J. E., H. C. Muller-Landau, M. D. Visser, E. Jongejans, S. J. Wright and H. de Kroon. 2019. Signs of stabilisation and stable coexistence. *Ecology Letters* 22:1957-1975.
- Brown, A. J., P. S. White and R. K. Peet. 2021. Environmental context alters the magnitude of conspecific negative density dependence in a temperate forest. *Ecosphere* 12:e03406.
- Cannon, P. G., D. P. Edwards and R. P. Freckleton. 2021. Asking the Wrong Question in Explaining Tropical Diversity. *Trends in Ecology & Evolution*.
- Chisholm, R. A. and T. Fung. 2018. Comment on “Plant diversity increases with the strength of negative density dependence at the global scale”. *Science* 360.
- Chisholm, R. A. and H. C. Muller-Landau. 2011. A theoretical model linking interspecific variation in density dependence to species abundances. *Theoretical Ecology* 4:241-253.
- Comita, L. S., H. C. Muller-Landau, S. Aguilar and S. P. Hubbell. 2010. Asymmetric Density Dependence Shapes Species Abundances in a Tropical Tree Community. *Science* 329:330-332.
- Comita, L. S., S. A. Queenborough, S. J. Murphy, J. L. Eck, K. Xu, M. Krishnadas, N. Beckman, Y. Zhu and L. Gomez-Aparicio. 2014. Testing predictions of the Janzen-Connell hypothesis: a meta-analysis of experimental evidence for distance- and density-dependent seed and seedling survival. *Journal of Ecology* 102:845-856.
- Connell, J. H. 1971. On the Role of Natural Enemies in Preventing Competitive Exclusion in Some Marine Animals and in Rain Forest Trees. Pages 298–312 in P. J. d. Boer and G. R. Gradwell, editors. *Dynamics of Populations*. Centre for Agricultural Publishing and Documentation, Wageningen, Netherlands.
- de Lima, R. A., O. L. Phillips, A. Duque, J. S. Tello, S. J. Davies, A. A. de Oliveira, S. Muller, E. N. Honorio Coronado, E. Vilanova and A. Cuni-Sanchez. 2022. Making forest data fair and open. *Nature Ecology & Evolution* 6:656-658.
- Detto, M., M. D. Visser, S. J. Wright and S. W. Pacala. 2019. Bias in the detection of negative density dependence in plant communities. *Ecology Letters* 22:1923-1939.
- Hedges, L. V. and J. L. Vevea. 1998. Fixed-and random-effects models in meta-analysis. *Psychological Methods* 3:486.
- Hülsmann, L., R. A. Chisholm and F. Hartig. 2021. Is Variation in Conspecific Negative Density Dependence Driving Tree Diversity Patterns at Large Scales? *Trends in Ecology & Evolution* 36:151-163.
- Hülsmann, L. and F. Hartig. 2018. Comment on “Plant diversity increases with the strength of negative density dependence at the global scale”. *Science* 360.

- Janzen, D. H. 1970. Herbivores and the number of tree species in tropical forests. *American Naturalist* 104:501.
- Johnson, D. J., W. T. Beaulieu, J. D. Bever and K. Clay. 2012. Conspecific Negative Density Dependence and Forest Diversity. *Science* 336:904-907.
- Lamanna, C., B. Blonder, C. Violle, N. J. Kraft, B. Sandel, I. Šímová, J. C. Donoghue, J.-C. Svenning, B. J. McGill and B. Boyle. 2014. Functional trait space and the latitudinal diversity gradient. *Proceedings of the National Academy of Sciences* 111:13745-13750.
- LaManna, J. A., F. A. Jones, D. M. Bell, R. J. Pabst and D. C. Shaw. 2022. Tree species diversity increases with conspecific negative density dependence across an elevation gradient. *Ecology Letters* n/a.
- LaManna, J. A., S. A. Mangan, A. Alonso, N. A. Bourg, W. Y. Brockelman, S. Bunyavejchewin, L.-W. Chang, J.-M. Chiang, G. B. Chuyong, K. Clay, R. Condit, S. Cordell, S. J. Davies, T. J. Furniss, C. P. Giardina, I. A. U. N. Gunatilleke, C. V. S. Gunatilleke, F. He, R. W. Howe, S. P. Hubbell, C.-F. Hsieh, F. M. Inman-Narahari, D. Janík, D. J. Johnson, D. Kenfack, L. Korte, K. Král, A. J. Larson, J. A. Lutz, S. M. McMahon, W. J. McShea, H. R. Memiaghe, A. Nathalang, V. Novotny, P. S. Ong, D. A. Orwig, R. Ostertag, G. G. Parker, R. P. Phillips, L. Sack, I.-F. Sun, J. S. Tello, D. W. Thomas, B. L. Turner, D. M. Vela Díaz, T. Vrška, G. D. Weiblen, A. Wolf, S. Yap and J. A. Myers. 2017. Plant diversity increases with the strength of negative density dependence at the global scale. *Science* 356:1389-1392.
- May, F., T. Wiegand, A. Huth and J. M. Chase. 2020. Scale-dependent effects of conspecific negative density dependence and immigration on biodiversity maintenance. *Oikos* 129:1072-1083.
- Mood, C. 2010. Logistic Regression: Why We Cannot Do What We Think We Can Do, and What We Can Do About It. *European sociological review* 26:67-82.
- Nishizawa, K., N. Shinohara, M. W. Cadotte and A. S. Mori. 2022. The latitudinal gradient in plant community assembly processes: A meta-analysis. *Ecology Letters* 25:1711-1724.
- Pontarp, M., L. Bunnefeld, J. S. Cabral, R. S. Etienne, S. A. Fritz, R. Gillespie, C. H. Graham, O. Hagen, F. Hartig, S. Huang, R. Jansson, O. Maliet, T. Münkemüller, L. Pellissier, T. F. Rangel, D. Storch, T. Wiegand and A. H. Hurlbert. 2018. The Latitudinal Diversity Gradient: Novel Understanding through Mechanistic Eco-evolutionary Models. *Trends in Ecology & Evolution* 34:211-223.
- Song, X., J. Y. Lim, J. Yang and M. S. Luskin. 2020. When do Janzen–Connell effects matter? A phylogenetic meta-analysis of conspecific negative distance and density dependence experiments. *Ecology Letters*.
- Stump, S. M. and L. S. Comita. 2018. Interspecific variation in conspecific negative density dependence can make species less likely to coexist. *Ecology Letters* 21:1541-1551.
- Terborgh, J. 2012. Enemies Maintain Hyperdiverse Tropical Forests. *American Naturalist* 179:303-314.
- Wu, J., N. G. Swenson, C. Brown, C. Zhang, J. Yang, X. Ci, J. Li, L. Sha, M. Cao and L. Lin. 2016. How does habitat filtering affect the detection of conspecific and phylogenetic density dependence? *Ecology* 97:1182-1193.
- Xu, L., A. T. Clark, M. Rees and L. A. Turnbull. 2022. Estimating competition in metacommunities: accounting for biases caused by dispersal. *Methods in Ecology and Evolution*.

- Yenni, G., P. B. Adler and S. K. M. Ernest. 2012. Strong self-limitation promotes the persistence of rare species. *Ecology* 93:456-461.
- Zhu, Y., S. A. Queenborough, R. Condit, S. P. Hubbell, K. P. Ma and L. S. Comita. 2018. Density-dependent survival varies with species life-history strategy in a tropical forest. *Ecology Letters* 21:506-515.

Reviewer Reports on the First Revision:

Referees' comments:

Referee #1 (Remarks to the Author):

I have read the new version of the manuscript by Hülsmann and colleagues and the detailed letter provided by the authors in response to reviewer comments. The authors have carefully considered comments from the previous review round and have conducted additional analyses to demonstrate the robustness of their results to methodological choices. Most of my initial comments concerned the clarity of the presentation and how it could be improved. I am satisfied with changes made to the text in response to my comments. In my view, the authors have also convincingly responded to the more specific methodological concerns raised by other reviewers, and the resulting adjustments to the text make the presentation even stronger. My assessment is that the study is scientifically sound.

The question of whether the results are newsworthy enough to merit publication in Nature was raised by one of the reviewers. I tend to agree that the headline results are not as clear and striking as those associated with many articles published in Nature. Nevertheless, the study has high potential to make an important contribution to the research field. The documented lack of clear latitudinal patterns in average CNDD at this specific stage of the tree life cycle contradicts results from a previous high-profile study and are not in line with key predictions from the Janzen-Connell hypothesis. These 'negative results' could inspire others to develop and implement protocols for investigating large-scale (latitudinal) patterns in CNDD at other life stages where they might perhaps be more likely to occur. If published, the study will also set high methodological standards for future comparative studies on CNDD and its role in plant diversity maintenance.

Nature asks reviewers to provide 'a specific comment on the appropriateness of any statistical tests, and the accuracy of the description of any error bars and probability values'. I can confirm that my understanding is that the statistical tests used are indeed appropriate for testing the hypotheses addressed in this manuscript, and that error bars and probability values are accurately described in the contexts in which they are used (e.g. figure legends).

Referee #2 (Remarks to the Author):

General comments on manuscript revisions:

I thank the authors for their careful attention to the reviewers' comments. However, I still have some concerns about how the results are interpreted and presented. I present those comments below with the hope that they can further help the authors revise and improve their interesting manuscript.

Abstract: Based on the authors responses to the reviewers (details below), the abstract should state that while average CDD did not differ among latitudes, CDD was found to be stronger for species at rare and intermediate abundances. This is a finding that is somewhat hidden in Supplemental Data Fig. 5, but these are key findings that should be highlighted. Also, the abstract states (lines 19-20) that CDD did not differ, but this is incorrect given Supplemental Data Fig. 5. Average CDD did not differ, but CDD did differ for rare and intermediate abundance species. This should be clarified.

LL105-108: Here the authors state quite declaratively that the differences between Lamanna et al. 2017 and their paper are due to statistical biases in the 2017 paper. Given further studies of spatial data (e.g., Lamanna et al. 2021 Ecosphere), it is not clear that statistical biases in the earlier study are completely responsible for the inferences made there. More importantly, the static data study looked at life-history transitions from adult to sapling, whereas the current manuscript only examines sapling survival. Could not the differences between these two papers be due to the fact that the 2017 paper examined CDD across more life stages? While it is clear the authors find strong bases in the 2017 paper and subsequent re-analyses, this is not a universally agreed upon conclusion and indeed has been refuted in the literature (e.g., Lamanna et al. 2021). I suggest the authors temper these statements and acknowledge that differences in the number of life stages examined in the two analyses could have also contributed to the differences in the latitudinal gradient in average CDD.

LL 123-125: Here, the authors state that CDD was stronger for rare species in the tropics vs. the temperate zone. However, this is also the case for intermediate abundant species (Supplemental Data Fig. 5). This should be mentioned here as well.

LL 176-178: Here, the authors state that they find only indirect evidence that CDD contributes to the difference in diversity between tropical and temperate forests. However, as they state in their rebuttals, this relies on assumptions about how CDD might maintain diversity (only via average CDD) that are unresolved. Furthermore, the findings of stronger CDD for rare and intermediate species in tropical vs. temperate forests are left out of the discussion at this point, but should be included as they are important findings even if we don't know fully their impact on the latitudinal diversity gradient. Thus, this statement should be tempered.

Response to rebuttal to Reviewer 1's comments:

In response to Reviewer 1's comments on lines 496-497, the authors state the one of their reasons for not calculating species-specific μ values is that CDD slopes would not be comparable across species. However, the total density effects are calculated at a much greater distance (different μ value) than the conspecific density effects in their model – shouldn't the same rationale apply to comparisons (implicitly happening in the model) between conspecific effects and total density effects. In other words, shouldn't the same logic suggest that the μ values for conspecific density and total density be the same to allow comparison between these values (which is essential for inferring stabilizing CDD from the conspecific density parameter)?

Response to rebuttal to Reviewer 2's comments:

In their response to Reviewer 2's comments about latitudinal gradients in CDD for rare and intermediate species, the authors add a supplemental figure (Supplemental Data Fig. 5) and claim in the main text that rare species had stronger CDD in the tropics relative to rare species in the temperate zone. However, this was also the case for intermediate species. The authors agree that both rare and intermediate species had stronger CDD in the tropics than in the temperate zone (and this is shown in Extended Data Fig. 5), why is this not clearly stated in the text (the text only states that rare species had stronger CDD in the tropics but not intermediate abundance species)?

While the authors state that the Janzen-Connell hypothesis is about differences in average CDD, it is clear from the literature and from the authors own statements that we are not even sure if differences in average CDD might lead to differences in stabilization (vs. differences for rare species or variability among species). The authors also seem to still conflate their inference on average CDD with CDD overall. This is evident throughout the manuscript, for example in lines 19-23: "Overall, our results support the idea that CNDD, although not stronger, may be more effective in regulating population abundances in tropical than in temperate forests." The "although not stronger" statement here is misleading because CDD is stronger for rare and intermediate species. They could fix this by replacing "CDD" with "average CDD". I also strongly suggest that the findings of stronger CDD for rare and intermediate abundant species in tropics vs. temperate forests be more explicitly stated in the main text and abstract. Otherwise, this important finding is buried in supplemental figures and misleading statements only stating the finding for rare species.

I also fail to see why the authors are so insistent on comparing differences in average CDD as the definitive test of Janzen-Connell when they are not actually comparing average CDD. Rare species have been lumped together, and many species are not even included. This should at a minimum be acknowledged more clearly in the manuscript as potentially complicating the evaluation and interpretation of "average CDD" which is not actually an average of all species, but of a subset of species.

In addition, it seems the authors are presenting the latitudinal gradients in CDD for rare and intermediate-abundant species in supplemental data Fig. 5 due to their own assumptions of causation (stated in their rebuttal to previous Reviewer 2), namely that "we assume the causality to work in the direction that latitudinal gradients in environment affect the relationship between CNDD ~ abundance". This is an assumption, yet this assumption leads to important findings from this manuscript being buried in the supplemental figures (i.e. significant latitudinal gradients in CDD for rare and intermediate-abundance species). These assumptions, in my opinion, are not satisfactory justification for not having these important findings in the main manuscript figures, abstract, etc. I suggest the authors reconsider this decision based on an assumption which may not hold (we have no idea which way the causation goes in this instance).

Response to rebuttal to Reviewer 3's comments:

The authors state in their rebuttal to Reviewer 2 and 3 that the meta-regressions incorporate variability in CDD among species and this allows them to ignore latitude in their test of CDD changing with latitude. However, the variability in CDD among species can be explained by species abundance (or assumedly relative abundance). Even if variability among species is modeled as a random effect in the model, that variability is conveyed to uncertainty in latitudinal effects more so if it's completely random vs. a fixed effect of abundance. This should be acknowledged and discussed. I also still feel that Supplemental Data Fig. 5 should be in the main manuscript due to all of these reasons.

The authors also respond that their meta-regressions have a sample size of 2,435 species, but this is misleading. The latitudinal comparison has only a sample size of 23 plots – species within plots are at the same latitude and cannot contribute uniquely to the latitudinal pattern. This should be clarified as Reviewer 3 suggests. I also suggest that the sample size of 23 plots be added back into the manuscript (instead of $n=2435$). For the reasons above, this sample size is misleading, perhaps more so for the latitudinal comparison.

Referee #3 (Remarks to the Author):

The authors did already changed the tone of the manuscript, but overall the focus of the introduction is still too much on the statistical problems instead of the novelty of this study and the broader context. From the statistical standpoint, the authors have clarified my concerns to a large degree. What remains somehow is still to focus more on the novelty of this study. I have a few suggestions how to probably improve it a bit further. Especially the main paragraph in the introduction (Line 38-49), which is essential in my opinion, could profit from some improvements.

Line 42: I would suggest to delete everything starting at “but suffer”. You do not cite any reasons for your claim here, and the first part of the sentence is enough to make your point.

Line 45: Somehow, from your previous two sentences, it seems like that seed and seedling survival studies do not account for the full range of life stages over which you expect CNDD, plus they are time expensive and therefore large scale comparisons are difficult. Yet, large scale studies only use static data, and therefore again the full range of life stages cannot be understood.

I would in general avoid to much technical detail as in jumping directly into regression dilution at this point. This is better placed in the context of the methods.

Line 53: It states statistical pitfalls here, but what “process” is becoming undetectable by regression dilution here? The reader might not be interested that former studies had statistical pitfalls. All studies normally do. Here I would expect a sentence that rather states that dynamical mortality data allows now to look into XY aspect of CNDD. At this point you can really show where your study is novel.

Line 55-57: Again, rephrase this and state what you do new, not other studies limitations. E.g.

“Dynamical data also allows for a more detailed evaluation of stabilization and species coexistence processes, that have been identified as difficult to detect until now. Here we ... “

Line 67: This could be streamlined. You state that you test “three prediction arising from the hypothesis that CNDD is more influential for maintaining local tree species diversity in the tropics (see following

sections)...”, yet the next paragraph has a header stating: “No detectable latitudinal pattern in average CNDD”, which is the result from your study. So where are the predictions that you are actually stating. Its fine I think to state the result as a header, but then I would suggest to add the three predictions after the sentence at line 69, to make clear to the reader what you are actually testing. Practically, move Line 71-72, and give a glimpse of what the reader will expect from line 112-113 and 148-149.

Line 133: Tone this sentence down. There might be other things your study does not detect either, so better write something: “Our multi-site approach using dynamic data provides significant evidence that a large-scale pattern of stronger CNDD for rare versus common species exists in the tropics, but not in the temperate zone (Fig.2b), a pattern that has previously been reported (cite: 8), but controversy discussed due to statistical limitations (cite: 11,12), (Extended Data Fig.1b, 2b, 3b, 4b, Extended Data Table 3, 4).”

Line 148: Remove “intuition”, but rather say “predict”

Author Rebuttals to First Revision:

Revision for Nature 2022-10-17122A

I. Referees' comments:

A. Referee #1 (Remarks to the Author):

I have read the new version of the manuscript by Hülsmann and colleagues and the detailed letter provided by the authors in response to reviewer comments. The authors have carefully considered comments from the previous review round and have conducted additional analyses to demonstrate the robustness of their results to methodological choices. Most of my initial comments concerned the clarity of the presentation and how it could be improved. I am satisfied with changes made to the text in response to my comments. In my view, the authors have also convincingly responded to the more specific methodological concerns raised by other reviewers, and the resulting adjustments to the text make the presentation even stronger. My assessment is that the study is scientifically sound.

We thank the referee for their assessment that the study is scientifically sound.

The question of whether the results are newsworthy enough to merit publication in Nature was raised by one of the reviewers. I tend to agree that the headline results are not as clear and striking as those associated with many articles published in Nature. Nevertheless, the study has high potential to make an important contribution to the research field. The documented lack of clear latitudinal patterns in average CNDD at this specific stage of the tree life cycle contradicts results from a previous high-profile study and are not in line with key predictions from the Janzen-Connell hypothesis. These 'negative results' could inspire others to develop and implement protocols for investigating large-scale (latitudinal) patterns in CNDD at other life stages where they might perhaps be more likely to occur.

We appreciate the referee's supportive statement about the novelty of our results.

We would like to emphasize that, from a scientific viewpoint, we consider the 'negative result' (i.e., the lack of a latitudinal gradient in average CNDD) to be one of the most interesting results of our paper, as it challenges the traditional belief held in (tropical) ecology that CNDD is higher in the tropics.

Our other results related to abundance and interspecific variation in CNDD provide additional important insights into the role of CNDD in structuring forest tree communities and pave the way towards reconciling the old idea of stronger CNDD in the tropics (which we reject) with the intuition of many ecologists that local biotic interactions are more important for structuring tropical forest communities, compared to temperate forests.

Overall, we believe that results, if confirmed by further studies, may constitute a far greater advance for the field than a study that would have merely confirmed Janzen & Connell's original hypothesis.

If published, the study will also set high methodological standards for future comparative studies on CNDD and its role in plant diversity maintenance.

Nature asks reviewers to provide ‘a specific comment on the appropriateness of any statistical tests, and the accuracy of the description of any error bars and probability values’. I can confirm that my understanding is that the statistical tests used are indeed appropriate for testing the hypotheses addressed in this manuscript, and that error bars and probability values are accurately described in the contexts in which they are used (e.g. figure legends).

Thank you for explicitly checking the statistical representation and ensuring that it is reliable. Moreover, we appreciate your comment that our novel methodological approach to CNDD using tree mortality over multiple censuses sets the bar for future studies, which is critical given that there has been much uncertainty about the robustness of methods used to assess and compare CNDD.

B. Referee #2 (Remarks to the Author):

General comments on manuscript revisions:

I thank the authors for their careful attention to the reviewers’ comments. However, I still have some concerns about how the results are interpreted and presented. I present those comments below with the hope that they can further help the authors revise and improve their interesting manuscript.

We thank the referee for another thoughtful review. We have thoroughly considered all comments and hope that our revision will address the remaining concerns.

Abstract: Based on the authors responses to the reviewers (details below), the abstract should state that while average CDD did not differ among latitudes, CDD was found to be stronger for species at rare and intermediate abundances. This is a finding that is somewhat hidden in Supplemental Data Fig. 5, but these are key findings that should be highlighted. Also, the abstract states (lines 19-20) that CDD did not differ, but this is incorrect given Supplemental Data Fig. 5. Average CDD did not differ, but CDD did differ for rare and intermediate abundance species. This should be clarified.

We state in the abstract that CNDD-abundance relationships differ between temperate and tropical forests, and that the latitudinal gradient in this relationship suggests that CNDD affects species abundance distributions more strongly in tropical forests (L15-17). The latitudinal gradient in CNDD for rare and intermediate-abundance species that goes along with this pattern could not be explicitly mentioned in the abstract due to space constraints, but we have moved the graphical representation of this result to the main text (Fig. 3b) and devoted more space to it in the discussion (see below). As suggested by the referee, we have ensured that we refer to “average CNDD” when stating that we found no latitudinal CNDD gradient throughout the entire manuscript (e.g., L14, L68ff, L100, L180, L240).

LL105-108: Here the authors state quite declaratively that the differences between Lamanna et al. 2017 and their paper are due to statistical biases in the 2017 paper. Given further

studies of spatial data (e.g., Lamanna et al. 2021 Ecosphere), it is not clear that statistical biases in the earlier study are completely responsible for the inferences made there. More importantly, the static data study looked at life-history transitions from adult to sapling, whereas the current manuscript only examines sapling survival. Could not the differences between these two papers be due to the fact that the 2017 paper examined CDD across more life stages? While it is clear the authors find strong bases in the 2017 paper and subsequent re-analyses, this is not a universally agreed upon conclusion and indeed has been refuted in the literature (e.g., Lamanna et al. 2021). I suggest the authors temper these statements and acknowledge that differences in the number of life stages examined in the two analyses could have also contributed to the differences in the latitudinal gradient in average CDD.

We have toned down these statements, removed the direct reference to LaManna et al. (2017) and instead emphasize our findings (L101ff). We have also pointed out here (as well as later in the Discussion) that the discrepancy between our results and previous studies could result from looking at survival vs recruitment (L101-102). Further, we have made clear that our finding may be limited to survival at the sapling stage (L104).

LL 123-125: Here, the authors state that CDD was stronger for rare species in the tropics vs. the temperate zone. However, this is also the case for intermediate abundant species (Supplemental Data Fig. 5). This should be mentioned here as well.

We have added “intermediate-abundance species” and the corresponding p-value (L120-122).

LL 176-178: Here, the authors state that they find only indirect evidence that CDD contributes to the difference in diversity between tropical and temperate forests. However, as they state in their rebuttals, this relies on assumptions about how CDD might maintain diversity (only via average CDD) that are unresolved. Furthermore, the findings of stronger CDD for rare and intermediate species in tropical vs. temperate forests are left out of the discussion at this point, but should be included as they are important findings even if we don't know fully their impact on the latitudinal diversity gradient. Thus, this statement should be tempered.

We have revised this opening paragraph of the discussion to focus on the patterns identified rather than their implications for a latitudinal diversity gradient (L178ff). We do interpret the patterns at the end of the discussion but have emphasized that such conclusions reflect our own views and can only be made under certain assumptions (L236ff). In terms of the finding of strong CNDD for rare and intermediate species in tropical vs temperate forests, we have added a sentence at the end of the first paragraph acknowledging this result (L187-188).

Response to rebuttal to Reviewer 1's comments:

In response to Reviewer 1's comments on lines 496-497, the authors state the one of their reasons for not calculating species-specific μ values is that CDD slopes would not be comparable across species. However, the total density effects are calculated at a much greater distance (different μ value) than the conspecific density effects in their model – shouldn't the same rationale apply to comparisons (implicitly happening in the model)

between conspecific effects and total density effects. In other words, shouldn't the same logic suggest that the μ values for conspecific density and total density be the same to allow comparison between these values (which is essential for inferring stabilizing CDD from the conspecific density parameter)?

The reviewer is correct that while the total/stabilizing estimator is identical for the same μ values, it is slightly different from the HNDD/CNDD estimator for different μ values. We discussed this issue extensively when designing our CNDD estimator, and we believe this is a feature rather than a bug.

The advantage of our approach is that total (i.e., unspecific) density effects, which are likely due to general resource competition (space, light, nutrients) and should be caused equally by conspecific and heterospecific individuals, are treated separately from excess conspecific density effects, which are presumably biotically mediated (i.e. CNDD-HNDD in previous analyses). Since the two underlying processes are completely different, it is natural to assume that they should operate at different spatial scales. Our grid search approach supports this assumption (L563).

When fitting CNDD and HNDD separately, however, there is no option to separate the μ values of the two processes, because the CNDD estimator subsumes both resource competition and actual stabilizing CNDD. Moreover, to be able to compare both estimators, one would indeed prefer that they measure CNDD at the same spatial scale.

Thus, while we agree with the reviewer that the effect of different μ values is slightly different in the two approaches, we believe that our choice of first correcting for general (total) density effects and looking for excess CNDD is both statistically and biologically preferable, because it directly estimates the stabilizing component of CNDD (i.e., stabilizing CNDD = CNDD - HNDD) and allows considering different scales for general and biotic density effects.

Response to rebuttal to Reviewer 2's comments:

In their response to Reviewer 2's comments about latitudinal gradients in CDD for rare and intermediate species, the authors add a supplemental figure (Supplemental Data Fig. 5) and claim in the main text that rare species had stronger CDD in the tropics relative to rare species in the temperate zone. However, this was also the case for intermediate species. The authors agree that both rare and intermediate species had stronger CDD in the tropics than in the temperate zone (and this is shown in Extended Data Fig. 5), why is this not clearly stated in the text (the text only states that rare species had stronger CDD in the tropics but not intermediate abundance species)?

We do not disagree with this – in our mind, species were on a range between rare and common, and thus we referred to the “rare” side as showing the pattern the reviewer refers to. It is true that the pattern is statistically significant up to intermediate community abundances. In the results and discussion, we now report the result for both rare and intermediate abundant species (L120-122, L187-188).

While the authors state that the Janzen-Connell hypothesis is about differences in average

CDD, it is clear from the literature and from the authors own statements that we are not even sure if differences in average CDD might lead to differences in stabilization (vs. differences for rare species or variability among species). The authors also seem to still conflate their inference on average CDD with CDD overall. This is evident throughout the manuscript, for example in lines 19-23: "Overall, our results support the idea that CNDD, although not stronger, may be more effective in regulating population abundances in tropical than in temperate forests." The "although not stronger" statement here is misleading because CDD is stronger for rare and intermediate species. They could fix this by replacing "CDD" with "average CDD". I also strongly suggest that the findings of stronger CDD for rare and intermediate abundant species in tropics vs. temperate forests be more explicitly stated in the main text and abstract. Otherwise, this important finding is buried in supplemental figures and misleading statements only stating the finding for rare species.

As suggested, we now specify "average CNDD" where applicable (e.g., L14, L68ff, L100, L180, L240) and have more explicitly stated the latitudinal CNDD gradient for rare/intermediate abundant species in the main text (L120-122, L187-188) and Fig. 3b. We have also added this representation, i.e., CNDD as a function of latitude for three abundances, to all Extended Data Figures where we evaluate the robustness of our analysis pipeline. However, given the limited word count for the abstract, we were not able to include in the abstract that the significant interaction of abundance and latitude, which we discuss in L15-17, results also in stronger CNDD in tropical versus temperate forests for rare and intermediate-abundant species.

I also fail to see why the authors are so insistent on comparing differences in average CDD as the definitive test of Janzen-Connell when they are not actually comparing average CDD. Rare species have been lumped together, and many species are not even included. This should at a minimum be acknowledged more clearly in the manuscript as potentially complicating the evaluation and interpretation of "average CDD" which is not actually an average of all species, but of a subset of species.

By no means do we insist on comparing differences in average CNDD across latitudes. From the outset, we tested three hypothesized patterns that were designed to elucidate the different ways in which CNDD might affect species abundances and diversity.

Our understanding of their seminal papers is that Janzen & Connell hypothesized the first pattern (average CNDD) to be the relevant explanation for latitudinal differences in community richness. We therefore put some emphasis on the fact that this "classical" idea does not seem to be supported by our data. However, by this we did not mean to lessen the emphasis on the other patterns.

We hope that the restructured figures (Figs. 2-4), each testing one hypothesized pattern, better reflect this and give equal weight to each test.

We do now acknowledge that very rare species do not contribute to the results to the same extent as species with more observations due to data limitations (L602-604). We would like to emphasize however that it is not true that "many species are not even included". Only in the rare case that mortality models for the rare species groups did not converge, those

species did not enter the analysis pipeline. We provide this information in Extended Data Table 2.

In addition, it seems the authors are presenting the latitudinal gradients in CDD for rare and intermediate-abundant species in supplemental data Fig. 5 due to their own assumptions of causation (stated in their rebuttal to previous Reviewer 2), namely that “we assume the causality to work in the direction that latitudinal gradients in environment affect the relationship between CNDD ~ abundance”. This is an assumption, yet this assumption leads to important findings from this manuscript being buried in the supplemental figures (i.e. significant latitudinal gradients in CDD for rare and intermediate-abundance species). These assumptions, in my opinion, are not satisfactory justification for not having these important findings in the main manuscript figures, abstract, etc. I suggest the authors reconsider this decision based on an assumption which may not hold (we have no idea which way the causation goes in this instance).

As suggested by the referee, we have moved the plot of CNDD vs latitude for different species abundances from the Extended Data Figures to our new Fig. 3 in the main manuscript and refer to it in the results and discussion (see above).

That being said, we disagree with the characterization that our choice of presentation was mainly subjective. The referee quotes us as saying that “we assume the causality ...” and suggest that an assumption is “not satisfactory justification” for our choice. However, we had provided justification for our assumption, in particular in the following sentence (see point (B) of the last response):

“Our interpretation is that environmental differences across latitude causally affect the CNDD ~ abundance relationship and not vice versa (technically, we view latitude as a moderator of the CNDD ~ abundance relationship).”

The reason for this interpretation is further explained in the paper, where we discuss the possible mechanisms that would give rise to the observed data. As in any observational study, we cannot ultimately prove that this causal interpretation is correct, but we do believe that it is good scientific practice to present the data in the way that seems biologically sensible to the authors. Reversely, we do not really see a plausible causal mechanism that would let abundance modify the correlation of CNDD ~ Latitude.

Response to rebuttal to Reviewer 3’s comments:

The authors state in their rebuttal to Reviewer 2 and 3 that the meta-regressions incorporate variability in CDD among species and this allows them to ignore latitude in their test of CDD changing with latitude. However, the variability in CDD among species can be explained by species abundance (or assumedly relative abundance). Even if variability among species is modeled as a random effect in the model, that variability is conveyed to uncertainty in latitudinal effects more so if it’s completely random vs. a fixed effect of abundance. This should be acknowledged and discussed. I also still feel that Supplemental Data Fig. 5 should be in the main manuscript due to all of these reasons.

The two meta-regressions (see Table 1) with and without species abundance test two individual latitudinal patterns related to the Janzen-Connell hypothesis. As noted above, we have checked that we give equal weight to the evidence we obtain from each.

Regarding the first pattern, where we want to test if average CNDD depends on latitude: The referee is right that using precision covariates, i.e. predictors that explain a relevant amount of variability in the response, can be beneficial to increase statistical power (Laubach et al. 2021). However, for testing the relationship between average CNDD and latitude, species abundance is not suited as a precision variable, because it is potentially influenced by both CNDD and latitude and thus a potential collider. Including a collider in the model could create a collider bias on the CNDD ~ Latitude relationship (Lederer et al. 2019).

Fig. 1 Directed acyclic graph for the effect of latitude on CNDD. Assuming that CNDD controls species abundance, abundance serves as a collider for the relationship between latitude and CNDD and including it in a model would result in collider bias (Lederer et al. 2019).

We have now moved Extended Data Fig. 5 to the main text (current Fig. 3) and give it more space in the discussion (see above).

The authors also respond that their meta-regressions have a sample size of 2,435 species, but this is misleading. The latitudinal comparison has only a sample size of 23 plots – species within plots are at the same latitude and cannot contribute uniquely to the latitudinal pattern. This should be clarified as Reviewer 3 suggests. I also suggest that the sample size of 23 plots be added back into the manuscript (instead of $n=2435$). For the reasons above, this sample size is misleading, perhaps more so for the latitudinal comparison.

We have now clarified that the sample size for the meta-regression corresponds to 2435 species or species groups from 23 forest sites (L383, L424-425, and Extended Data). Stating $n = 23$ would equally be misleading, so we prefer this more precise specification.

C. Referee #3 (Remarks to the Author):

The authors did already changed the tone of the manuscript, but overall the focus of the introduction is still too much on the statistical problems instead of the novelty of this study and the broader context. From the statistical standpoint, the authors have clarified my concerns to a large degree. What remains somehow is still to focus more on the novelty of this study. I have a few suggestions how to probably improve it a bit further. Especially the main paragraph in the introduction (Line 38-49), which is essential in my opinion, could profit from some improvements.

We thank the referee for these suggestions. We have edited the abstract, the main text, and especially the main paragraph of the introduction to reduce the focus on statistical problems of past studies (see L10-11, L45-46, L51ff, see also next comments).

Line 42: I would suggest to delete everything starting at “but suffer”. You do not cite any reasons for your claim here, and the first part of the sentence is enough to make your point.

As suggested, we have removed the part about statistical power to minimize technical aspects as suggested by the reviewer. Moreover, we have reworded the statement on “limited comparability” and added a reference (L42).

Line 45: Somehow, from your previous two sentences, it seems like that seed and seedling survival studies do not account for the full range of life stages over which you expect CNDD, plus they are time expensive and therefore large scale comparisons are difficult. Yet, large scale studies only use static data, and therefore again the full range of life stages cannot be understood. I would in general avoid too much technical detail as in jumping directly into regression dilution at this point. This is better placed in the context of the methods.

As suggested by the referee, we have removed the technical details (L45-46, L51-53) and explain our approach and its advantages in the Methods section (L482ff).

Line 53: It states statistical pitfalls here, but what “process” is becoming undetectable by regression dilution here? The reader might not be interested that former studies had statistical pitfalls. All studies normally do. Here I would expect a sentence that rather states that dynamical mortality data allows now to look into XY aspect of CNDD. At this point you can really show where your study is novel.

We have reformulated our approach to highlight its advantages and novelty and refer to the Methods section for the issues of previous studies and how we addressed these (L51-53).

Line 55-57: Again, rephrase this and state what you do new, not other studies limitations. E.g. “Dynamical data also allows for a more detailed evaluation of stabilization and species coexistence processes, that have been identified as difficult to detect until now. Here we ... “

As explained above, we have moved all details on the statistical pitfalls in previous studies to the Methods section (L482ff) and now focus on the advantages of our approach (L51-53).

For clarity, we would like to add that regression dilution in studies assuming proportionality (as is the case with static data) does not dilute the signal of CNDD, but rather leads to an overestimation of CNDD or the detection of CNDD where none is actually present (Detto et al. 2019). Thus, our approach based on dynamic data does not detect “more” CNDD, but rather estimates it more accurately.

Line 67: This could be streamlined. You state that you test “three prediction arising from the hypothesis that CNDD is more influential for maintaining local tree species diversity in the tropics (see following sections)...”, yet the next paragraph has a header stating: “No detectable latitudinal pattern in average CNDD”, which is the result from your study. So

where are the predictions that you are actually stating. Its fine I think to state the result as a header, but then I would suggest to add the three predictions after the sentence at line 69, to make clear to the reader what you are actually testing. Practically, move Line 71-72, and give a glimpse of what the reader will expect from line 112-113 and 148-149.

As suggested, we now specify the three indicators for which we examine latitudinal patterns at the end of the introduction (L62ff). Also, we have reworded the reference to the three subsections (L65). We prefer not to explain the full rationale for each predicted pattern here, leaving those explanations for the individual subsections where the patterns are tested and discussed in more detail (see L68ff, L107ff and L152ff).

Line 133: Tone this sentence down. There might be other things your study does not detect either, so better write something: "Our multi-site approach using dynamic data provides significant evidence that a large-scale pattern of stronger CNDD for rare versus common species exists in the tropics, but not in the temperate zone (Fig.2b), a pattern that has previously been reported (cite: 8), but controversy discussed due to statistical limitations (cite: 11,12), (Extended Data Fig.1b, 2b, 3b, 4b, Extended Data Table 3, 4)."

As suggested by the referee and in response to R2, we have reworded the sentences and instead of criticizing previous studies, we emphasize the advantages of our approach (L130-136).

Line 148: Remove "intuition", but rather say "predict"

We have rephrased the entire sentence (L152ff).

II. References

- Detto, M., M. D. Visser, S. J. Wright and S. W. Pacala. 2019. Bias in the detection of negative density dependence in plant communities. *Ecology Letters* 22:1923-1939.
- LaManna, J. A., S. A. Mangan, A. Alonso, N. A. Bourg, W. Y. Brockelman, S. Bunyavejchewin, L.-W. Chang, J.-M. Chiang, G. B. Chuyong, K. Clay, R. Condit, S. Cordell, S. J. Davies, T. J. Furniss, C. P. Giardina, I. A. U. N. Gunatilleke, C. V. S. Gunatilleke, F. He, R. W. Howe, S. P. Hubbell, C.-F. Hsieh, F. M. Inman-Narahari, D. Janík, D. J. Johnson, D. Kenfack, L. Korte, K. Král, A. J. Larson, J. A. Lutz, S. M. McMahon, W. J. McShea, H. R. Memiaghe, A. Nathalang, V. Novotny, P. S. Ong, D. A. Orwig, R. Ostertag, G. G. Parker, R. P. Phillips, L. Sack, I.-F. Sun, J. S. Tello, D. W. Thomas, B. L. Turner, D. M. Vela Díaz, T. Vrška, G. D. Weiblen, A. Wolf, S. Yap and J. A. Myers. 2017. Plant diversity increases with the strength of negative density dependence at the global scale. *Science* 356:1389-1392.
- Laubach, Z. M., E. J. Murray, K. L. Hoke, R. J. Safran and W. Perng. 2021. A biologist's guide to model selection and causal inference. *Proceedings of the Royal Society B* 288:20202815.
- Lederer, D. J., S. C. Bell, R. D. Branson, J. D. Chalmers, R. Marshall, D. M. Maslove, D. E. Ost, N. M. Punjabi, M. Schatz, A. R. Smyth, P. W. Stewart, S. Suissa, A. A. Adjei, C. A. Akdis, E. Azoulay, J. Bakker, Z. K. Ballas, P. G. Bardin, E. Barreiro, R. Bellomo, J. A. Bernstein,

V. Brusasco, T. G. Buchman, S. Chokroverty, N. A. Collop, J. D. Crapo, D. A. Fitzgerald, L. Hale, N. Hart, F. J. Herth, T. J. Iwashyna, G. Jenkins, M. Kolb, G. B. Marks, P. Mazzone, J. R. Moorman, T. M. Murphy, T. L. Noah, P. Reynolds, D. Riemann, R. E. Russell, A. Sheikh, G. Sotgiu, E. R. Swenson, R. Szczesniak, R. Szymusiak, J. L. Teboul and J. L. Vincent. 2019. Control of Confounding and Reporting of Results in Causal Inference Studies. Guidance for Authors from Editors of Respiratory, Sleep, and Critical Care Journals. *Annals of the American Thoracic Society* 16:22-28.

Reviewer Reports on the Second Revision:

Referees' comments:

Referee #2:

Remarks to the Author:

The authors have made adequate changes and revisions to their manuscript in response to my previous comments, and I feel they have addressed those previous comments satisfactorily. This is an important paper with findings that will move the field of ecology forward. I only have a few minor suggestions the authors may want to consider before publication.

1. The authors state in their response that they did not have space in the abstract to include some of the most important results (in my opinion), specifically, findings of a latitudinal gradient in CNDD for species at rare and intermediate abundances. However, I think an exception to the space limits of the abstract should be made in this case, this is an important finding, and one that can be written as a brief sentence. I will also add my agreement with another reviewer that the sentence in the abstract about methodological limitations in previous studies is unnecessary and detracts from the novelty of this study, perhaps that sentence could be removed in the abstract and a sentence highlighting one of the most important findings (latitudinal gradient in CNDD for species at rare and intermediate abundances) could be added.

2. I very much appreciate the authors clarifying throughout their manuscript that while there was no evidence of differences in average sapling survival CDD across latitudes, there was a latitudinal gradient in CDD for species at rare and intermediate abundances. This is an important finding. I also appreciate that the authors moved this result to a main figure (Fig. 3). The only place where I felt the text was still a little misleading on this was the discussion section paragraph beginning on line 204. Here, the authors state that they found stronger CNDD for rare species in the tropics, and that motivates further research targeted at underlying mechanisms. However, stronger CNDD was found not only for rare species, but also for species at intermediate abundances (there was a latitudinal gradient essentially for over 60% of the species in their study). Just mentioning the finding of a latitudinal gradient for rare species here potentially misleads a reader and downplays these important findings. Indeed, a finding of stronger CNDD for species at intermediate abundance also warrants further research targeted at underlying mechanisms. Could the authors please add "for species at rare and intermediate abundances in the tropics..." instead of what is currently written in line 204 and amend the rest of the paragraph as appropriate.

Referee #3:

Remarks to the Author:

This manuscript has now been through two rounds of review during which the authors have done a tremendous job in addressing my concerns. My main remaining concern from the last round of reviews

was that the manuscript was still too technical and focusing on statistical problems of former studies. I can see now that the authors have toned down the statistical criticism of other studies, which actually helped to better show the novelty of their study.

I do not have any further provisions and would recommend the paper for publication.

Author Rebuttals to Second Revision:

Revision for Nature 2022-10-17122B

I. Referees' comments:

A. Referee #2 (Remarks to the Author):

The authors have made adequate changes and revisions to their manuscript in response to my previous comments, and I feel they have addressed those previous comments satisfactorily. This is an important paper with findings that will move the field of ecology forward. I only have a few minor suggestions the authors may want to consider before publication.

We thank the referee for their helpful feedback and detailed comments during the review process, which helped us to improve our manuscript.

1. The authors state in their response that they did not have space in the abstract to include some of the most important results (in my opinion), specifically, findings of a latitudinal gradient in CNDD for species at rare and intermediate abundances. However, I think an exception to the space limits of the abstract should be made in this case, this is an important finding, and one that can be written as a brief sentence. I will also add my agreement with another reviewer that the sentence in the abstract about methodological limitations in previous studies is unnecessary and detracts from the novelty of this study, perhaps that sentence could be removed in the abstract and a sentence highlighting one of the most important findings (latitudinal gradient in CNDD for species at rare and intermediate abundances) could be added.

As previously stated, we do not agree with the notion that the latitudinal gradient for species at rare and intermediate abundances is one of the most important results of our study. We prefer presenting this finding in the abstract as a stronger control of species abundance in the tropics, which we believe has a better theoretical basis. We have the impression that the reviewer wants us to include such a statement to emphasize the part of the evidence that supports a latitudinal gradient of CNDD. However, since only one of the three hypothesized patterns provides evidence for this claim, we think that mentioning both the stronger control of species abundances in the tropics AND the latitudinal gradient in rare/intermediate abundant species (two sides of the same coin) would result in an unbalanced

presentation of our findings. In addition, we believe that the sentence about the methodological limitations of previous studies is crucial to express the novelty of our study (see also R3, who is happy with our new toned-down presentation of these limitations).

2. I very much appreciate the authors clarifying throughout their manuscript that while there was no evidence of differences in average sapling survival CDD across latitudes, there was a latitudinal gradient in CDD for species at rare and intermediate abundances. This is an important finding. I also appreciate that the authors moved this result to a main figure (Fig. 3). The only place where I felt the text was still a little misleading on this was the discussion section paragraph beginning on line 204. Here, the authors state that they found stronger CNDD for rare species in the tropics, and that motivates further research targeted at underlying mechanisms. However, stronger CNDD was found not only for rare species, but also for species at intermediate abundances (there was a latitudinal gradient essentially for over 60% of the species in their study). Just mentioning the finding of a latitudinal gradient for rare species here potentially misleads a reader and downplays these important findings. Indeed, a finding of stronger CNDD for species at intermediate abundance also warrants further research targeted at underlying mechanisms. Could the authors please add “for species at rare and intermediate abundances in the tropics...” instead of what is currently written in line 204 and amend the rest of the paragraph as appropriate.

We have now expanded and clarified our statement to include also species of intermediate abundance (L 205-207).

B. Referee #3 (Remarks to the Author):

This manuscript has now been through two rounds of review during which the authors have done a tremendous job in addressing my concerns. My main remaining concern from the last round of reviews was that the manuscript was still too technical and focusing on statistical problems of former studies. I can see now that the authors have toned down the statistical criticism of other studies, which actually helped to better show the novelty of their study.

We thank the referee for their positive feedback and the thoughtful comments during the review rounds, which were helpful in improving our manuscript.

I do not have any further provisions and would recommend the paper for publication.